# Efficient Few-Shot Continual Learning
# in Vision-Language Models

**Aristeidis Panos**                                                         *ap2313@cam.ac.uk*
*University of Cambridge*

**Rahaf Aljundi**                                               *rahaf.al.jundi@toyota-europe.com*
*Toyota Motor Europe*

**Daniel Olmeda Reino**                                 *daniel.olmeda.reino@toyota-europe.com*
*Toyota Motor Europe*

**Richard E. Turner**                                                        *ret26@cam.ac.uk*
*University of Cambridge*

**Reviewed on OpenReview:** *https://openreview.net/forum?id=sQ1w92WWOV*

## Abstract

Vision-language models (VLMs) excel at tasks like visual question answering and image captioning, but their reliance on frozen, pretrained image encoders like CLIP often leads to persistent vision errors that degrade downstream performance. Moreover, real-world deployment demands that VLMs continually adapt to new, scarce data in a few-shot setting without forgetting prior knowledge. To meet these challenges, we introduce LoRSU (Low-Rank Adaptation with Structured Updates), a lightweight and robust technique for few-shot continual learning of VLMs' image encoders. Our approach leverages theoretical insights to identify and update only the most critical parameters, achieving significant resource efficiency. Specifically, we demonstrate that LoRSU reduces computational overhead by over $25\times$ compared to full VLM updates, without sacrificing performance. In experiments on VQA benchmarks under a few-shot continual learning protocol, LoRSU demonstrates superior scalability, efficiency, and accuracy, offering a practical solution for dynamic, resource-constrained vision-language applications.

## 1 Introduction

Large Language Models (LLMs) have revolutionized natural language understanding and generation, enabling significant advancements across diverse applications. As intelligent agents are increasingly expected to operate in real-world multimodal environments, integrating visual understanding becomes essential. Vision-Language Models (VLMs) extend LLMs by incorporating visual information, either through pretrained vision encoders or end-to-end multimodal training. These models have demonstrated state-of-the-art performance in vision language tasks such as visual question answering (VQA) and image captioning, highlighting their potential for general-purpose multimodal reasoning (Chen et al., 2024; Wang et al., 2024a).

Approaches relying on pre-trained image encoders typically use variants of the CLIP model (Radford et al., 2021), which is kept frozen in the vision-language binding process (Liu et al., 2024). CLIP is a widely deployed vision transformer with strong zero-shot capabilities across various tasks and domains. However, several existing works have highlighted various weaknesses of CLIP on out-of-domain data (Liu et al., 2024; Zhu et al., 2023; Chen et al., 2023; Li et al., 2023; Tong et al., 2024). When deploying VLMs as visual assistants in new domains, it is expected that VLMs can be updated using a few images gathered from the target environment whenever deficiencies are noted.

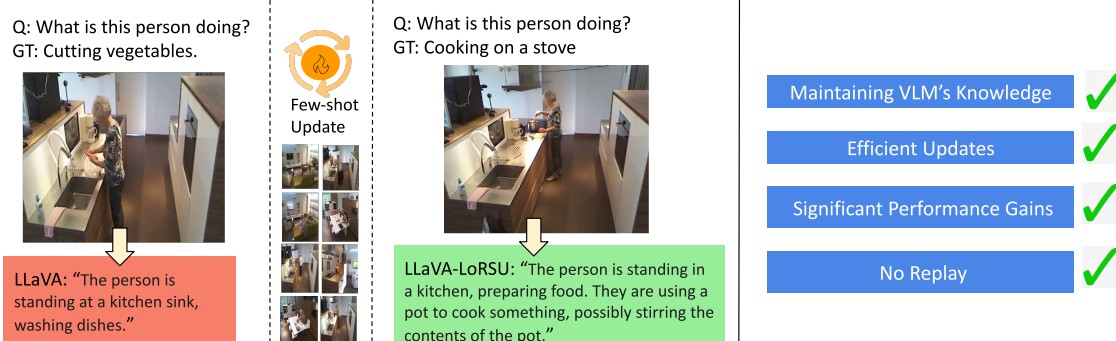

Figure 1: (1st column) Incorrect response of the pretrained LLaVA to a sample from the TSI dataset. (2nd column) A sample of few-shot data used for fine-tuning LLaVA. (3rd column) Correct response of LLaVA to a test TSI image after fine-tuning with LoRSU. (4th column) A set of desiderata for few-shot continual learning with VLMs that our method satisfies.

Continual learning allows a model to be continuously updated as new data from new tasks or domains are encountered (De Lange et al., 2021). Recent literature on continual learning (CL) of vision-language models focuses on updating either the LLM (Srivastava et al., 2024) and/or language projection layers (Das et al., 2024), while maintaining a frozen image encoder, as noted in a recent review on continual learning of VLMs (Huang et al., 2025).

In vision-language models, the LLM component provides reasoning and factual knowledge, while the image encoder's role is to extract robust and accurate visual features. In this work, we argue that adapting VLMs to new visual domains or tasks is more effective and efficient when the image encoder is updated rather than the LLM. Limiting updates to the vision encoder can lead to unprecedented stability in the performed updates.

To enable reliable few-shot continual learning (FSCL) under large vision-language models, we posit a set of desiderata capturing four main aspects: 1) Maintaining the VLM's knowledge, since models capture a vast body of generic knowledge, updates on new domains/tasks should not drastically deteriorate this knowledge (as also noted in He et al. (2023a); Zhang et al. (2024)); 2) Significant performance gains, as it has been noted that continual learning solutions, when coupled with pre-trained models, can lead to worse performance than zero-shot or very marginal improvement (Janson et al., 2022); 3) Efficient updates, given the large model sizes, few-shot updates that require a large compute budget become impractical; and 4) No replay (Rolnick et al., 2019), as replaying a set of examples that cover the model's previous knowledge can significantly increase the update memory and compute footprint.

Under this set of desiderata, we introduce a novel parameter-efficient fine-tuning (PEFT) method called LoRSU (Low-Rank Adaptation with Structured Updates). We show, for the first time, that it is possible to perform continual few-shot updates with zero replay of previous data or storage of previous models, while achieving strong performance gains of up to ($\sim 20\%$) with negligible deterioration of the model's previous knowledge ($\sim 1\%$).

Our method localizes the updates to specific layers in the vision encoder, namely the Attention and MLP layers, and selects a small set of parameters with the highest sensitivity to the received few-shot data. This approach limits the updates to a small number of relevant parameters, striking a balance between adaptability to the new task and preservation of information from previous tasks.

The third column of Figure 1 demonstrates the correct response of LLaVA after fine-tuning the image encoder separately using our method on a few-shot sample from the TSI dataset (Das et al., 2019) (shown in the second column). This is contrasted with the incorrect response of the pre-trained LLaVA, depicted in the first column.

Through extensive experiments, we demonstrate that updating the image encoder is essential for improving the performance of the VLM that relies on it. More importantly, this approach is computationally efficient,

as the image encoder has significantly fewer parameters compared to the language model, especially when updated separately. Additionally, the method is less prone to forgetting, particularly the LLM knowledge.

We evaluated our approach on various VQA tasks, comparing it to state-of-the-art CL methods and the PEFT baselines in various few-shot CL settings. We show significant improvements in the performance of the full VLM model across all settings, with very low rates of forgetting, without using any replay buffer of data from previous tasks. By selectively updating the image encoder, our method provides a robust and efficient solution for handling visual shifts. This targeted adaptation strategy avoids the need to modify the entire model, preserving existing knowledge while ensuring strong performance in new domains.

The contributions of this paper are as follows:

- We propose LoRSU, a novel replay-free PEFT method tailored for FSCL.

- We introduce two new VQA datasets, TSI and DALLE, created to expose the limitations of pre-trained image encoders in VLMs.

- We conduct the first large-scale study of FSCL in VLMs (spanning more than 700 experiments), evaluating LoRSU across ten diverse VQA datasets and benchmarking against state-of-the-art PEFT and CL methods. LoRSU consistently outperforms all baselines.

- To the best of our knowledge, this work is the first to investigate FSCL in generative VLMs, showing for the first time strong performance gains and negligible previous knowledge deterioration under the strict CL setting with no use of previous models or examples.

## 2 Related Work

**Continual Learning.** Our work falls within the CL literature, where a model needs to be updated incrementally as new data arrive, accumulating knowledge over tasks and reducing forgetting of previously acquired information De Lange et al. (2021).

**Continual Learning for Multimodal Language Models.** Recent work surveys efficiency and forgetting in continual LLM updates Wu et al. (2024) and explores VQA-based adaptation with frozen vision encoders Srivastava et al. (2024), instruction tuning via expanding projection heads He et al. (2023b). Das et al. (2024) introduced a pseudo-rehearsal strategy for vision-language models, updating only the language projection layer. Our method adapts only the vision encoder, preserving language capabilities.

**Continual Learning with Few-Shot Updates.** Verwimp et al. (2023) posits that an ideal continual learning solution would enable continual correction of model's mistakes at a lower computational cost than retraining from scratch. However, most continual few-shot learning from pre-trained models focuses on classification tasks and introduces solutions that cannot scale to large multimodal models. Panos et al. (2023) update the vision encoder on the first task only, later adapting a covariance matrix for incoming tasks. Goswami et al. (2024) calibrate the covariance matrix for new classes based on semantic similarity. Zhao et al. (2024) introduce few and slow updates, proposing a transfer loss function and a cross-classification loss to mitigate catastrophic forgetting. Few-shot updates can also be viewed through the lens of model editing Sinitsin et al. (2020). MEND Mitchell et al. (2022) scales model editing to large language models by transforming the gradient obtained from fine-tuning, through a low-rank decomposition fed to auxiliary networks designed to make fast, local edits to a pre-trained model, requiring a set of unrelated examples to prevent forgetting. ROME Meng et al. (2022) applies causal tracing to identify layers where incorrect factual knowledge is stored, applying a low-rank update. However, ROME does not scale to continual updates or non-association types of updates. Cheng et al. (2023) studied multi-modal editing, showing negligible deterioration in multi-modal task performance when updating language models but severe forgetting when updating vision encoders. To the contrary, our method focuses on adapting the vision encoder rather than updating the factual knowledge in the LLM, yet achieving strong performance gains and negligible forgetting.

**Continual Learning of Pre-Trained Image Encoders.** SPT He et al. (2023a) estimates a mask of updates based on parameter sensitivity, performing low-rank or sparse updates. SPU Zhang et al. (2024)

localizes updates to the first feed-forward layer of each transformer block, inspired by knowledge neuron theory Dai et al. (2021). Our approach generalizes updates to all layers, selecting relevant parameters and maintaining gradient norms, combined with LoRA on selected attention heads for adaptivity and stability, achieving SOTA performance on continual fewshot multimodal tasks.

Our work builds upon two key lines of research: structured sparse updates (SPU) (Zhang et al., 2024) and low-rank adaptation (LoRA) (Hu et al., 2021). Whilst SPU demonstrates the value of gradient-based parameter selection in the MLP layers, and LoRA shows the efficiency of low-rank updates, neither addresses the unique challenges of VLM continual learning where both attention mechanisms and feed-forward layers must be adapted jointly. LoRSU's contribution lies in the insight that these techniques should be applied differentially: structured sparsity for MLP layers and selective LoRA for attention heads, with head selection based on cross-parameter gradient importance.

## 3   Low-Rank Adaptation with Structured Updates

Few-shot continual learning is a highly practical and challenging scenario, where models must incrementally adapt to new tasks with limited supervision while retaining previously acquired knowledge. This setting closely mirrors real-world applications, such as interactive AI assistants and autonomous systems, where models receive a continuous stream of novel data but only sparse supervision per update.

To address the challenge of efficiently fine-tuning large-scale visual encoders and transformer-based models in a few-shot continuous learning setting, without causing catastrophic forgetting (i.e., degradation in performance on previously learned tasks), we propose a novel parameter-efficient fine-tuning method called *Low-Rank Adaptation with Structured Updates* (**LoRSU**) illustrated in Fig. 2.

LoRSU updates specific parameters within each transformer block in a resource-efficient manner, mitigating the risk of generic knowledge loss when fine-tuning for new tasks. Specifically, we selectively update a subset of parameters from the first linear layer in the MLP block of each transformer layer, as proposed in Zhang et al. (2024). Although this approach reduces the fine-tuning burden, it may limit the model flexibility as the remaining parameters in the transformer block remain fixed. To enhance flexibility, we further update the most informative attention heads based on the gradient of task-specific loss. More specifically, let a dataset $\mathcal{D}_t = \{\mathbf{x}_n, \mathbf{y}_n\}_{n=1}^{N_t}$ for the current task $t$ where $\mathbf{x}_n$ is an image with text description $\mathbf{y}_n$. We define $\mathcal{L}(\boldsymbol{\theta}; \mathcal{D}_t) := \mathcal{L}_t(\boldsymbol{\theta})$ as the loss used for training the model and $\boldsymbol{\theta} \in \mathbb{R}^d$ is the full set of model's parameters. The standard Multi-head Self-Attention Mechanism (MSA) Vaswani et al. (2017), comprised of $H$ $D_h$-dimensional heads, is defined as the concatenation of multiple self-attention (SA) blocks where $\mathbf{q}^{(i)} = W_q^{(i)} Z^\top, \mathbf{k}^{(i)} = W_k^{(i)} Z^\top, \mathbf{v}^{(i)} = W_v^{(i)} Z^\top \in \mathbb{R}^{D_h \times N}$, are the query, key and value matrices, which are used to compute the self-attention outputs as follows

$$A^{(i)} = \mathrm{softmax}(\mathbf{q}^{(i)^\top} \mathbf{k}^{(i)} / \sqrt{D_h}) \in \mathbb{R}^{N \times N}, \ \mathrm{SA}_i(Z) = A^{(i)} \mathbf{v}^{(i)^\top} \in \mathbb{R}^{N \times D_h}, \ i = 1, \ldots, H. \tag{1}$$

$Z \in \mathbb{R}^{N \times D}$ is the input matrix of $N$ tokens of dimension $D$ and $W_q^{(i)}, W_k^{(i)}$, and $W_k^{(i)}$ are the query, key, and value matrices of learnable parameters for head $i$, respectively. The final MSA function is defined as $\mathrm{MSA}(Z) = \mathrm{Concat}\left[SA_1(Z), \ldots, SA_H(Z)\right] W_o \in \mathbb{R}^{N \times D}, \ W_o \in \mathbb{R}^{HD_h \times D}$. Since we care to update the parameters of the heads that cause the largest changes in $\mathcal{L}_t(\boldsymbol{\theta})$, we compute the loss gradient with respect to the parameters of each head, and then update only those heads with the largest cumulative contribution to the loss change. Since the matrices $W_q^{(i)}, W_k^{(i)}, W_v^{(i)}$ are all the parameters of head $i$, we can define an importance score for each head by adding the squared values of their corresponding gradients $G_q^{(i)} = \nabla_{W_q^{(i)}} \mathcal{L}_t$, $G_k^{(i)} = \nabla_{W_k^{(i)}} \mathcal{L}_t$, and $G_v^{(i)} = \nabla_{W_v^{(i)}} \mathcal{L}_t$, as follows

$$s_i = \sum_{m,l} \left( (G_q^{(i)}[m,l])^2 + (G_k^{(i)}[m,l])^2 + (G_v^{(i)}[m,l])^2 \right). \tag{2}$$

We provide a theoretical justification of (2) in the next section. We update only the top-$k$ heads, based on their importance scores $\{s_1, \ldots, s_H\}$, $I \subset \{1, \ldots, H\}$, to be updated on the current task. Nevertheless,

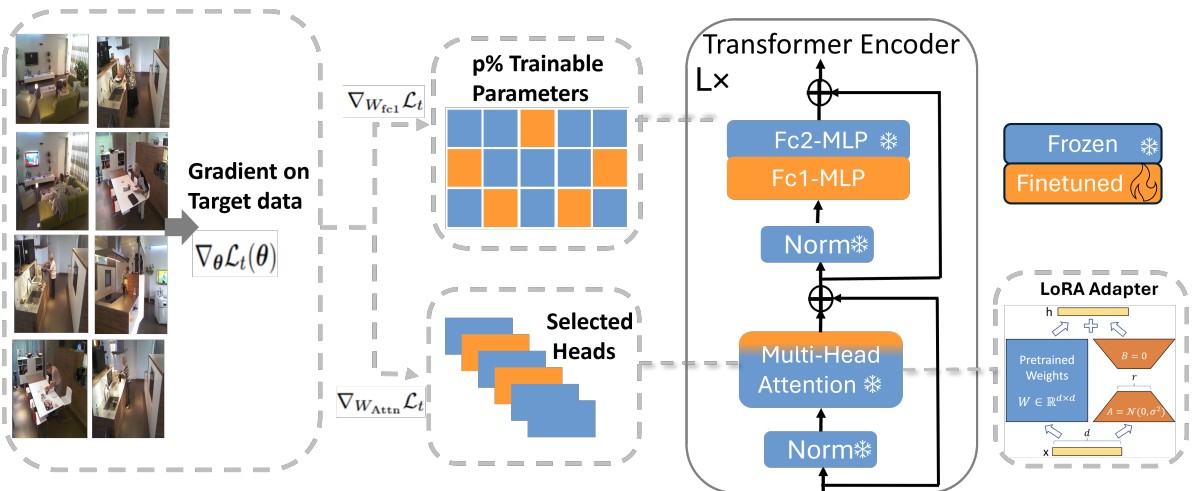

Figure 2: LoRSU mechanism: After computing the gradient $\nabla_{\boldsymbol{\theta}}\mathcal{L}_t(\boldsymbol{\theta})$ over the target dataset at time $t$, LoRSU picks a small number of attention heads and a small number of parameters from the first linear layer of the MLP module in the transformer block based on the magnitude of the gradients of $\nabla_{W_{\text{Attn}}}\mathcal{L}_t$ and $\nabla_{W_{\text{fc1}}}\mathcal{L}_t$, respectively. Computational efficiency is ensured by introducing LoRA adapters to the attention weight matrices.

the number of parameters remains high due to the large weight matrices. Therefore, we parametrize the original weights using LoRA Hu et al. (2021) to further reduce the computational burden. The matrices $W_q^{(i)}, W_k^{(i)}, W_v^{(i)}, i \in I$ are now defined as

$$W_\alpha^{(i)'} = W_\alpha^{(i)} + A_\alpha^{(i)} B_\alpha^{(i)}, \quad \alpha \in \{q, k, v\}. \tag{3}$$

Finally, to ensure that we only update $W_q^{(i)}, W_k^{(i)}, W_v^{(i)}, \forall i \in I$ we use a binary mask on the gradient vector with respect to all parameters of all attention heads. We keep the projection matrix $W_o$ frozen. We note that most modern implementations of transformer blocks concatenate the three attention weight matrices $W_q, W_k, W_v$ into one and thus we only need to apply LoRA once to this concatenated matrix.

Regarding the first linear layer in the MLP module, $W_{\text{fc1}} \in \mathbb{R}^{d \times D}$, we mask the gradients of $W_{\text{fc1}}$ so only the most important parameters for the current task to be updated, i.e. we use the following biased gradient update.

$$\hat{\nabla}_{W_{\text{fc1}}}\mathcal{L}_t = M_{\text{fc1}} \odot \nabla_{W_{\text{fc1}}}\mathcal{L}_t, \tag{4}$$

where $M_{\text{fc1}} \in \{0,1\}^{d \times D}$ is a zero-one mask that is built by choosing a proportion of the largest squared values of $\nabla_{W_{\text{fc1}}}\mathcal{L}_t$ in a similar manner as in Zhang et al. (2024) and $\odot$ is the Hadamard product.

**Theoretical justification.** The importance scores in (2) can be derived from the following constrained (binary) optimization problem[1]

$$\mathbf{p}^* = \underset{\mathbf{p} \in \{0,1\}^d}{\arg\max} \frac{\|\mathbf{p} \odot \nabla_W \mathcal{L}(\boldsymbol{\theta}_0)\|^2}{\|\nabla_W \mathcal{L}(\boldsymbol{\theta}_0)\|^2}, \text{s.t.} \bigcup_{\ell=1}^{G} I_\ell \subset \{1, 2, \ldots, d\}, \ I_i \cap I_j = \emptyset, \ \forall i \neq j, \tag{5}$$

$$\text{and} \ \ C = \sum_{\ell=1}^{G} c_\ell, \ \ c_\ell \leq |I_\ell| \ \ \forall \ell, \quad \|\mathbf{p}\|_0 \leq C,$$

where $\boldsymbol{\theta}_0$ is the vector of the pretrained parameters before using $\mathcal{D}_t$ for fine-tuning the model. The groups of parameters $I_i$ correspond to the parameters of a specific module (e.g. Self-Attention or MLP projector) we aim to learn, hence the constraint of mutually exclusiveness, $I_i \cap I_j = \emptyset$, between different pairs of parameter

---

[1]For notational simplicity, we assume a single transformer block for this case.

groups. Also note that we allowed one to choose a subset $c_\ell$ of the parameters of a specific group $I_\ell$ which is the underpinning mechanism of LoRSU choosing attention heads and parameters of fc1. The mask $\mathbf{p}^*$ is chosen so that the gradient norm of the masked gradients is as large as possible under the sparsity constraints. We prove in Appendix A that the indices of the nonzero values of $\mathbf{p}^*$ can be found using the importance scores in (2) and the magnitudes of the gradients with respect to the fc1 parameters.

**Connection to Fisher Information Matrix.** The gradient magnitude criterion in (2) is fundamentally related to the empirical Fisher Information Matrix. The squared gradient magnitudes we use correspond to the diagonal entries of the empirical Fisher, which measures parameter sensitivity to the current task. However, our approach represents a conceptually distinct and inverse paradigm compared to traditional Fisher-based regularization methods like EWC Kirkpatrick et al. (2017). Specifically, it estimates the diagonal of Fisher Information matrix for previous tasks and uses it to regularize important parameters from changing, attempting to preserve past knowledge. On the other hand, LoRSU estimates Fisher Information (via squared gradients) for the current task and uses it to select which parameters to update, allowing task-relevant parameters to adapt while freezing others. This inverse analogy makes sense precisely because we start from a strong foundation model: rather than preventing important-to-past parameters from moving, we allocate capacity for the new task by identifying and updating only parameters with high sensitivity to current data. As we show in our experimental evaluation, this type of approach significantly outperforms traditional Fisher-based regularization methods.

## 4 Experiments

We conduct a series of experiments under three different few-shot continual learning (FSCL) settings (CL-5, CL-20, and CL-50 shots) to thoroughly investigate the performance of LoRSU based on ten VQA datasets. By adopting this paradigm, we aim to assess the adaptability and efficiency of LoRSU under constrained learning conditions, ensuring that it remains both computationally feasible and effective in improving downstream performance.

### 4.1 Datasets

**Regular VQA datasets.** To capture a broad spectrum of visual shift and reasoning challenges—from classification-style tasks to open-ended question answering—we select four representative VQA datasets from the large pool of available benchmarks. These cover spatial reasoning, robust domain shifts, multimodal pattern recognition, and fine-grained perception in scientific imagery: *VSR* Liu et al. (2023), for spatial reasoning; *HM* Kiela et al. (2020), a classification-style VQA for detecting hateful memes under strong domain shift; *MMVP* Tong et al. (2024), a challenging dataset assessing multimodal visual patterns with substantial distributional shifts; *VisOnly* Kamoi et al. (2024), for fine-grained visual perception in scientific figures.

**Classification-to-VQA datasets.** We convert four classification datasets to multiple-choice VQA tasks with five answer choices: *GTS* Stallkamp et al. (2012), German traffic signs; *CAn* Wang et al. (2024b), for testing robustness to spurious features in animal images; *AIR* Maji et al. (2013), a fine-grained aircraft dataset; *ESAT* Helber et al. (2019), for land cover classification in satellite images.

**TSI & DALLE.** We introduce two novel datasets to explore domain shift deterioration indpendently from model's knowledge of present concepts: *TSI* Das et al. (2019), a classification dataset of 10K training and 5K test images of 27 activity classes; *DALLE*, generated by querying DALL · E 2, with 660 images from 22 activity classes in TSI.

For FSCL, we split each dataset into 5 sets of disjoint classes/categories and use 5/20/50 shot settings for model fine-tuning. Dataset splits are detailed in Appendix C.

### 4.2 Experimental Setting

**Metrics.** Our proposed metrics aim to reflect the desiderata outlined in the introduction: 1) maintaining the VLM's generic knowledge, 2) achieving significant performance improvements beyond zero-shot capabilities, 3) ensuring computational efficiency, and 4) avoiding replay-based methods. Standard continual learning

(CL) metrics typically measure accuracy and forgetting only within the set of adapted tasks/classes, without considering a model's pre-existing knowledge or broader capabilities. However, VLMs encapsulate extensive generic knowledge across diverse domains, making it essential to evaluate how continual adaptation affects their overall knowledge and performance.

To assess our first two desiderata, we propose two complementary metrics. First, we introduce the *Target Improvement (TI)* accuracy, which quantifies the knowledge accumulation capability by measuring the change in accuracy relative to the zero-shot performance on the test split of each target dataset after continual fine-tuning. Positive TI values indicate improvements over the pre-trained model's generic knowledge. Second, to evaluate the retention of the VLM's broader knowledge base and potential positive backward transfer, we define the *Control Change (CC)* accuracy. CC computes the average change in accuracy across a set of control datasets—datasets distinct from the current target task—to gauge whether fine-tuning leads to forgetting or, conversely, positive transfer to unrelated tasks. Both TI and CC metrics are computed after the final continual learning session. Finally, to address how accuracy and forgetting evolve throughout continual adaptation explicitly, we include standard continual learning metrics such as *Average Accuracy (ACC)* and *Backward Transfer (BWT)* (Lopez-Paz & Ranzato, 2017). Unlike TI and CC, these traditional metrics focus exclusively on adapted tasks without considering the broader generic performance of the model on other datasets.

**Implementation details.** Please see Appendix B.

**Models.** For most of our experiments, we consider the popular Vision Language Model LLaVA-v1.5 (Liu et al., 2024) that leverages a frozen CLIP image encoder. Specifically, LLaVA utilises a frozen OpenAI-CLIP-L-14 Radford et al. (2021) with a LLM (Vicuna-7b (Chiang et al., 2023)). The two modules are connected through a two-layer MLP projector that aligns image and text features. The LLM and the MLP projector are optimized during the visual instruction tuning while CLIP remains frozen. LLaVA concatenates adjacent tokens from CLIP-L-14 and processes them with an MLP projector as input to LLama-2 (7B-chat) (Touvron et al., 2023); the MLP projector and the language model are optimized while the image encoder remains frozen. Finally, we also consider MiniGPTv2 (Chen et al., 2023) that uses the same LLM as LLaVA but a frozen EVA-CLIP-g-14 image encoder. We chose LLaVA for its representative architecture, as it is widely adopted and uses CLIP-L-14, one of the most common vision encoders in VLMs, making our findings broadly relevant.

**Baselines.** We compare LoRSU to the following methods that also use the CLIP loss to fine-tune the image encoder: *LN* (Perez et al., 2018; Panos et al., 2023) is used for both few-shot and CL. Only the image encoder LayerNorm modules' parameters are optimized. *F-FT* is the standard fine-tuning technique where all image encoder parameters undergo gradient updates. *F-EWC* fine-tunes all the image encoder parameters with EWC regularization (Kirkpatrick et al., 2017). *LoRA* (Hu et al., 2021) a popular PEFT method which parameterizes incremental updates by two low-dimensional matrices and only fine-tunes them. *AdaLoRA* (Zhang et al., 2023) dynamically adjusts the low-rank update budget allocation during training. *SPU* (Zhang et al., 2024) is a PEFT baseline, specifically designed to tackle catastrophic forgetting in CL scenarios, that utilises structured sparsity based on gradient information to fine-tune the most significant parameters of the fc1 module in the transformer block.

### 4.3 Offline performance on different VLMs

In this first experiment, we compare the performance of two VLMs (LLaVA and MiniGPTv2) on TSI and DALLE. TSI data depict elderly people activities (age bias), blurred faces (blurring effect) and is captured from a mounted camera with relatively low resolution, yet the actions are easily recognizable to the human eye. DALLE that is composed of same activity classes under clear concept centred images (see Appendix F). First, results in Table 1 illustrate the visual domain shift of TSI with respect to models

Table 1: Offline fine-tuning results for MiniGPTv2 and LLaVA-1.5 under two adaptation strategies compared to zero-shot baseline (no FT): LLM-only tuning (LLaMA+Pj) and vision-encoder tuning. Best scores in each column are in bold. Updating the vision encoder separately leads to best gains.

| FT Method | MiniGPTv2 | | LLaVA-1.5 | |
|---|---|---|---|---|
| | DALLE | TSI | DALLE | TSI |
| No FT (Zr-shot) | 83.6 | 62.9 | 91.1 | 53.1 |
| LLaMA+Pj | 86.5 | 82.3 | 88.5 | 73.3 |
| Vision-encoder FT | **87.1** | **86.0** | **91.1** | **75.5** |

pretraining given the significantly lower performance of TSI compared to DALLE indicating an update is indeed on TSI to improve the performance. Next, we compare two adaptation strategies in an offline fine-tuning setting to zero-shot (no updates), LLM-only tuning (LLaMA+Pj). Across both models, tuning only the vision encoder separately yields the largest gains. These consistent improvements strongly corroborate our claim that vision-only fine-tuning is an efficient and effective strategy for adapting visual-language models under visual shift setting. For all subsequent experiments, we employ only the LLaVA model.

## 4.4 CLIP-based Updates

We evaluate the performance of the Vision-Language Model (VLM) when only the image encoder is fine-tuned using the CLIP loss in a CL setting. This experiment compares six strong CLIP-based baselines with our proposed method, LoRSU. Table 3 reports the average accuracies of TI/CC over three runs; detailed results can be found in appendix D. We observe that LoRSU consistently achieves superior TI scores across datasets and CL settings, underscoring its ability to enhance task-specific performance effectively. Furthermore, LoRSU maintains CC accuracies that take consistently small negative or even positive values, highlighting its capacity to preserve or slightly im-

Table 2: *Average accuracy* (ACC) (↑) and *backward transfer* (BWT) (↑) scores (%) across different continual learning (**CL**) setting and fine-tuning datasets (**FTD**). LoRSU achieves top ACC at BWT >-1% (i.e., close to zero forgetting). The highest scores across methods are in bold.

| Setting | FTD | LoRA | | SPU | | LoRSU | |
|---|---|---|---|---|---|---|---|
| | | ACC | BWT | ACC | BWT | ACC | BWT |
| CL-5 | GTS | 79.2 | -7.1 | 80.8 | **0.5** | **81.1** | 0.4 |
| | ESAT | 73.8 | -3.4 | 79.8 | 1.5 | **82.2** | **2.0** |
| CL-20 | GTS | 77.2 | -9.1 | 82.8 | -0.6 | **83.5** | **-0.4** |
| | ESAT | 64.1 | -18.3 | 82.0 | **2.0** | **82.7** | 0.1 |
| CL-50 | GTS | 79.3 | -10.3 | 83.8 | -0.7 | **84.7** | **-0.5** |
| | ESAT | 61.4 | -27.8 | 81.2 | -2.4 | **82.1** | **-0.5** |

prove performance on control datasets while fine-tuning on target datasets. Even in datasets where other methods struggle (e.g., CAn, ESAT), LoRSU often performs better, maintaining positive CC scores. For instance, In ESAT (CL-50) containing challenging satellite images, LoRSU achieves the highest TI (7.0) with a positive CC (0.2), outperforming SPU (TI=5.8, CC=0.1) and all other methods.

**CL metrics.** We assess the performance of LoRSU against LoRA and SPU in terms of ACC and BWT across two out-of-domain datasets, GTS and ESAT. Since LoRA and SPU have similar number of trainable parameters as LoRSU and competitive performance in our previous experiment, we choose those for comparison. Table 2 shows that LoRSU's performs well with respect to these metrics, following similar patterns as TI and CC in Table 3. Increasing the number of shots generally improves accuracy, particularly for structured update methods (SPU and LoRSU); however, this improvement comes with increased negative backward transfer for SPU, whereas LoRSU demonstrates comparatively less forgetting. *Crucially, LoRSU strikes the best balance: it leverages up to 50 shots for top-end accuracy with forgetting (negative BWT) less than 1%.* Similar patterns are observed in additional datasets in the Appendix D.3. Therefore, our results demonstrate that LoRSU effectively meets all four desiderata: (1) preserving the VLM's generic knowledge, (2) achieving substantial improvements over zero-shot performance, (3) maintaining computational efficiency, and (4) eliminating the need for replay-based methods.

## 4.5 CLIP-based vs. Perplexity-based Updates

Traditionally, LLMs and VLMs achieve impressive performance through fine-tuning with the perplexity loss. We evaluate how the CLIP-based fine-tuning methods, LoRSU and LoRA, perform compared to their perplexity-based counterparts, LoRSU-Ppl and LoRA-Ppl, respectively. Furthermore, we seek to explore how these methods compare to parameter-efficient fine-tuning approaches when the entire VLM (LoRA-F) or only the LLM component (LoRA-L) is updated. The results in Table 4 highlight the strong and robust performance of LoRSU and LoRSU-Ppl compared to other baseline methods in various settings. Both LoRSU and LoRSU-Ppl achieve minimal negative or even positive changes in CC, indicating reduced catastrophic forgetting and improved retention of generic knowledge compared to baselines. The use of the perplexity loss in LoRSU-Ppl demonstrates a considerable improvement in TI accuracy over LoRSU when fine-tuned for VQA

Table 3: Performance comparison of LoRSU with the CLIP loss against baselines fine-tuning the image encoder using the same loss. We report the *Target Improvement* (**TI** (↑)) and *Control Change* (**CC** (↑)) accuracies across three different continual learning (CL) settings and five fine-tuning datasets (**FTD**). Greener shades indicate higher positive values, while redder shades signify lower negative values. The highest accuracies across methods for each dataset are underlined. LoRSU achieves the best TI and CC.

| Setting | FTD | FT Method | | | | | | | | | | | | |
|---|---|---|---|---|---|---|---|---|---|---|---|---|---|---|
| | | LN | | F-FT | | F-EWC | | LoRA | | AdaLoRA | | SPU | | LoRSU | |
| | | TI | CC | TI | CC | TI | CC | TI | CC | TI | CC | TI | CC | TI | CC |
| | GTS | 3.5 | -1.5 | 3.7 | -6.5 | 5.0 | -11.5 | 0.7 | -4.8 | -0.9 | -4.9 | 5.4 | -0.6 | 6.4 | -0.7 |
| | TSI | 0.8 | 0.0 | 7.4 | -1.1 | 8.5 | -1.0 | -0.1 | -2.8 | 1.1 | 0.2 | 0.9 | 0.1 | 3.2 | 0.1 |
| CL-5 | CAn | -2.4 | -0.2 | -2.4 | -2.2 | -16.7 | -9.4 | -1.3 | -4.6 | -1.0 | -0.1 | -0.4 | 0.1 | 0.3 | 0.3 |
| | AIR | 0.3 | -1.6 | 2.0 | -2.7 | 2.9 | -2.8 | 1.3 | -3.7 | 0.4 | 0.0 | 3.1 | 0.1 | 4.8 | 0.4 |
| | ESAT | 4.2 | 0.6 | -10.3 | -1.4 | -8.4 | -2.1 | -1.6 | -0.7 | 1.9 | 0.1 | 4.5 | 0.1 | 6.8 | 0.2 |
| | GTS | 5.2 | -5.9 | 4.6 | -7.3 | 6.7 | -15.6 | 2.5 | -10.5 | 0.2 | -2.2 | 7.9 | -1.3 | 8.6 | -1.0 |
| | TSI | 5.1 | -1.9 | 15.3 | -3.4 | 16.0 | -32.5 | 8.5 | -4.4 | 1.3 | -9.6 | 7.8 | -0.3 | 10.6 | -0.1 |
| CL-20 | CAn | -2.4 | -0.4 | 0.3 | -2.9 | 0.1 | -5.1 | -2.3 | -5.4 | -3.5 | -2.5 | 0.1 | 0.5 | 1.1 | 0.3 |
| | AIR | -0.2 | -3.0 | 9.3 | -1.8 | 10.2 | -2.0 | 5.3 | -2.7 | 2.7 | -0.7 | 3.0 | -0.2 | 5.9 | -0.5 |
| | ESAT | 0.9 | -0.1 | -24.9 | -1.7 | -22.0 | -3.8 | -11.5 | -0.5 | -6.8 | -2.7 | 5.4 | 0.3 | 6.6 | 0.2 |
| | GTS | 4.8 | -6.5 | 3.4 | -9.8 | 5.3 | -12.9 | 3.1 | -11.1 | 1.0 | -3.3 | 7.7 | -1.5 | 9.7 | -1.3 |
| | TSI | 7.0 | -3.0 | 17.2 | -4.6 | 22.4 | -13.4 | 18.2 | -6.3 | 7.9 | -1.9 | 12.2 | -0.5 | 19.1 | -0.3 |
| CL-50 | CAn | -5.7 | -3.3 | -1.0 | -4.9 | 0.6 | -9.7 | -0.4 | -4.4 | -1.8 | -0.8 | 0.6 | -0.3 | 1.3 | -0.5 |
| | AIR | 1.8 | -3.9 | 10.0 | -3.1 | 10.9 | -3.3 | 7.8 | -3.8 | 4.6 | -0.9 | 6.2 | -0.6 | 8.2 | -0.7 |
| | ESAT | 4.6 | 0.1 | -41.4 | -3.3 | -38.1 | -2.0 | -14.5 | -3.6 | -17.3 | -2.4 | 5.8 | 0.1 | 7.0 | 0.2 |

datasets. For example, LoRSU-Ppl achieves 10% higher TI accuracy than LoRSU on VSR. We hypothesize that the perplexity loss acts as an additional signal that optimizes the image encoder to complement the frozen language model more effectively, improving the alignment between visual and textual modalities in VQA. However, we observe that LoRSU achieves a balance between task-specific improvements and generalization, consistently demonstrating higher CC accuracy compared to LoRSU-Ppl in most datasets. Updating the LLM tend to have higher TI under VSR and HM datasets compared to pure distributional shift datasets (e.g. GTS, TSI, ESAT) indicating that updating the LLM can be less optimal under distribution shift scenarios compared to the vision encoder updates suggested in LoRSU. Lastly, although LoRA-F achieves high TI scores on many datasets, it suffers significantly from forgetting, underscoring the importance of LoRSU's structured updates in CL scenarios.

## 4.6 Ablation Studies

We systematically evaluate LoRSU's design choices by varying the number of tuned attention heads $k$, the LoRA adapter rank $r$, and head-selection strategies (random vs. all-heads). We provide a summary of these results in Table 5; detailed results are reported in Appendix E. Regarding the impact of LoRA's rank we see that performance peaks at r=64 for both GTS and TSI datasets, with graceful degradation at other values, demonstrating robustness. Using $k = 2$ attention heads provides an optimal balance between performance and efficiency, with diminishing returns beyond this point. Finally, the results demonstrate that our gradient-based head selection (LoRSU) consistently outperforms both random selection (LoRSU-Rand) and updating all heads (LoRSU-AAH) across all CL settings, validating our structured update approach.

## 4.7 Computational Efficiency

Table 4: Performance comparison between LoRSU using the CLIP loss (*LoRSU*) or the perplexity loss (LoRSU-Ppl) and other baselines that fine-tune only the vision encoder (*LoRA, LoRA-Ppl*), only the LLM (*LoRA-L*), or both of them (*LoRA-F*). We report the *Target Improvement* (**TI** (↑)) and *Control Change* (**CC** (↑)) for each CL setting. † and ‡ denote classification-to-VQA and regular VQA datasets, respectively. The highest accuracies across methods for each fine-tuning dataset (**FTD**) are underlined.

| Setting | FTD | FT Method | | | | | | | | | | | |
|---|---|---|---|---|---|---|---|---|---|---|---|---|---|
| | | LoRA-L | | LoRA | | LoRSU | | LoRA-Ppl | | LoRA-F | | LoRSU-Ppl | |
| | | TI | CC | TI | CC | TI | CC | TI | CC | TI | CC | TI | CC |
| CL-5 | GTS† | -4.1 | -0.2 | 0.7 | -4.8 | 6.4 | -0.7 | -7.5 | -3.0 | -2.7 | -1.8 | 1.6 | -1.0 |
| | TSI† | 6.0 | -0.1 | -0.1 | -2.8 | 3.2 | 0.1 | 10.9 | -2.4 | -8.0 | -2.4 | 13.1 | 1.5 |
| | CAn† | -3.3 | -0.2 | -1.3 | -4.6 | 0.3 | 0.3 | -3.5 | -5.5 | -4.1 | -1.6 | 0.2 | -0.2 |
| | AIR† | -1.7 | 0.3 | 1.3 | -3.7 | 4.8 | 0.4 | -0.7 | -1.5 | 9.6 | -1.9 | 5.8 | -0.2 |
| | ESAT† | -0.2 | -0.1 | -1.6 | -0.7 | 6.8 | 0.2 | -0.6 | 0.4 | 5.4 | -0.5 | 3.7 | 0.1 |
| | VSR‡ | 16.8 | -0.6 | 0.5 | -4.0 | 0.4 | 0.2 | 10.2 | -12.5 | 18.0 | -10.6 | 10.5 | -1.2 |
| | HM‡ | 7.4 | -2.7 | -0.4 | -6.8 | 0.6 | 0.4 | -1.2 | -1.2 | 6.0 | -4.5 | -0.8 | 0.2 |
| | VisOnly‡ | -0.4 | -0.1 | -1.1 | -4.5 | 0.9 | 0.1 | 0.3 | -0.3 | 0.2 | -0.4 | 2.7 | 0.7 |
| CL-20 | GTS† | -1.4 | 0.1 | 2.5 | -10.5 | 8.6 | -1.0 | -0.5 | -6.4 | -1.4 | -0.8 | 3.9 | -0.7 |
| | TSI† | 5.9 | 0.0 | 8.5 | -4.4 | 10.6 | -0.1 | 6.5 | -11.6 | 2.9 | -3.1 | 13.9 | -0.6 |
| | CAn† | -1.9 | -0.6 | -2.3 | -5.4 | 1.1 | 0.3 | -3.7 | -8.8 | -2.1 | -1.7 | 0.5 | -1.2 |
| | AIR† | 3.7 | 0.3 | 5.3 | -2.7 | 5.9 | -0.5 | 4.8 | -3.5 | 16.3 | -0.3 | 6.0 | -0.3 |
| | ESAT† | 0.7 | 0.4 | -11.5 | -0.5 | 6.6 | 0.2 | -1.2 | -0.1 | -4.6 | -0.0 | 2.9 | -0.1 |
| | VSR‡ | 22.2 | 1.0 | 0.4 | -3.9 | 0.1 | -0.2 | 19.5 | -0.3 | 23.3 | -5.1 | 22.9 | -1.6 |
| | HM‡ | 10.6 | -2.2 | -1.8 | -5.8 | 0.7 | 0.2 | 10.7 | -0.1 | 11.7 | -1.4 | 10.9 | -0.2 |
| | VisOnly‡ | -2.3 | 0.7 | -1.0 | -4.7 | 0.2 | 0.1 | -2.0 | 0.5 | -1.0 | 0.2 | 1.7 | 0.5 |
| CL-50 | GTS† | -0.7 | -0.3 | 3.1 | -11.1 | 9.7 | -1.3 | -1.4 | -6.7 | -3.9 | -2.1 | 6.9 | -0.4 |
| | TSI† | 9.9 | -0.0 | 18.2 | -6.3 | 19.1 | -0.4 | -1.6 | -16.5 | 15.1 | -0.7 | 22.0 | -1.1 |
| | CAn† | -1.8 | -0.7 | -0.4 | -4.4 | 1.3 | -0.5 | -1.8 | -9.8 | -2.1 | -1.1 | 1.0 | -3.4 |
| | AIR† | 4.6 | 0.4 | 7.8 | -3.8 | 8.2 | -0.7 | 6.2 | -3.1 | 17.9 | -0.9 | 8.9 | -0.4 |
| | ESAT† | 1.0 | 0.2 | -14.5 | -3.6 | 7.0 | 0.2 | 1.7 | 0.2 | -9.5 | -0.6 | -0.7 | -0.5 |
| | VSR‡ | 21.9 | 1.0 | 0.4 | -4.5 | 2.3 | -0.3 | 20.2 | -5.3 | 21.0 | 1.1 | 23.4 | -3.6 |
| | HM‡ | 10.2 | -2.1 | 0.7 | -4.5 | 0.3 | 0.2 | 12.5 | -1.5 | 12.3 | -3.7 | 12.2 | 0.2 |
| | VisOnly‡ | -2.4 | 0.6 | -0.2 | -6.8 | 0.3 | -0.1 | -2.0 | 0.7 | 0.2 | 0.2 | 0.3 | 0.1 |

In Figure 3, we assess the computational benefits of LoRSU (CLIP loss) compared to baseline methods. We focus on two key metrics: trainable parameters and TFLOPs. LoRSU requires 25× fewer computation resources than LoRA-F and LoRSU-Ppl, demonstrating the suitability of using CLIP loss when computational resources are limited. Unlike perplexity loss, which requires forward and backward passes through both the vision encoder and LLM, the CLIP loss operates solely on the vision encoder, significantly reducing computational overhead. This makes LoRSU more scalable, enabling efficient CL even in resource-constrained settings.

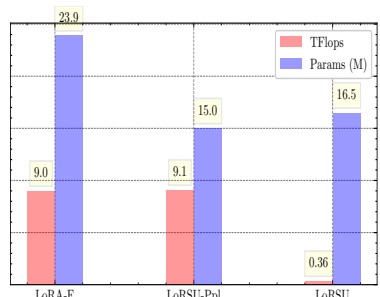

Figure 3: TFlops and trainable parameters comparison.

Table 5: Ablation study on LoRSU (with CLIP loss) hyperparameters in the CL-50 setting. **(Top)** Effect of LoRA rank $r$ on target dataset (TSI) accuracy. **(Middle)** Effect of number of attention heads $k$. **(Bottom)** Comparison of head selection strategies: gradient-based (LoRSU), random (LoRSU-Rand), and all heads (LoRSU-AAH) using GTS with ESAT as control. Results show $r = 64$ and $k = 2$ provide optimal performance, with gradient-based selection outperforming alternatives.

| (a) LoRA Rank Ablation | | | | | | | |
|---|---|---|---|---|---|---|---|
| **FT Dataset** | **rank ($r$)** | **Target** | **DALLE** | **VSR** | **HM** | **MMVP** | **VisOnly** |
| | 8 | 67.2 | 91.1 | 51.5 | 61.6 | 58.0 | 31.5 |
| | 32 | 68.9 | 91.2 | 51.5 | 61.6 | 58.0 | 31.6 |
| **TSI** | **64** | **72.1** | 90.5 | 51.6 | 61.4 | 58.0 | 31.6 |
| | 128 | 65.8 | 90.6 | 51.5 | 62.1 | 56.7 | 31.6 |

| (b) Number of Attention Heads Ablation | | | | | | | |
|---|---|---|---|---|---|---|---|
| **FT Dataset** | **# heads** | **Target** | **DALLE** | **VSR** | **HM** | **MMVP** | **VisOnly** |
| | 0 | 64.2 | 90.8 | 51.5 | 61.8 | 57.3 | 31.5 |
| | 1 | 64.8 | 90.5 | 51.5 | 61.6 | 58.0 | 32.0 |
| **TSI** | **2** | **72.1** | 90.5 | 51.6 | 61.4 | 58.0 | 31.6 |
| | 4 | 66.8 | 90.5 | 51.5 | 62.1 | 58.0 | 31.4 |

| (c) Head Selection Strategy (GTS) | | | |
|---|---|---|---|
| **Method** | **TI ($\uparrow$)** | **CC ($\uparrow$)** | **Params (M)** |
| **LoRSU-Rand** | $7.8 \pm 0.5$ | $-18.1 \pm 0.8$ | 0.36 |
| **LoRSU-AAH** | $9.1 \pm 0.1$ | $-19.6 \pm 0.5$ | 2.88 |
| **LoRSU** | $\mathbf{9.7 \pm 0.1}$ | $\mathbf{-14.3 \pm 0.7}$ | 0.36 |

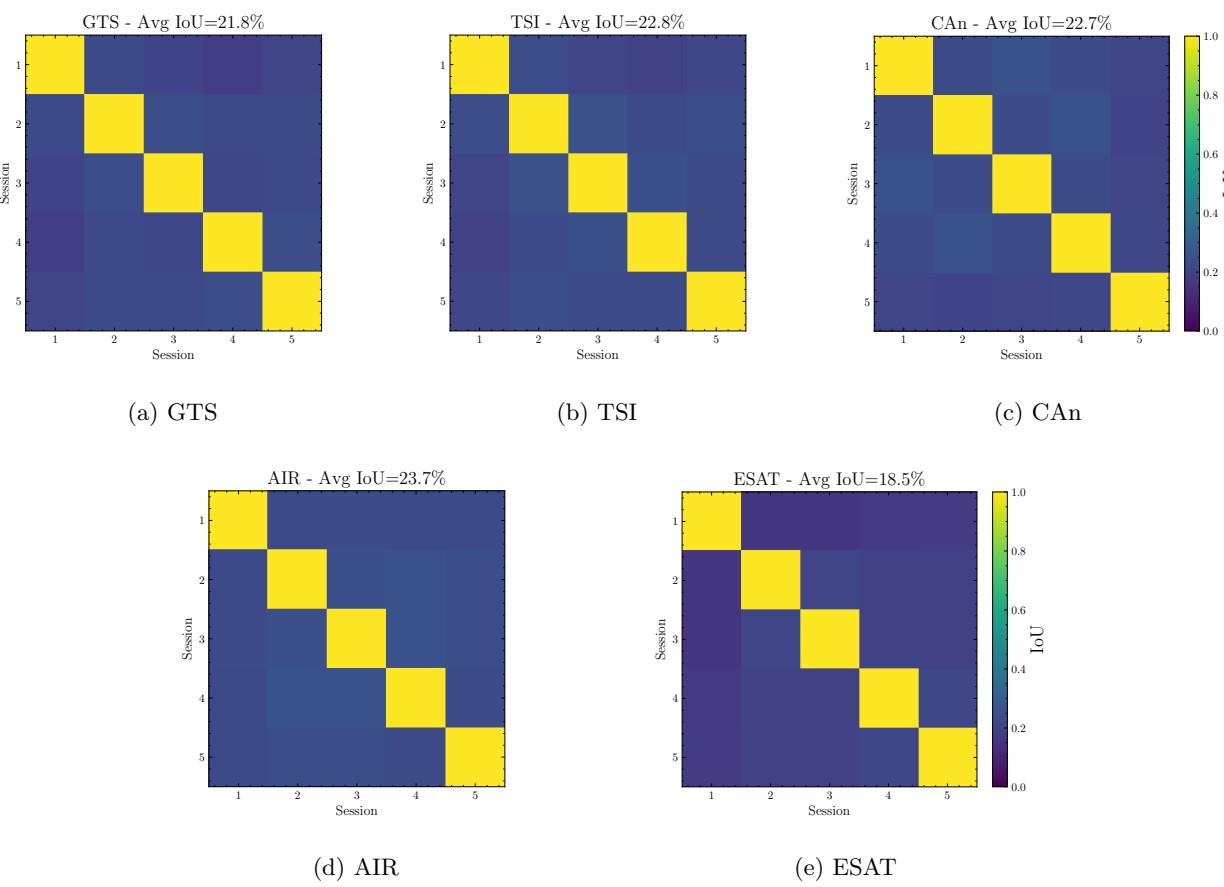

Figure 4: Jaccard similarity matrices of LoRSU's binary masks **p** across CL sessions for five datasets. The heatmaps show the Intersection over Union (IoU) between the binary masks selected in different continual learning sessions. Diagonal elements are 1.0 (perfect overlap), while off-diagonal elements indicate the degree of overlap between different sessions for parameters chosen for update. We include the average IoUs for each dataset in the title of the panel.

### 4.8 Parameter Overlap Analysis

To understand why LoRSU achieves effective plasticity-stability balance, we
analyze the overlap of binary parameter masks selected across the 5. We compute the Jaccard index (Jaccard, 1901), also known as Intersection over Union (IoU), between masks from different sessions for five representative datasets in the CL-50 setting with the CLIP loss. The results are illustrated in Fig. 4 whilst exact values are given in Appendix D.1.

Results show consistent parameter overlap of only 15-25% across all session pairs. This small overlap provides insight into LoRSU's effectiveness. First, sufficient separation (75-85% distinct parameters) provides plasticity for learning new tasks without interfering with previous knowledge, allowing task-specific adaptation. At the same time, moderate overlap (15-25% shared parameters) ensures the model continues to leverage the foundation model's general-purpose visual features rather than fragmenting into isolated task-specific modules. This overlap also prevents catastrophic forgetting: if each task used completely disjoint parameters, the cumulative updates across sessions would shift a large proportion of the pretrained parameters away from their foundation model values, undermining the stability and general capabilities acquired during pretraining.

This consistent parameter overlap emerges naturally from our gradient-based selection: high-magnitude gradients identify parameters that are both critical for the new task and exhibit moderate overlap with previous task parameters, balancing learning and retention.

## 5 Discussion

We introduced LoRSU, a novel parameter-efficient fine-tuning method specifically designed for few-shot continual learning scenarios with VLMs. Unlike existing approaches, LoRSU operates without relying on a replay buffer, making it uniquely suited for resource-constrained settings. Through more than 700 experiments, we demonstrate that LoRSU satisfies all four desiderata: (1) preserving the VLM's generic knowledge, (2) attaining substantial improvements over zero-shot performance, (3) maintaining computational efficiency, and (4) eliminating the need for replay-based methods. LoRSU outperforms 12 baselines in over 80% of evaluations across 10 datasets and 3 settings, achieving the highest TI accuracies in most cases while maintaining stable or even positive CC accuracies. To the best of our knowledge, we are the first to explore few-shot continual learning of VLMs. Whilst we focus on CLIP and LLaVA due to computational constraints, our method is generic to any transformer model, and we plan to extend it to other VLMs and image encoders. Another promising direction is using a smaller LLM proxy model in perplexity-based methods like LoRSU-Ppl, which has shown strong VQA performance. This could improve scalability and LoRSU's use in resource-limited settings. Finally, LoRSU's binary mask-based structured updates ensure efficient, precise parameter updates, but scaling to larger architectures like LLMs poses challenges. Replacing binary masks with more scalable solutions for vast parameter spaces will be crucial to manage memory and processing demands, offering opportunities for further refinement.

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

# A  Proof for the optimal mask $\mathbf{p}^*$

**Definition A.1.** The operator TOP-$C : \mathbb{R}^d \to \mathbb{R}^d$, for $1 \leq C \leq d$ is defined as

$$(\text{TOP-}C(\mathbf{x}))_{\pi(i)} := \begin{cases} x_{\pi(i)}, & i \leq C \\ 0, & \text{otherwise,} \end{cases}$$

where $\mathbf{x} = (x_1, \ldots, x_d)^\top \in \mathbb{R}^d$ and $\pi$ is a permutation of $\{1, 2, \ldots, d\}$ such that $|x_{\pi(i)}| \geq |x_{\pi(i+1)}|$, for $i = 1, \ldots, d-1$, i.e. the TOP-$S$ operator keeps only the $S$ largest elements of $\mathbf{x}$ in magnitude and truncates the rest to zero.

**Lemma A.2.** *For any* $\mathbf{x} \in \mathbb{R}^d - \{\mathbf{0}\}$, $1 \leq C \leq d$, *the optimal mask*

$$\mathbf{p}^* = \arg\max_{\mathbf{p} \in \{0,1\}^d} \frac{\|\mathbf{p} \odot \mathbf{x}\|^2}{\|\mathbf{x}\|^2}, \quad s.t. \ \|\mathbf{p}\|_0 \leq C,$$

*has zeros everywhere except the $C$ largest elements of $\mathbf{x}$ in magnitude.*

*Proof.* Rewriting the optimization problem as

$$\max_{\mathbf{p} \in \{0,1\}^d} \sum_{i=1}^{d} p_i x_i^2, \quad \text{s.t.} \ \sum_{i=1}^{d} p_i \leq C,$$

Notice that this is a trivial binary knapsack problem with maximum weight capacity $C$ and weights equal to one. Hence, the maximum is attained when we pick the top $C$ maximal $x_i^2$ elements. □

*Remark* A.3.

It holds that TOP-$S(\mathbf{x}) = \mathbf{p}^* \odot \mathbf{x}$.

**Corollary A.4.** *The optimal mask* $\mathbf{p}^*$ *in* (5) *has zeros everywhere except for the indices* $i \in \{j : \exists \ell \in \{1, \ldots, G\}, \text{ such that } j \in \{\pi_\ell(1), \ldots, \pi_\ell(c_\ell)\}\}$, *where* $\pi_\ell$ *is the same permutation as in Definition* $A.1$ *for the set of indices* $I_\ell$.

*Proof.* The result follows from the mutual exclusiveness of $I_\ell$ in the constraints of (5) and Lemma A.2. □

# B  Implementation Details

We describe below the implementation details of section 4.

- All the experiments are conducted on a single NVIDIA A100 GPU.

- We have included error bars over three runs for all experiments.

- We use PyTorch Paszke et al. (2019) to implement all the algorithms.

- We use Adam (Kingma, 2014) as an optimizer for the methods that utilize the CLIP loss for fine tuning and AdamW (Loshchilov, 2017) for those ones that use the perplexity loss.

- A learning rate scheduler of Cosine Annealing with Warmup is employed for all methods.

- For all experiments, we set the learning rate $1 \times 10^{-5}$ and $2 \times 10^{-5}$, for LoRSU and LoRSU-Ppl, respectively.

- We set batch size to 16 for all methods that fine-tune the vision encoder through CLIP loss. We reduce the batch size to 8 for those methods that fine-tune the vision encoder through perplexity loss or those that fine-tune the LLM. This was due to GPU memory limitations.

- All methods run for 20, 15, and 10 epochs for the CL-5, CL-10, and CL-50 settings, respectively.

- For LoRA (-Ppl), we set rank $r = 64$ while LoRA-L and LoRA-F use $r = 8$, for all experiments.

- For AdaLoRA, we set the initial rank to 70 and the final average rank to 64.

- The adapters of LoRA and AdaLoRA are applied to all weight matrices of each of the transformer blocks.

- For SPU, we use sparsity=15% for all experiments.

- For LoRSU (-Ppl) we use sparsity=10%, rank=64, and we pick the top-2 attention heads for all experiments.

The choice of the above hyperparameters ensures that LoRA (-Ppl), LoRA-L, LoRA-F, AdaLoRA. SPU, and LoRSU (-Ppl) have similar number of trainable parameters.

## C Datasets

Details on all datasets used in section 4 are presented here.

### C.1 VQA Datasets

We evaluate the performance of LoRSU on ten visual question answering (VQA) datasets falling in two broad categories: regular VQA datasets and classification datasets converted to VQA datasets.

**Regular VQA datasets.** We consider four standard VQA datasets used for benchmarking VLMs' performance Duan et al. (2024): *VSR* Liu et al. (2023), the Visual Spatial Reasoning corpus consists of caption-image pairs labeled as True or False, where each caption describes the spatial relation between two objects in the image. VLMs evaluate whether the caption accurately reflects the image. *HM* Kiela et al. (2020), the Hateful Memes dataset designed to detect multimodal hateful memes. *MMVP* Tong et al. (2024), the Multimodal Visual Patterns dataset is a challenging benchmark which has been built on images that CLIP perceives as similar despite their clear visual differences. *VisOnly* Kamoi et al. (2024), a novel dataset created to directly assess the visual perception abilities of VLMs in answering questions about geometric and numerical details in scientific figures. This dataset allows us to assess fine-grained visual perception in VLMs independently of other abilities, such as reasoning, making it the most challenging among the previously mentioned datasets.

**Classification-to-VQA datasets.** We convert four popular multi-class classification datasets into multiple-choice VQA problems, where each question has five choices, and the VLM is tasked with selecting the correct answer. These datasets are introduced as examples of scenarios where visual domain shifts are encountered, allowing us to examine the utility of updating the image encoder; a critical consideration often overlooked in many standard VQA datasets.The datasets include: *GTS* Stallkamp et al. (2012), the German Traffic Sign dataset, which Zhang et al. (2024) considered as an out-of-distribution dataset for CLIP pretraining; *CAn* Wang et al. (2024b), a recent dataset created to test CLIP's robustness with animal images containing realistic spurious features such as unexpected backgrounds; *AIR* Maji et al. (2013), a fine-grained aircraft classification dataset; *ESAT* Helber et al. (2019), a dataset of satellite images used for land cover classification.

**TSI & DALLE.** In addition to these existing datasets, we introduce two novel VQA datasets: TSI and DALLE, both designed to explore the effects of domain shift. For more details see sections F and C.2.

We follow the common practice in few-shot continual learning Panos et al. (2023) to construct the sequences. We divide each dataset into 5 sets of disjoint classes/categories and consider 5/20/50 shot settings where only 5/20/50 images per class in the current set are used for fine-tuning the model. More details on how we split each of these datasets for the CL settings are provided in appendix C.

### C.2 TSI & DALLE

We start with the description of how we constructed our newly introduced VQA datasets *TSI* and *DALLE*.

Table 6: The original action names of the Toyota Smarthome dataset and their corresponding captions used to create the Toyota Smarthome Images (TSI) dataset. We use ✗ to denore the actions that are ambiguous and were not used to build the TSI dataset. The final prompt is created as "*The person in this image is {caption}*".

| Original Class name/Action | Generated Caption |
|---|---|
| Cook.Cleandishes | washing dishes |
| Cook.Cleanup | cleaning up |
| Cook.Cut | cutting food |
| Cook.Stir | stirring the pot |
| Cook.Usestove | ✗ |
| Cook.Cutbread | cutting bread |
| Drink.Frombottle | holding a bottle |
| Drink.Fromcan | holding a can |
| Drink.Fromcup | holding a cup |
| Drink.Fromglass | holding a glass |
| Eat.Attable | eating |
| Eat.Snack | ✗ |
| Enter | walking |
| Getup | ✗ |
| Laydown | lying down |
| Leave | walking |
| Makecoffee.Pourgrains | using a white coffee machine |
| Makecoffee.Pourwater | using a white coffee machine |
| Maketea.Boilwater | boiling water in a black kettle |
| Maketea.Insertteabag | making tea |
| Pour.Frombottle | holding a bottle |
| Pour.Fromcan | holding a can |
| Pour.Fromkettle | holding a black kettle |
| Readbook | reading a book |
| Sitdown | sitting down |
| Takepills | ✗ |
| Uselaptop | using a laptop |
| Usetablet | using a tablet |
| Usetelephone | using a cordless phone |
| Walk | walking |
| WatchTV | watching TV |

**TSI.** To extract images from the videos of the Toyota Smart Home dataset (TSI), we discretized each video clip into 2 frames per second and then selected the frame in the middle of the total time duration of the video clip. In Table 6 we describe the actions that were selected and the corresponding prompt used for CLIP classification. We also note dropping few actions to avoid ambiguous classes. Note that we did not use any extra data for this CL benchmark and all the images were created from the already available videos of the dataset.

The TSI dataset focuses on elderly individuals performing daily activities in domestic settings, which may not represent the full diversity of how these activities are performed across age groups, abilities, and cultural contexts, and it could potentially reflect or amplify age-related biases.

**DALLE.** We generated images from DALL · E 2 using OpenAI python package and we used the prompt "*A person {a}*" where $a \in \{$ *using a white coffee machine, eating, cutting bread, stirring the pot, holding a glass, watching TV, holding a bottle, walking, making tea, cutting food, holding a cup, using a laptop, lying*

*down, holding a can, person holding a black kettle, reading a book, cleaning up, sitting down, using a tablet, boiling water in a black kettle, using a cordless phone, washing dishes}.*

In Table 7, we present the average number of images per session used to update the model for each CL setting. Finally, Table 8 provides characteristics of the datasets used for evaluating performance.

### C.3 Continual Learning Splits

For the continual learning settings of section 4, we split all datasets into five non-overlapping continual learning (CL) splits based on the classes/categories of each dataset. Unless stated otherwise, we use the training split of each dataset to construct these CL splits.

**GTS Stallkamp et al. (2012).**   We split the 43 classes of GTS as follows:

- *Session 1:* $[25, 2, 11, 1, 40, 27, 5, 9, 17]$.
- *Session 2:* $[32, 29, 20, 39, 21, 15, 23, 10, 3]$.
- *Session 3:* $[18, 38, 42, 14, 22, 35, 34, 19, 33]$.
- *Session 4:* $[12, 26, 41, 0, 37, 6, 13, 24]$.
- *Session 5:* $[30, 28, 31, 7, 16, 4, 36, 8]$.

**TSI Das et al. (2019).**   We split the 27 action categories of TSI as follows:

- *Session 1:* $[WatchTV, Laydown, Sitdown, Pour.Fromkettle, Enter, Drink.Frombottle]$.
- *Session 2:* $[Eat.Attable, Pour.Frombottle, Cook.Cleandishes, Maketea.Boilwater, Leave, Cook.Cleanup]$.
- *Session 3:* $[Maketea.Insertteabag, Makecoffee.Pourwater, Drink.Fromcan, Readbook, Cutbread]$.
- *Session 4:* $[Drink.Fromcup, Drink.Fromglass, Usetablet, Pour.Fromcan, Usetelephone]$.
- *Session 5:* $[Walk, Cook.Stir, Makecoffee.Pourgrains, Cook.Cut, Uselaptop]$.

**CAn Wang et al. (2024b).**   The 45 classes of CAn are split as follows:

- *Session 1:* $[102, 9, 20, 56, 23, 30, 357, 291, 144]$.
- *Session 2:* $[41, 293, 42, 49, 54, 57, 70, 279, 305]$.
- *Session 3:* $[71, 10, 76, 79, 349, 16, 81, 83, 100]$.
- *Session 4:* $[130, 30, 133, 150, 275, 276, 58, 277, 80]$.
- *Session 5:* $[39, 290, 37, 296, 316, 337, 89, 360, 128]$.

The indices of CAn correspond to those of ImageNet Deng et al. (2009) since the dataset was built based on these 45 animal classes of ImageNet.

**AIR Maji et al. (2013).**   We split the 100 aircraft types of AIR as follows:

- *Session 1:* $[23, 8, 11, 7, 48, 13, 1, 91, 94, 54, 16, 63, 52, 41, 80, 2, 47, 87, 78, 66]$.
- *Session 2:* $[19, 6, 24, 10, 59, 30, 22, 29, 83, 37, 93, 81, 43, 99, 86, 28, 34, 88, 44, 14]$.
- *Session 3:* $[84, 70, 4, 20, 15, 21, 31, 76, 57, 67, 73, 50, 69, 25, 98, 46, 96, 0, 72, 35]$.
- *Session 4:* $[58, 92, 3, 95, 56, 90, 26, 40, 55, 89, 75, 71, 60, 42, 9, 82, 39, 18, 77, 68]$.
- *Session 5:* $[32, 79, 12, 85, 36, 17, 64, 27, 74, 45, 61, 38, 51, 62, 65, 33, 5, 53, 97, 49]$.

**ESAT Helber et al. (2019).** We split the 10 different land terrain classes of ESAT as follows:

- *Session 1:* $[0, 1]$.
- *Session 2:* $[2, 3]$.
- *Session 3:* $[4, 5]$.
- *Session 4:* $[6, 7]$.
- *Session 5:* $[8, 9]$.

**DALLE.** This dataset was only used for performance evaluation (control dataset), and not fine-tuning.

**VSR Liu et al. (2023).** The images of this VQA dataset are labeled according to 36 different categories that describe the dominant object of the image. We create the CL splits as follows:

- *Session 1:* [*oven, dining table, spoon, boat, cake, donut, sandwich*].
- *Session 2:* [*fire hydrant, elephant, airplane, truck, apple, hot dog, sheep*].
- *Session 3:* [*kite, baseball glove, cow, tie, scissors, toaster, tv*].
- *Session 4:* [*bicycle, banana, couch, teddy bear, bus, umbrella, bird*].
- *Session 5:* [*potted plant, bowl, broccoli, bottle, knife, orange, person, pizza*].

**HM Kiela et al. (2020).** For the hateful memes dataset, since there was not any labeling information of the images so we can spli the images in a meaningful way, we randomly split the training images into five disjoint sets to create our final CL splits.

**MMVP Tong et al. (2024).** This is the only dataset where no training split is available and it is comprised of just 300 images. For this reason, we only used it for evaluation in our experiments in the main paper. However, for completeness, we included results in Table 27 where we fine-tune on it. We use 150 images for training which are equally split into five sessions and the rest of the 150 images are used for evaluation. Thus, the setting can be considered as a 30-shot CL setting.

**VisOnly Kamoi et al. (2024).** This dataset categorizes its samples into seven categories describing the nature of the geometric and numerical information in scientific figures. We created the splits as follows:

- *Session 1: Geometry-Triangle.*
- *Session 2: Geometry-Quadrilateral.*
- *Session 3: Geometry-Length*
- *Session 4: Geometry-Angle.*
- *Session 5:* [*Geometry-Area, 3D-Size, 3D-Angle*].

# D  Detailed Results

## D.1  Exact IoU results

We provide the exact Jaccard similarity values for the binary masks across sessions for each dataset in tables 9 through 13.

Table 7: Average number of images per session (5 sessions in total) for each dataset used for fine-tuning.

| Setting | FT Dataset | | | | | | | |
|---|---|---|---|---|---|---|---|---|
| | GTS | TSI | CAn | AIR | ESAT | VSR | HM | VisOnly |
| CL-5 | 43.0 | 27.0 | 45.0 | 100.0 | 10.0 | 100.0 | 100.0 | 7.0 |
| CL-20 | 170.0 | 84.0 | 180.0 | 400.0 | 40.0 | 274.6 | 300.0 | 28.0 |
| CL-50 | 430.0 | 253.8 | 450.0 | 1000.0 | 100.0 | 485.2 | 600.0 | 70.0 |

Table 8: Characteristics of the datasets used for performance evaluation in section 4.

| Eval Datasets | GTS | TSI | CAn | AIR | ESAT | DALLE | VSR | HM | MMVP | VisOnly |
|---|---|---|---|---|---|---|---|---|---|---|
| # Samples | 3,990 | 4,908 | 1,796 | 3,333 | 17,000 | 660 | 1,222 | 2,000 | 150 | 1,150 |
| # Classes | 43 | 27 | 45 | 100 | 10 | 27 | 36 | NaN | NaN | 7 |

Table 9: Jaccard similarity matrix for the **GTSRB** dataset.

| Session | 1 | 2 | 3 | 4 | 5 |
|---|---|---|---|---|---|
| 1 | 1.000 | 0.225 | 0.200 | 0.183 | 0.209 |
| 2 | 0.225 | 1.000 | 0.240 | 0.222 | 0.232 |
| 3 | 0.200 | 0.240 | 1.000 | 0.216 | 0.219 |
| 4 | 0.183 | 0.222 | 0.216 | 1.000 | 0.241 |
| 5 | 0.209 | 0.232 | 0.219 | 0.241 | 1.000 |

Table 10: Jaccard similarity matrix for the **TSI** dataset.

| Session | 1 | 2 | 3 | 4 | 5 |
|---|---|---|---|---|---|
| 1 | 1.000 | 0.236 | 0.211 | 0.193 | 0.212 |
| 2 | 0.236 | 1.000 | 0.249 | 0.223 | 0.241 |
| 3 | 0.211 | 0.249 | 1.000 | 0.244 | 0.229 |
| 4 | 0.193 | 0.223 | 0.244 | 1.000 | 0.229 |
| 5 | 0.212 | 0.241 | 0.229 | 0.229 | 1.000 |

Table 11: Jaccard similarity matrix for the **CAn** dataset.

| Session | 1 | 2 | 3 | 4 | 5 |
|---|---|---|---|---|---|
| 1 | 1.000 | 0.233 | 0.250 | 0.227 | 0.210 |
| 2 | 0.233 | 1.000 | 0.227 | 0.252 | 0.201 |
| 3 | 0.250 | 0.227 | 1.000 | 0.234 | 0.210 |
| 4 | 0.227 | 0.252 | 0.234 | 1.000 | 0.212 |
| 5 | 0.210 | 0.201 | 0.210 | 0.212 | 1.000 |

Table 12: Jaccard similarity matrix for the **AIR** dataset.

| Session | 1 | 2 | 3 | 4 | 5 |
|---|---|---|---|---|---|
| **1** | 1.000 | 0.224 | 0.232 | 0.224 | 0.226 |
| **2** | 0.224 | 1.000 | 0.244 | 0.250 | 0.242 |
| **3** | 0.232 | 0.244 | 1.000 | 0.252 | 0.239 |
| **4** | 0.224 | 0.250 | 0.252 | 1.000 | 0.231 |
| **5** | 0.226 | 0.242 | 0.239 | 0.231 | 1.000 |

Table 13: Jaccard similarity matrix for the **ESAT** dataset.

| Session | 1 | 2 | 3 | 4 | 5 |
|---|---|---|---|---|---|
| **1** | 1.000 | 0.157 | 0.153 | 0.170 | 0.173 |
| **2** | 0.157 | 1.000 | 0.208 | 0.198 | 0.192 |
| **3** | 0.153 | 0.208 | 1.000 | 0.195 | 0.194 |
| **4** | 0.170 | 0.198 | 0.195 | 1.000 | 0.211 |
| **5** | 0.173 | 0.192 | 0.194 | 0.211 | 1.000 |

Table 14: Accuracy scores (%) for LLaVA with the pretrained (*Zr-Shot*) or fine-tuned image encoder. All baselines use *GTS* dataset for fine-tuning the image encoder (the LLM remains frozen) via CLIP loss. We include error bars over 3 runs.

| Setting | Method | VQA Datasets (Acc %) | | | | | | | | | |
|---|---|---|---|---|---|---|---|---|---|---|---|
| | | GTS | TSI | CAn | AIR | ESAT | DALLE | VSR | HM | MMVP | VisOnly |
| | **Zr-Shot** | 75.6 | 53.1 | 82.7 | 60.4 | 76.1 | 91.1 | 51.5 | 61.2 | 58.0 | 31.3 |
| **CL-5** | **LN** | $79.1_{\pm1.2}$ | $53.6_{\pm0.5}$ | $81.2_{\pm0.6}$ | $61.0_{\pm1.2}$ | $58.9_{\pm0.9}$ | $91.1_{\pm1.3}$ | $51.9_{\pm1.5}$ | $62.7_{\pm1.1}$ | $59.6_{\pm0.2}$ | $31.8_{\pm0.4}$ |
| | **F-FT** | $79.3_{\pm0.6}$ | $55.1_{\pm0.8}$ | $76.8_{\pm1.3}$ | $58.8_{\pm1.0}$ | $25.6_{\pm0.9}$ | $89.2_{\pm1.2}$ | $51.7_{\pm0.9}$ | $62.1_{\pm0.8}$ | $56.4_{\pm0.4}$ | $30.9_{\pm0.2}$ |
| | **F-EWC** | $80.6_{\pm0.6}$ | $37.4_{\pm1.3}$ | $63.2_{\pm0.7}$ | $55.8_{\pm1.4}$ | $26.1_{\pm1.4}$ | $81.5_{\pm1.1}$ | $51.8_{\pm1.4}$ | $61.2_{\pm0.6}$ | $53.8_{\pm0.4}$ | $31.2_{\pm0.4}$ |
| | **LoRA** | $76.3_{\pm0.8}$ | $52.6_{\pm1.4}$ | $73.3_{\pm0.6}$ | $56.7_{\pm1.2}$ | $49.3_{\pm0.8}$ | $87.1_{\pm1.3}$ | $51.8_{\pm1.2}$ | $61.3_{\pm1.2}$ | $58.1_{\pm0.3}$ | $31.6_{\pm0.4}$ |
| | **AdaLoRA** | $74.7_{\pm0.9}$ | $49.7_{\pm0.7}$ | $79.6_{\pm0.9}$ | $56.3_{\pm0.8}$ | $42.5_{\pm0.8}$ | $91.6_{\pm1.1}$ | $52.0_{\pm0.8}$ | $60.9_{\pm1.2}$ | $57.1_{\pm0.3}$ | $31.7_{\pm0.2}$ |
| | **SPU** | $81.0_{\pm1.4}$ | $53.7_{\pm1.5}$ | $82.5_{\pm0.7}$ | $61.0_{\pm1.0}$ | $67.8_{\pm0.6}$ | $91.6_{\pm1.3}$ | $52.0_{\pm0.6}$ | $62.0_{\pm1.3}$ | $58.2_{\pm0.2}$ | $31.6_{\pm0.2}$ |
| | **LoRSU** | $82.0_{\pm1.3}$ | $53.5_{\pm1.3}$ | $82.4_{\pm0.8}$ | $60.8_{\pm1.4}$ | $66.6_{\pm0.9}$ | $91.5_{\pm1.4}$ | $51.6_{\pm0.7}$ | $61.7_{\pm1.4}$ | $59.8_{\pm0.2}$ | $31.6_{\pm0.2}$ |
| **CL-20** | **LN** | $80.8_{\pm0.6}$ | $49.5_{\pm0.7}$ | $77.7_{\pm1.0}$ | $59.7_{\pm0.5}$ | $32.7_{\pm0.6}$ | $89.8_{\pm0.9}$ | $51.8_{\pm0.7}$ | $62.3_{\pm0.3}$ | $57.5_{\pm0.1}$ | $31.2_{\pm0.2}$ |
| | **F-FT** | $80.2_{\pm0.8}$ | $54.5_{\pm0.7}$ | $74.9_{\pm0.8}$ | $57.2_{\pm1.0}$ | $23.2_{\pm0.7}$ | $86.7_{\pm0.4}$ | $51.9_{\pm0.9}$ | $61.6_{\pm1.0}$ | $58.3_{\pm0.2}$ | $31.7_{\pm0.3}$ |
| | **F-EWC** | $82.3_{\pm0.9}$ | $35.5_{\pm0.9}$ | $55.7_{\pm0.4}$ | $35.4_{\pm0.3}$ | $28.7_{\pm0.9}$ | $72.4_{\pm0.8}$ | $51.6_{\pm0.7}$ | $60.9_{\pm0.8}$ | $53.5_{\pm0.2}$ | $31.0_{\pm0.3}$ |
| | **LoRA** | $78.1_{\pm0.8}$ | $55.6_{\pm0.3}$ | $59.0_{\pm0.9}$ | $47.6_{\pm0.4}$ | $26.0_{\pm0.6}$ | $83.6_{\pm0.8}$ | $52.1_{\pm0.5}$ | $62.1_{\pm1.0}$ | $53.7_{\pm0.3}$ | $30.8_{\pm0.2}$ |
| | **AdaLoRA** | $75.8_{\pm0.8}$ | $51.9_{\pm0.5}$ | $79.3_{\pm0.9}$ | $59.3_{\pm0.4}$ | $62.1_{\pm0.4}$ | $90.7_{\pm1.0}$ | $51.6_{\pm0.5}$ | $61.1_{\pm0.6}$ | $57.7_{\pm0.2}$ | $31.7_{\pm0.2}$ |
| | **SPU** | $83.5_{\pm0.6}$ | $53.1_{\pm0.6}$ | $82.2_{\pm0.7}$ | $60.7_{\pm0.8}$ | $62.0_{\pm0.4}$ | $91.5_{\pm0.4}$ | $51.9_{\pm0.5}$ | $61.8_{\pm0.7}$ | $58.8_{\pm0.2}$ | $31.5_{\pm0.2}$ |
| | **LoRSU** | $84.2_{\pm0.9}$ | $52.9_{\pm0.6}$ | $82.2_{\pm0.5}$ | $60.7_{\pm0.6}$ | $64.7_{\pm0.6}$ | $90.8_{\pm0.5}$ | $51.9_{\pm0.4}$ | $61.7_{\pm0.5}$ | $59.5_{\pm0.1}$ | $31.6_{\pm0.2}$ |
| **CL-50** | **LN** | $80.4_{\pm0.2}$ | $50.4_{\pm0.1}$ | $74.9_{\pm0.1}$ | $58.3_{\pm0.0}$ | $30.4_{\pm0.3}$ | $89.0_{\pm0.1}$ | $51.8_{\pm0.0}$ | $62.0_{\pm0.3}$ | $58.7_{\pm0.1}$ | $31.4_{\pm0.1}$ |
| | **F-FT** | $79.0_{\pm0.1}$ | $48.9_{\pm0.2}$ | $65.0_{\pm0.2}$ | $55.0_{\pm0.3}$ | $23.5_{\pm0.0}$ | $86.8_{\pm0.2}$ | $52.0_{\pm0.1}$ | $60.8_{\pm0.1}$ | $54.9_{\pm0.1}$ | $30.7_{\pm0.1}$ |
| | **F-EWC** | $80.9_{\pm0.2}$ | $45.2_{\pm0.4}$ | $60.5_{\pm0.4}$ | $43.2_{\pm0.0}$ | $26.9_{\pm0.3}$ | $78.5_{\pm0.1}$ | $52.0_{\pm0.0}$ | $58.7_{\pm0.1}$ | $52.9_{\pm0.0}$ | $31.7_{\pm0.1}$ |
| | **LoRA** | $78.7_{\pm0.0}$ | $50.7_{\pm0.0}$ | $62.1_{\pm0.2}$ | $47.4_{\pm0.1}$ | $24.2_{\pm0.2}$ | $82.9_{\pm0.3}$ | $51.7_{\pm0.3}$ | $61.0_{\pm0.2}$ | $54.3_{\pm0.1}$ | $30.8_{\pm0.0}$ |
| | **AdaLoRA** | $76.6_{\pm0.4}$ | $50.4_{\pm0.0}$ | $79.0_{\pm0.2}$ | $57.4_{\pm0.1}$ | $58.3_{\pm0.1}$ | $90.4_{\pm0.2}$ | $51.6_{\pm0.2}$ | $61.8_{\pm0.3}$ | $55.4_{\pm0.1}$ | $31.8_{\pm0.1}$ |
| | **SPU** | $83.3_{\pm0.3}$ | $53.8_{\pm0.2}$ | $81.8_{\pm0.2}$ | $61.1_{\pm0.4}$ | $58.8_{\pm0.0}$ | $91.0_{\pm0.2}$ | $51.8_{\pm0.4}$ | $62.1_{\pm0.1}$ | $59.5_{\pm0.1}$ | $32.2_{\pm0.1}$ |
| | **LoRSU** | $85.3_{\pm0.1}$ | $54.2_{\pm0.1}$ | $81.9_{\pm0.2}$ | $60.5_{\pm0.2}$ | $61.4_{\pm0.3}$ | $91.0_{\pm0.1}$ | $51.7_{\pm0.2}$ | $62.2_{\pm0.4}$ | $58.9_{\pm0.1}$ | $31.8_{\pm0.1}$ |

Table 15: Accuracy scores (%) for LLaVA with the pretrained (*Zr-Shot*) or fine-tuned image encoder. All baselines use *TSI* dataset for fine-tuning the image encoder (the LLM remains frozen) via CLIP loss. We include error bars over 3 runs.

| Setting | Method | VQA Datasets (Acc %) | | | | | | | | | |
|---|---|---|---|---|---|---|---|---|---|---|---|
| | | GTS | TSI | CAn | AIR | ESAT | DALLE | VSR | HM | MMVP | VisOnly |
| | **Zr-Shot** | 75.6 | 53.1 | 82.7 | 60.4 | 76.1 | 91.1 | 51.5 | 61.2 | 58.0 | 31.3 |
| | **LN** | $75.4_{\pm1.0}$ | $53.9_{\pm0.6}$ | $82.6_{\pm1.3}$ | $60.0_{\pm1.0}$ | $75.9_{\pm0.8}$ | $91.1_{\pm1.3}$ | $51.7_{\pm1.4}$ | $61.9_{\pm1.0}$ | $58.4_{\pm0.3}$ | $30.9_{\pm0.3}$ |
| | **F-FT** | $73.8_{\pm0.5}$ | $60.5_{\pm1.1}$ | $81.6_{\pm0.9}$ | $59.5_{\pm1.5}$ | $70.4_{\pm1.0}$ | $91.1_{\pm1.2}$ | $51.8_{\pm0.9}$ | $61.5_{\pm1.3}$ | $56.9_{\pm0.2}$ | $31.3_{\pm0.3}$ |
| | **F-EWC** | $74.9_{\pm1.1}$ | $61.6_{\pm1.0}$ | $82.1_{\pm1.1}$ | $58.8_{\pm0.9}$ | $72.3_{\pm1.2}$ | $89.9_{\pm1.4}$ | $51.9_{\pm0.9}$ | $62.4_{\pm1.4}$ | $55.5_{\pm0.4}$ | $31.5_{\pm0.3}$ |
| **CL-5** | **LoRA** | $73.4_{\pm1.0}$ | $53.0_{\pm0.9}$ | $80.2_{\pm0.6}$ | $58.8_{\pm0.7}$ | $59.1_{\pm1.4}$ | $90.2_{\pm1.1}$ | $51.6_{\pm1.3}$ | $61.2_{\pm1.4}$ | $56.7_{\pm0.4}$ | $31.7_{\pm0.4}$ |
| | **AdaLoRA** | $75.6_{\pm0.8}$ | $54.2_{\pm0.6}$ | $82.6_{\pm1.1}$ | $60.0_{\pm1.3}$ | $75.7_{\pm1.3}$ | $91.1_{\pm1.2}$ | $51.6_{\pm0.9}$ | $62.1_{\pm1.0}$ | $59.5_{\pm0.3}$ | $31.7_{\pm0.2}$ |
| | **SPU** | $75.4_{\pm0.7}$ | $54.0_{\pm1.1}$ | $83.0_{\pm1.3}$ | $60.1_{\pm0.6}$ | $75.7_{\pm1.5}$ | $91.3_{\pm1.3}$ | $51.9_{\pm1.4}$ | $61.7_{\pm0.9}$ | $58.5_{\pm0.4}$ | $31.6_{\pm0.4}$ |
| | **LoRSU** | $75.9_{\pm0.9}$ | $56.3_{\pm0.7}$ | $82.7_{\pm0.9}$ | $60.8_{\pm1.0}$ | $76.2_{\pm1.4}$ | $91.3_{\pm1.2}$ | $51.6_{\pm0.9}$ | $61.7_{\pm0.8}$ | $57.7_{\pm0.3}$ | $31.2_{\pm0.3}$ |
| | **LN** | $72.9_{\pm0.5}$ | $58.2_{\pm0.5}$ | $78.9_{\pm0.9}$ | $56.8_{\pm0.4}$ | $69.3_{\pm0.9}$ | $91.4_{\pm0.8}$ | $51.6_{\pm0.8}$ | $62.6_{\pm0.5}$ | $56.3_{\pm0.3}$ | $31.3_{\pm0.2}$ |
| | **F-FT** | $72.1_{\pm0.7}$ | $68.4_{\pm0.4}$ | $80.0_{\pm0.7}$ | $55.4_{\pm0.4}$ | $58.8_{\pm0.8}$ | $88.4_{\pm0.3}$ | $51.8_{\pm0.6}$ | $62.3_{\pm0.5}$ | $56.9_{\pm0.2}$ | $31.2_{\pm0.3}$ |
| | **F-EWC** | $23.3_{\pm0.6}$ | $69.1_{\pm0.6}$ | $20.4_{\pm0.7}$ | $20.1_{\pm0.6}$ | $24.2_{\pm0.6}$ | $17.7_{\pm0.7}$ | $51.7_{\pm0.7}$ | $56.9_{\pm0.8}$ | $49.6_{\pm0.3}$ | $31.1_{\pm0.1}$ |
| **CL-20** | **LoRA** | $68.5_{\pm0.7}$ | $61.6_{\pm0.3}$ | $76.7_{\pm0.9}$ | $55.3_{\pm0.7}$ | $55.6_{\pm0.6}$ | $88.8_{\pm0.8}$ | $51.9_{\pm0.3}$ | $61.4_{\pm0.6}$ | $59.1_{\pm0.3}$ | $31.1_{\pm0.3}$ |
| | **AdaLoRA** | $70.3_{\pm0.5}$ | $54.4_{\pm0.4}$ | $72.4_{\pm0.5}$ | $43.6_{\pm0.8}$ | $34.6_{\pm0.7}$ | $77.0_{\pm0.3}$ | $52.2_{\pm0.9}$ | $62.6_{\pm0.4}$ | $57.0_{\pm0.1}$ | $31.9_{\pm0.3}$ |
| | **SPU** | $75.5_{\pm0.7}$ | $60.9_{\pm0.8}$ | $82.3_{\pm0.4}$ | $59.2_{\pm0.5}$ | $73.7_{\pm1.0}$ | $91.2_{\pm0.7}$ | $51.7_{\pm0.8}$ | $61.8_{\pm0.9}$ | $58.2_{\pm0.3}$ | $32.0_{\pm0.2}$ |
| | **LoRSU** | $75.9_{\pm0.6}$ | $63.7_{\pm0.4}$ | $82.8_{\pm0.8}$ | $60.4_{\pm0.3}$ | $73.4_{\pm0.6}$ | $90.9_{\pm0.6}$ | $51.7_{\pm0.4}$ | $61.5_{\pm0.7}$ | $58.8_{\pm0.2}$ | $31.9_{\pm0.2}$ |
| | **LN** | $73.0_{\pm0.2}$ | $60.1_{\pm0.2}$ | $79.6_{\pm0.3}$ | $57.7_{\pm0.4}$ | $61.3_{\pm0.4}$ | $89.6_{\pm0.4}$ | $51.9_{\pm0.0}$ | $61.3_{\pm0.0}$ | $55.5_{\pm0.1}$ | $31.3_{\pm0.1}$ |
| | **F-FT** | $72.5_{\pm0.4}$ | $70.3_{\pm0.1}$ | $78.3_{\pm0.4}$ | $53.4_{\pm0.0}$ | $50.6_{\pm0.2}$ | $89.1_{\pm0.3}$ | $52.3_{\pm0.3}$ | $61.1_{\pm0.2}$ | $57.1_{\pm0.1}$ | $31.7_{\pm0.0}$ |
| | **F-EWC** | $48.0_{\pm0.3}$ | $75.5_{\pm0.2}$ | $59.5_{\pm0.4}$ | $38.8_{\pm0.1}$ | $42.6_{\pm0.3}$ | $82.5_{\pm0.0}$ | $52.5_{\pm0.1}$ | $56.4_{\pm0.3}$ | $55.4_{\pm0.1}$ | $31.3_{\pm0.1}$ |
| **CL-50** | **LoRA** | $66.1_{\pm0.2}$ | $71.3_{\pm0.3}$ | $76.0_{\pm0.1}$ | $56.0_{\pm0.1}$ | $44.5_{\pm0.2}$ | $88.9_{\pm0.3}$ | $51.8_{\pm0.1}$ | $60.4_{\pm0.2}$ | $56.3_{\pm0.1}$ | $31.6_{\pm0.1}$ |
| | **AdaLoRA** | $73.1_{\pm0.2}$ | $61.0_{\pm0.0}$ | $80.6_{\pm0.0}$ | $52.0_{\pm0.4}$ | $72.2_{\pm0.3}$ | $88.9_{\pm0.3}$ | $51.7_{\pm0.2}$ | $62.0_{\pm0.4}$ | $59.1_{\pm0.0}$ | $31.2_{\pm0.1}$ |
| | **SPU** | $75.4_{\pm0.0}$ | $65.3_{\pm0.1}$ | $81.8_{\pm0.1}$ | $59.7_{\pm0.2}$ | $72.3_{\pm0.1}$ | $90.8_{\pm0.2}$ | $51.9_{\pm0.1}$ | $61.9_{\pm0.4}$ | $58.0_{\pm0.1}$ | $31.8_{\pm0.0}$ |
| | **LoRSU** | $75.3_{\pm0.2}$ | $72.2_{\pm0.4}$ | $82.4_{\pm0.3}$ | $59.7_{\pm0.3}$ | $72.5_{\pm0.3}$ | $90.8_{\pm0.3}$ | $51.7_{\pm0.2}$ | $61.7_{\pm0.4}$ | $58.5_{\pm0.1}$ | $31.7_{\pm0.0}$ |

Table 16: Accuracy scores (%) for LLaVA with the pretrained (*Zr-Shot*) or fine-tuned image encoder. All baselines use *CAn* dataset for fine-tuning the image encoder (the LLM remains frozen) via CLIP loss. We include error bars over 3 runs.

| Setting | Method | VQA Datasets (Acc %) | | | | | | | | | |
|---|---|---|---|---|---|---|---|---|---|---|---|
| | | GTS | TSI | CAn | AIR | ESAT | DALLE | VSR | HM | MMVP | VisOnly |
| | **Zr-Shot** | 75.6 | 53.1 | 82.7 | 60.4 | 76.1 | 91.1 | 51.5 | 61.2 | 58.0 | 31.3 |
| | **LN** | $74.3_{\pm1.5}$ | $52.9_{\pm1.4}$ | $80.3_{\pm1.4}$ | $58.9_{\pm0.7}$ | $72.4_{\pm1.2}$ | $91.1_{\pm0.8}$ | $52.0_{\pm0.9}$ | $61.5_{\pm1.2}$ | $61.7_{\pm0.3}$ | $32.1_{\pm0.4}$ |
| | **F-FT** | $73.5_{\pm1.1}$ | $50.6_{\pm0.9}$ | $80.3_{\pm0.8}$ | $56.5_{\pm0.6}$ | $63.1_{\pm0.6}$ | $91.3_{\pm1.5}$ | $51.7_{\pm1.4}$ | $61.8_{\pm0.8}$ | $58.4_{\pm0.2}$ | $31.3_{\pm0.4}$ |
| | **F-EWC** | $65.9_{\pm1.5}$ | $39.1_{\pm0.7}$ | $66.0_{\pm1.3}$ | $40.0_{\pm0.9}$ | $41.7_{\pm0.7}$ | $86.2_{\pm0.8}$ | $51.8_{\pm1.3}$ | $59.9_{\pm1.0}$ | $57.6_{\pm0.4}$ | $31.3_{\pm0.2}$ |
| **CL-5** | **LoRA** | $69.7_{\pm1.4}$ | $44.8_{\pm1.1}$ | $81.4_{\pm0.7}$ | $56.9_{\pm1.0}$ | $50.7_{\pm1.3}$ | $92.9_{\pm1.3}$ | $52.0_{\pm1.0}$ | $61.8_{\pm1.5}$ | $56.5_{\pm0.4}$ | $31.3_{\pm0.4}$ |
| | **AdaLoRA** | $75.5_{\pm1.4}$ | $53.2_{\pm0.7}$ | $81.7_{\pm0.6}$ | $60.1_{\pm0.7}$ | $72.0_{\pm1.2}$ | $92.1_{\pm0.9}$ | $51.9_{\pm1.4}$ | $61.8_{\pm1.5}$ | $59.0_{\pm0.3}$ | $31.9_{\pm0.3}$ |
| | **SPU** | $76.0_{\pm0.9}$ | $53.2_{\pm0.6}$ | $82.3_{\pm1.1}$ | $60.3_{\pm1.3}$ | $75.7_{\pm0.9}$ | $91.3_{\pm1.3}$ | $51.7_{\pm0.8}$ | $61.5_{\pm1.2}$ | $58.4_{\pm0.3}$ | $31.4_{\pm0.4}$ |
| | **LoRSU** | $75.2_{\pm0.8}$ | $52.7_{\pm0.9}$ | $83.0_{\pm1.0}$ | $60.1_{\pm0.7}$ | $76.8_{\pm1.0}$ | $91.8_{\pm1.4}$ | $51.6_{\pm1.1}$ | $62.3_{\pm1.2}$ | $58.7_{\pm0.3}$ | $31.4_{\pm0.4}$ |
| | **LN** | $72.9_{\pm0.5}$ | $54.0_{\pm0.9}$ | $80.3_{\pm0.6}$ | $57.3_{\pm0.4}$ | $73.3_{\pm0.4}$ | $90.7_{\pm0.4}$ | $51.8_{\pm0.8}$ | $61.9_{\pm0.9}$ | $61.0_{\pm0.1}$ | $31.4_{\pm0.1}$ |
| | **F-FT** | $72.9_{\pm0.5}$ | $47.9_{\pm0.6}$ | $83.0_{\pm0.7}$ | $56.9_{\pm0.9}$ | $62.7_{\pm0.9}$ | $90.6_{\pm0.9}$ | $51.9_{\pm0.4}$ | $61.3_{\pm0.4}$ | $56.5_{\pm0.2}$ | $31.5_{\pm0.3}$ |
| | **F-EWC** | $70.1_{\pm1.0}$ | $48.7_{\pm0.4}$ | $82.8_{\pm0.5}$ | $51.1_{\pm0.8}$ | $54.8_{\pm0.9}$ | $88.3_{\pm0.7}$ | $51.8_{\pm1.0}$ | $57.0_{\pm0.8}$ | $59.6_{\pm0.3}$ | $31.2_{\pm0.3}$ |
| **CL-20** | **LoRA** | $67.5_{\pm0.6}$ | $48.9_{\pm0.6}$ | $80.4_{\pm0.4}$ | $57.3_{\pm0.9}$ | $39.7_{\pm0.4}$ | $91.1_{\pm0.6}$ | $51.8_{\pm0.9}$ | $61.7_{\pm0.3}$ | $60.1_{\pm0.2}$ | $31.9_{\pm0.3}$ |
| | **AdaLoRA** | $72.5_{\pm1.0}$ | $51.5_{\pm1.0}$ | $79.2_{\pm0.4}$ | $54.1_{\pm1.0}$ | $65.5_{\pm0.7}$ | $90.6_{\pm0.8}$ | $51.7_{\pm0.9}$ | $61.9_{\pm0.9}$ | $56.5_{\pm0.3}$ | $31.7_{\pm0.3}$ |
| | **SPU** | $75.0_{\pm0.5}$ | $53.5_{\pm0.3}$ | $82.8_{\pm0.8}$ | $59.9_{\pm0.6}$ | $76.1_{\pm0.9}$ | $91.6_{\pm0.9}$ | $51.6_{\pm0.6}$ | $61.9_{\pm0.4}$ | $61.8_{\pm0.2}$ | $31.6_{\pm0.3}$ |
| | **LoRSU** | $75.3_{\pm0.8}$ | $53.1_{\pm0.9}$ | $83.8_{\pm0.9}$ | $58.8_{\pm1.0}$ | $75.5_{\pm0.7}$ | $92.0_{\pm0.3}$ | $51.9_{\pm0.4}$ | $62.3_{\pm0.6}$ | $60.4_{\pm0.2}$ | $31.6_{\pm0.2}$ |
| | **LN** | $71.1_{\pm0.1}$ | $50.4_{\pm0.3}$ | $77.0_{\pm0.3}$ | $57.5_{\pm0.3}$ | $57.9_{\pm0.1}$ | $89.7_{\pm0.1}$ | $51.6_{\pm0.1}$ | $62.4_{\pm0.3}$ | $56.1_{\pm0.1}$ | $31.9_{\pm0.0}$ |
| | **F-FT** | $70.1_{\pm0.1}$ | $48.9_{\pm0.3}$ | $81.7_{\pm0.0}$ | $56.2_{\pm0.2}$ | $47.5_{\pm0.1}$ | $89.9_{\pm0.3}$ | $52.0_{\pm0.1}$ | $61.2_{\pm0.1}$ | $57.7_{\pm0.1}$ | $31.1_{\pm0.1}$ |
| | **F-EWC** | $61.7_{\pm0.0}$ | $43.9_{\pm0.3}$ | $83.3_{\pm0.4}$ | $46.2_{\pm0.3}$ | $38.9_{\pm0.2}$ | $87.5_{\pm0.1}$ | $51.8_{\pm0.3}$ | $55.8_{\pm0.3}$ | $54.7_{\pm0.1}$ | $30.7_{\pm0.1}$ |
| **CL-50** | **LoRA** | $66.8_{\pm0.2}$ | $47.8_{\pm0.3}$ | $82.3_{\pm0.2}$ | $55.7_{\pm0.0}$ | $52.0_{\pm0.3}$ | $91.0_{\pm0.3}$ | $51.7_{\pm0.3}$ | $61.6_{\pm0.2}$ | $60.2_{\pm0.0}$ | $31.6_{\pm0.1}$ |
| | **AdaLoRA** | $73.5_{\pm0.0}$ | $49.9_{\pm0.1}$ | $80.9_{\pm0.4}$ | $55.7_{\pm0.4}$ | $77.8_{\pm0.1}$ | $93.1_{\pm0.0}$ | $51.5_{\pm0.1}$ | $61.4_{\pm0.3}$ | $56.9_{\pm0.0}$ | $31.6_{\pm0.1}$ |
| | **SPU** | $75.2_{\pm0.2}$ | $53.2_{\pm0.0}$ | $83.3_{\pm0.3}$ | $59.3_{\pm0.2}$ | $73.1_{\pm0.3}$ | $91.4_{\pm0.4}$ | $51.7_{\pm0.3}$ | $61.7_{\pm0.1}$ | $58.5_{\pm0.1}$ | $31.6_{\pm0.1}$ |
| | **LoRSU** | $75.0_{\pm0.2}$ | $51.8_{\pm0.1}$ | $84.0_{\pm0.4}$ | $58.5_{\pm0.2}$ | $72.7_{\pm0.3}$ | $91.9_{\pm0.3}$ | $51.7_{\pm0.1}$ | $62.3_{\pm0.4}$ | $58.1_{\pm0.0}$ | $31.7_{\pm0.1}$ |

Table 17: Accuracy scores (%) for LLaVA with the pretrained (*Zr-Shot*) or fine-tuned image encoder. All baselines use *AIR* dataset for fine-tuning the image encoder (the LLM remains frozen) via CLIP loss. We include error bars over 3 runs.

| Setting | Method | VQA Datasets (Acc %) | | | | | | | | | |
|---|---|---|---|---|---|---|---|---|---|---|---|
| | | GTS | TSI | CAn | AIR | ESAT | DALLE | VSR | HM | MMVP | VisOnly |
| | **Zr-Shot** | 75.6 | 53.1 | 82.7 | 60.4 | 76.1 | 91.1 | 51.5 | 61.2 | 58.0 | 31.3 |
| **CL-5** | **LN** | $73.4_{\pm0.8}$ | $51.3_{\pm1.2}$ | $80.2_{\pm0.6}$ | $60.7_{\pm1.5}$ | $66.9_{\pm0.7}$ | $91.3_{\pm0.6}$ | $51.9_{\pm0.9}$ | $62.4_{\pm1.2}$ | $58.5_{\pm0.2}$ | $30.6_{\pm0.2}$ |
| | **F-FT** | $72.5_{\pm1.2}$ | $50.5_{\pm0.5}$ | $79.9_{\pm0.9}$ | $62.4_{\pm0.9}$ | $60.7_{\pm1.4}$ | $90.6_{\pm0.5}$ | $51.7_{\pm0.9}$ | $60.9_{\pm1.1}$ | $58.3_{\pm0.4}$ | $31.4_{\pm0.3}$ |
| | **F-EWC** | $74.9_{\pm1.2}$ | $52.4_{\pm0.8}$ | $71.5_{\pm1.2}$ | $63.3_{\pm1.0}$ | $63.8_{\pm1.0}$ | $90.7_{\pm1.5}$ | $51.2_{\pm0.5}$ | $61.2_{\pm0.8}$ | $58.1_{\pm0.4}$ | $31.4_{\pm0.4}$ |
| | **LoRA** | $70.9_{\pm0.9}$ | $52.7_{\pm0.6}$ | $79.0_{\pm0.7}$ | $61.7_{\pm0.5}$ | $48.8_{\pm0.7}$ | $90.6_{\pm0.6}$ | $52.0_{\pm0.9}$ | $62.5_{\pm0.8}$ | $60.0_{\pm0.3}$ | $31.1_{\pm0.2}$ |
| | **AdaLoRA** | $75.0_{\pm1.0}$ | $53.3_{\pm0.8}$ | $83.7_{\pm0.9}$ | $60.8_{\pm0.8}$ | $75.2_{\pm1.5}$ | $91.7_{\pm1.0}$ | $51.6_{\pm0.8}$ | $61.6_{\pm0.8}$ | $56.9_{\pm0.3}$ | $31.9_{\pm0.4}$ |
| | **SPU** | $76.2_{\pm0.6}$ | $53.0_{\pm1.3}$ | $83.0_{\pm0.8}$ | $63.5_{\pm0.8}$ | $75.3_{\pm0.7}$ | $91.5_{\pm1.5}$ | $51.5_{\pm0.6}$ | $61.5_{\pm0.8}$ | $58.1_{\pm0.3}$ | $31.5_{\pm0.4}$ |
| | **LoRSU** | $76.2_{\pm0.8}$ | $53.4_{\pm1.4}$ | $82.5_{\pm1.0}$ | $65.2_{\pm1.3}$ | $76.0_{\pm0.9}$ | $91.8_{\pm0.8}$ | $51.6_{\pm0.8}$ | $62.1_{\pm1.1}$ | $59.0_{\pm0.4}$ | $31.2_{\pm0.3}$ |
| **CL-20** | **LN** | $70.3_{\pm0.9}$ | $53.7_{\pm0.6}$ | $77.9_{\pm1.0}$ | $60.2_{\pm0.4}$ | $56.3_{\pm0.7}$ | $90.6_{\pm0.3}$ | $51.7_{\pm1.0}$ | $62.8_{\pm0.7}$ | $58.1_{\pm0.1}$ | $31.8_{\pm0.3}$ |
| | **F-FT** | $73.0_{\pm0.6}$ | $54.1_{\pm0.6}$ | $80.3_{\pm0.9}$ | $69.7_{\pm0.5}$ | $62.7_{\pm0.5}$ | $90.0_{\pm0.4}$ | $51.9_{\pm0.3}$ | $61.8_{\pm0.4}$ | $58.9_{\pm0.1}$ | $31.4_{\pm0.1}$ |
| | **F-EWC** | $71.2_{\pm0.5}$ | $53.9_{\pm1.0}$ | $79.3_{\pm0.4}$ | $70.6_{\pm1.0}$ | $64.6_{\pm0.7}$ | $89.7_{\pm0.6}$ | $51.7_{\pm0.4}$ | $61.5_{\pm0.5}$ | $58.9_{\pm0.3}$ | $31.4_{\pm0.2}$ |
| | **LoRA** | $71.8_{\pm0.9}$ | $51.1_{\pm0.8}$ | $78.6_{\pm0.3}$ | $65.7_{\pm0.4}$ | $63.4_{\pm0.8}$ | $89.9_{\pm1.0}$ | $51.7_{\pm0.3}$ | $62.3_{\pm0.3}$ | $56.2_{\pm0.2}$ | $31.5_{\pm0.2}$ |
| | **AdaLoRA** | $73.4_{\pm0.8}$ | $51.6_{\pm0.6}$ | $81.2_{\pm0.9}$ | $63.1_{\pm0.6}$ | $73.8_{\pm0.8}$ | $90.8_{\pm0.5}$ | $52.1_{\pm0.4}$ | $62.7_{\pm0.8}$ | $57.7_{\pm0.2}$ | $31.2_{\pm0.1}$ |
| | **SPU** | $75.7_{\pm0.4}$ | $52.2_{\pm0.7}$ | $82.0_{\pm0.8}$ | $63.4_{\pm0.9}$ | $72.6_{\pm0.6}$ | $91.7_{\pm0.6}$ | $51.8_{\pm0.6}$ | $62.2_{\pm0.5}$ | $59.0_{\pm0.2}$ | $31.4_{\pm0.2}$ |
| | **LoRSU** | $75.7_{\pm0.9}$ | $52.6_{\pm0.9}$ | $81.4_{\pm0.7}$ | $66.3_{\pm0.7}$ | $73.0_{\pm0.8}$ | $90.9_{\pm0.8}$ | $51.9_{\pm0.8}$ | $61.8_{\pm0.8}$ | $56.9_{\pm0.1}$ | $31.6_{\pm0.3}$ |
| **CL-50** | **LN** | $69.6_{\pm0.4}$ | $54.0_{\pm0.1}$ | $76.9_{\pm0.3}$ | $62.2_{\pm0.2}$ | $50.9_{\pm0.2}$ | $90.2_{\pm0.0}$ | $52.0_{\pm0.3}$ | $62.8_{\pm0.4}$ | $57.7_{\pm0.1}$ | $31.5_{\pm0.1}$ |
| | **F-FT** | $71.2_{\pm0.3}$ | $50.3_{\pm0.3}$ | $78.3_{\pm0.2}$ | $70.4_{\pm0.4}$ | $59.9_{\pm0.0}$ | $90.1_{\pm0.1}$ | $51.9_{\pm0.1}$ | $61.8_{\pm0.3}$ | $57.5_{\pm0.1}$ | $31.3_{\pm0.1}$ |
| | **F-EWC** | $71.8_{\pm0.2}$ | $51.6_{\pm0.1}$ | $78.3_{\pm0.0}$ | $71.3_{\pm0.2}$ | $57.6_{\pm0.2}$ | $90.2_{\pm0.2}$ | $51.7_{\pm0.1}$ | $61.1_{\pm0.2}$ | $57.4_{\pm0.1}$ | $31.5_{\pm0.0}$ |
| | **LoRA** | $69.8_{\pm0.0}$ | $54.7_{\pm0.0}$ | $77.0_{\pm0.3}$ | $68.2_{\pm0.3}$ | $51.6_{\pm0.1}$ | $90.0_{\pm0.1}$ | $52.0_{\pm0.4}$ | $62.4_{\pm0.0}$ | $57.1_{\pm0.1}$ | $31.5_{\pm0.1}$ |
| | **AdaLoRA** | $74.2_{\pm0.3}$ | $52.0_{\pm0.2}$ | $82.4_{\pm0.1}$ | $65.0_{\pm0.2}$ | $72.6_{\pm0.0}$ | $91.9_{\pm0.1}$ | $51.7_{\pm0.2}$ | $60.7_{\pm0.1}$ | $55.6_{\pm0.0}$ | $31.3_{\pm0.0}$ |
| | **SPU** | $75.2_{\pm0.2}$ | $52.2_{\pm0.4}$ | $82.6_{\pm0.3}$ | $66.6_{\pm0.4}$ | $70.0_{\pm0.2}$ | $91.6_{\pm0.3}$ | $51.9_{\pm0.2}$ | $62.0_{\pm0.3}$ | $57.6_{\pm0.0}$ | $31.8_{\pm0.0}$ |
| | **LoRSU** | $75.4_{\pm0.4}$ | $52.7_{\pm0.3}$ | $81.6_{\pm0.2}$ | $68.6_{\pm0.3}$ | $69.7_{\pm0.3}$ | $91.5_{\pm0.2}$ | $51.7_{\pm0.4}$ | $62.2_{\pm0.1}$ | $58.7_{\pm0.1}$ | $31.1_{\pm0.1}$ |

Table 18: Accuracy scores (%) for LLaVA with the pretrained (*Zr-Shot*) or fine-tuned image encoder. All baselines use *ESAT* dataset for fine-tuning the image encoder (the LLM remains frozen) via CLIP loss. We include error bars over 3 runs.

| Setting | Method | VQA Datasets (Acc %) | | | | | | | | | |
|---|---|---|---|---|---|---|---|---|---|---|---|
| | | GTS | TSI | CAn | AIR | ESAT | DALLE | VSR | HM | MMVP | VisOnly |
| | **Zr-Shot** | 75.6 | 53.1 | 82.7 | 60.4 | 76.1 | 91.1 | 51.5 | 61.2 | 58.0 | 31.3 |
| **CL-5** | **LN** | $75.8_{\pm0.9}$ | $53.2_{\pm0.6}$ | $82.6_{\pm1.1}$ | $60.0_{\pm1.3}$ | $80.3_{\pm1.0}$ | $92.7_{\pm1.0}$ | $51.9_{\pm0.7}$ | $61.7_{\pm0.5}$ | $60.4_{\pm0.4}$ | $31.8_{\pm0.3}$ |
| | **F-FT** | $69.1_{\pm0.8}$ | $50.5_{\pm0.6}$ | $80.8_{\pm1.1}$ | $57.7_{\pm1.5}$ | $65.8_{\pm0.6}$ | $91.3_{\pm1.5}$ | $51.8_{\pm0.7}$ | $62.0_{\pm0.7}$ | $58.8_{\pm0.3}$ | $30.4_{\pm0.2}$ |
| | **F-EWC** | $66.3_{\pm0.9}$ | $52.1_{\pm1.4}$ | $79.3_{\pm1.0}$ | $56.8_{\pm1.3}$ | $67.7_{\pm1.3}$ | $90.9_{\pm0.8}$ | $51.9_{\pm1.3}$ | $62.0_{\pm1.2}$ | $55.4_{\pm0.2}$ | $30.9_{\pm0.4}$ |
| | **LoRA** | $73.2_{\pm1.3}$ | $49.3_{\pm1.2}$ | $80.6_{\pm0.9}$ | $60.4_{\pm1.1}$ | $74.5_{\pm0.8}$ | $92.3_{\pm1.3}$ | $52.0_{\pm1.1}$ | $61.6_{\pm1.1}$ | $57.4_{\pm0.4}$ | $31.4_{\pm0.3}$ |
| | **AdaLoRA** | $75.9_{\pm0.5}$ | $52.4_{\pm1.4}$ | $82.4_{\pm0.5}$ | $60.5_{\pm0.8}$ | $78.0_{\pm1.3}$ | $91.5_{\pm0.9}$ | $51.6_{\pm0.8}$ | $61.5_{\pm1.3}$ | $59.0_{\pm0.4}$ | $30.9_{\pm0.2}$ |
| | **SPU** | $75.8_{\pm0.8}$ | $53.2_{\pm1.4}$ | $82.8_{\pm1.4}$ | $60.5_{\pm1.5}$ | $80.6_{\pm0.9}$ | $91.5_{\pm1.1}$ | $51.7_{\pm0.6}$ | $61.7_{\pm1.5}$ | $57.5_{\pm0.4}$ | $31.5_{\pm0.2}$ |
| | **LoRSU** | $76.2_{\pm1.0}$ | $53.6_{\pm1.1}$ | $82.5_{\pm1.2}$ | $60.8_{\pm0.8}$ | $82.9_{\pm1.0}$ | $91.5_{\pm0.9}$ | $51.6_{\pm0.9}$ | $61.3_{\pm0.7}$ | $57.7_{\pm0.4}$ | $31.9_{\pm0.4}$ |
| **CL-20** | **LN** | $74.5_{\pm0.5}$ | $52.6_{\pm0.7}$ | $82.5_{\pm0.5}$ | $58.8_{\pm0.7}$ | $77.0_{\pm0.4}$ | $92.4_{\pm0.5}$ | $51.9_{\pm1.0}$ | $62.5_{\pm0.5}$ | $58.0_{\pm0.3}$ | $31.2_{\pm0.1}$ |
| | **F-FT** | $66.5_{\pm0.8}$ | $51.1_{\pm0.7}$ | $79.1_{\pm0.4}$ | $56.7_{\pm0.6}$ | $51.2_{\pm0.7}$ | $92.0_{\pm0.4}$ | $51.6_{\pm0.6}$ | $61.4_{\pm0.8}$ | $60.1_{\pm0.1}$ | $31.5_{\pm0.2}$ |
| | **F-EWC** | $69.3_{\pm0.3}$ | $51.2_{\pm1.0}$ | $60.5_{\pm0.8}$ | $57.1_{\pm0.6}$ | $54.1_{\pm0.6}$ | $89.7_{\pm0.6}$ | $51.9_{\pm0.6}$ | $60.9_{\pm0.7}$ | $58.4_{\pm0.2}$ | $31.8_{\pm0.2}$ |
| | **LoRA** | $71.1_{\pm0.7}$ | $50.9_{\pm0.5}$ | $80.3_{\pm1.0}$ | $59.4_{\pm0.7}$ | $64.6_{\pm0.7}$ | $91.1_{\pm0.7}$ | $52.0_{\pm0.4}$ | $62.3_{\pm0.6}$ | $62.3_{\pm0.2}$ | $31.3_{\pm0.1}$ |
| | **AdaLoRA** | $70.0_{\pm0.6}$ | $47.3_{\pm0.8}$ | $78.4_{\pm0.9}$ | $51.7_{\pm0.4}$ | $69.3_{\pm0.5}$ | $91.3_{\pm0.7}$ | $51.7_{\pm0.9}$ | $60.8_{\pm0.9}$ | $58.1_{\pm0.2}$ | $31.6_{\pm0.1}$ |
| | **SPU** | $75.6_{\pm0.9}$ | $53.1_{\pm0.3}$ | $82.8_{\pm0.9}$ | $59.9_{\pm0.8}$ | $81.5_{\pm0.6}$ | $92.3_{\pm0.4}$ | $51.9_{\pm0.5}$ | $61.5_{\pm0.8}$ | $58.8_{\pm0.2}$ | $31.7_{\pm0.1}$ |
| | **LoRSU** | $75.3_{\pm1.0}$ | $53.7_{\pm0.8}$ | $82.8_{\pm0.4}$ | $60.7_{\pm0.8}$ | $82.7_{\pm0.7}$ | $91.6_{\pm0.6}$ | $51.6_{\pm0.4}$ | $61.5_{\pm0.4}$ | $58.4_{\pm0.2}$ | $31.4_{\pm0.2}$ |
| **CL-50** | **LN** | $73.1_{\pm0.3}$ | $53.0_{\pm0.2}$ | $82.0_{\pm0.1}$ | $59.1_{\pm0.2}$ | $80.7_{\pm0.0}$ | $92.4_{\pm0.2}$ | $51.8_{\pm0.3}$ | $62.0_{\pm0.1}$ | $60.4_{\pm0.0}$ | $32.0_{\pm0.0}$ |
| | **F-FT** | $58.0_{\pm0.4}$ | $50.3_{\pm0.0}$ | $76.8_{\pm0.1}$ | $57.2_{\pm0.2}$ | $34.7_{\pm0.1}$ | $89.7_{\pm0.0}$ | $51.7_{\pm0.2}$ | $61.6_{\pm0.2}$ | $58.1_{\pm0.0}$ | $31.6_{\pm0.1}$ |
| | **F-EWC** | $59.0_{\pm0.1}$ | $64.5_{\pm0.1}$ | $77.2_{\pm0.1}$ | $56.3_{\pm0.1}$ | $38.0_{\pm0.2}$ | $87.3_{\pm0.2}$ | $51.9_{\pm0.2}$ | $60.7_{\pm0.2}$ | $58.2_{\pm0.1}$ | $31.8_{\pm0.0}$ |
| | **LoRA** | $62.8_{\pm0.3}$ | $47.2_{\pm0.4}$ | $72.4_{\pm0.4}$ | $54.4_{\pm0.2}$ | $61.6_{\pm0.4}$ | $90.2_{\pm0.3}$ | $51.7_{\pm0.2}$ | $62.0_{\pm0.1}$ | $60.8_{\pm0.0}$ | $30.9_{\pm0.1}$ |
| | **AdaLoRA** | $67.2_{\pm0.2}$ | $49.3_{\pm0.3}$ | $78.8_{\pm0.3}$ | $56.9_{\pm0.3}$ | $58.8_{\pm0.3}$ | $89.6_{\pm0.3}$ | $51.8_{\pm0.1}$ | $61.9_{\pm0.2}$ | $56.0_{\pm0.1}$ | $31.6_{\pm0.0}$ |
| | **SPU** | $75.1_{\pm0.3}$ | $53.4_{\pm0.2}$ | $82.5_{\pm0.2}$ | $60.2_{\pm0.3}$ | $81.9_{\pm0.1}$ | $92.3_{\pm0.3}$ | $51.8_{\pm0.1}$ | $61.6_{\pm0.1}$ | $57.1_{\pm0.1}$ | $31.9_{\pm0.0}$ |
| | **LoRSU** | $75.4_{\pm0.3}$ | $53.9_{\pm0.1}$ | $83.1_{\pm0.2}$ | $60.3_{\pm0.1}$ | $83.1_{\pm0.1}$ | $92.1_{\pm0.1}$ | $51.6_{\pm0.2}$ | $61.2_{\pm0.0}$ | $57.6_{\pm0.0}$ | $31.1_{\pm0.0}$ |

Table 19: *Average accuracy* (ACC) and *backward transfer* (BWT) scores (%) for LLaVA with the fine-tuned CLIP-L-14. Each column indicates the setting and fine-tuning method. We include error bars over 3 runs.

| | | Zr-Shot | | LoRA | | SPU | | LoRSU | |
|---|---|---|---|---|---|---|---|---|---|
| Setting | FTD | ACC (↑) | BWT (↑) | ACC (↑) | BWT (↑) | ACC (↑) | BWT (↑) | ACC (↑) | BWT (↑) |
| **CL-5** | **GTS** | 75.4 | 0.0 | $79.2_{\pm0.7}$ | $-7.1_{\pm0.8}$ | $80.8_{\pm0.5}$ | $0.5_{\pm0.6}$ | $81.1_{\pm0.6}$ | $0.4_{\pm0.7}$ |
| | **TSI** | 54.0 | 0.0 | $55.5_{\pm0.9}$ | $-2.5_{\pm0.6}$ | $55.5_{\pm0.6}$ | $0.2_{\pm0.5}$ | $57.0_{\pm0.8}$ | $0.5_{\pm0.6}$ |
| | **AIR** | 60.4 | 0.0 | $59.2_{\pm0.8}$ | $-2.1_{\pm0.7}$ | $64.7_{\pm0.5}$ | $2.8_{\pm0.6}$ | $65.0_{\pm0.7}$ | $2.5_{\pm0.6}$ |
| | **ESAT** | 76.4 | 0.0 | $73.8_{\pm0.9}$ | $-3.4_{\pm0.6}$ | $79.8_{\pm0.6}$ | $1.5_{\pm0.7}$ | $82.2_{\pm0.7}$ | $2.0_{\pm0.6}$ |
| **CL-20** | **GTS** | 75.4 | 0.0 | $77.2_{\pm0.4}$ | $-9.1_{\pm0.5}$ | $82.8_{\pm0.4}$ | $-0.6_{\pm0.3}$ | $83.5_{\pm0.6}$ | $-0.4_{\pm0.3}$ |
| | **TSI** | 54.0 | 0.0 | $60.6_{\pm0.3}$ | $-7.2_{\pm0.4}$ | $60.1_{\pm0.5}$ | $-1.7_{\pm0.3}$ | $62.1_{\pm0.3}$ | $-0.9_{\pm0.4}$ |
| | **AIR** | 60.4 | 0.0 | $64.3_{\pm0.4}$ | $-3.6_{\pm0.6}$ | $65.2_{\pm0.7}$ | $1.1_{\pm0.4}$ | $65.4_{\pm0.3}$ | $0.9_{\pm0.4}$ |
| | **ESAT** | 76.4 | 0.0 | $64.1_{\pm0.5}$ | $-18.3_{\pm0.7}$ | $82.0_{\pm0.4}$ | $2.0_{\pm0.2}$ | $82.7_{\pm0.5}$ | $0.1_{\pm0.3}$ |
| **CL-50** | **GTS** | 75.4 | 0.0 | $79.3_{\pm0.3}$ | $-10.3_{\pm0.5}$ | $83.8_{\pm0.2}$ | $-0.7_{\pm0.1}$ | $84.7_{\pm0.3}$ | $-0.5_{\pm0.2}$ |
| | **TSI** | 54.0 | 0.0 | $67.0_{\pm0.3}$ | $-8.1_{\pm0.6}$ | $61.8_{\pm0.2}$ | $-1.9_{\pm0.3}$ | $67.9_{\pm0.2}$ | $-1.1_{\pm0.3}$ |
| | **AIR** | 60.4 | 0.0 | $65.6_{\pm0.4}$ | $-6.1_{\pm0.3}$ | $67.1_{\pm0.3}$ | $0.5_{\pm0.2}$ | $67.7_{\pm0.3}$ | $0.7_{\pm0.3}$ |
| | **ESAT** | 76.4 | 0.0 | $61.4_{\pm0.3}$ | $-27.8_{\pm0.4}$ | $81.2_{\pm0.3}$ | $-2.4_{\pm0.2}$ | $82.1_{\pm0.4}$ | $-0.8_{\pm0.2}$ |

## D.2   CLIP-based Updates+

The detailed accuracies for all baselines and datasets used to create Table 3 of the main paper can be found in Tables 14 through 18.

## D.3   Extra ACC and BWT results

In Table 19 we present results of the ACC and BWT on extra datasets plus the ones in the main paper. The results follow the same patterns as in section 4 with LoRSU demonstrating the most consistent performance in both ACC and BWT compared to the other two baselines. SPU is close to LoRSU in terms of BWT but it significantly lacks behind in ACC.

## D.4   CLIP-based vs. Perplexity-based Updates+

The detailed accuracies for all baselines and datasets used to create Table 4 of the main paper can be found in Tables 20 through 24. We have also included results on fine-tuning the model using *MMVP* dataset in Table 27.

# E   Detailed Ablation Studies

## E.1   Ablation on the rank $r$ of LoRSU

In Table 29, we investigate the effect on performance of using different ranks for LoRSU. As the rank $r$ increases, the VQA accuracy on the target dataset slightly improves, peaking at $r = 64$. Beyond that, performance slightly decreases. Performance on other datasets remains relatively stable with small fluctuations.

## E.2   Ablation on the number of optimal attention heads of LoRSU

In Table 30, we examine how the number of attention heads chosen to be fine-tuned affects LoRSU's performance. We notice that more attention heads marginally improve the performance of the model while the extra flexibility can cause more forgetting, e.g. ESAT.

Table 20: Exact accuracy scores (%) for each baseline used to fine-tune the model on the *GTS* dataset under three different continual learning (5, 10, 50 shots) settings. We include error bars over 3 runs.

| Setting | Method | VQA Datasets (Acc %) | | | | | | | | | |
|---|---|---|---|---|---|---|---|---|---|---|---|
| | | GTS | TSI | CAn | AIR | ESAT | DALLE | VSR | HM | MMVP | VisOnly |
| | Zr-Shot | 75.6 | 53.1 | 82.7 | 60.4 | 76.1 | 91.1 | 51.5 | 61.2 | 58.0 | 31.3 |
| CL-5 | LoRA-L | $71.5_{\pm1.2}$ | $52.3_{\pm0.5}$ | $81.2_{\pm0.6}$ | $60.0_{\pm1.2}$ | $75.5_{\pm0.9}$ | $91.5_{\pm1.3}$ | $51.9_{\pm1.5}$ | $61.2_{\pm1.1}$ | $57.6_{\pm0.3}$ | $32.2_{\pm0.5}$ |
| | LoRA | $76.3_{\pm0.8}$ | $52.6_{\pm1.4}$ | $73.3_{\pm0.6}$ | $56.7_{\pm1.2}$ | $49.3_{\pm0.8}$ | $87.1_{\pm1.3}$ | $51.8_{\pm1.2}$ | $61.3_{\pm1.2}$ | $58.1_{\pm0.3}$ | $31.6_{\pm0.4}$ |
| | LoRSU | $82.0_{\pm1.3}$ | $53.5_{\pm1.3}$ | $82.4_{\pm0.8}$ | $60.8_{\pm1.4}$ | $66.6_{\pm0.9}$ | $91.5_{\pm1.4}$ | $51.6_{\pm0.7}$ | $61.7_{\pm1.4}$ | $59.8_{\pm0.2}$ | $31.6_{\pm0.2}$ |
| | LoRA-Ppl | $68.1_{\pm0.8}$ | $54.5_{\pm1.4}$ | $80.7_{\pm0.6}$ | $59.3_{\pm1.2}$ | $52.8_{\pm0.8}$ | $90.7_{\pm1.3}$ | $51.7_{\pm1.2}$ | $60.7_{\pm1.2}$ | $54.8_{\pm0.4}$ | $33.4_{\pm0.5}$ |
| | LoRA-F | $72.9_{\pm0.9}$ | $54.0_{\pm0.7}$ | $81.5_{\pm0.9}$ | $59.6_{\pm0.8}$ | $61.9_{\pm0.8}$ | $90.3_{\pm1.1}$ | $51.9_{\pm0.8}$ | $60.9_{\pm1.2}$ | $58.4_{\pm0.4}$ | $31.1_{\pm0.3}$ |
| | LoRSU-Ppl | $77.2_{\pm1.4}$ | $55.1_{\pm1.5}$ | $82.1_{\pm0.7}$ | $58.9_{\pm1.0}$ | $67.0_{\pm0.6}$ | $90.9_{\pm1.3}$ | $51.8_{\pm0.6}$ | $61.6_{\pm1.3}$ | $58.7_{\pm0.3}$ | $30.4_{\pm0.3}$ |
| CL-20 | LoRA-L | $74.2_{\pm0.9}$ | $52.2_{\pm0.9}$ | $82.1_{\pm0.5}$ | $59.6_{\pm1.0}$ | $75.9_{\pm0.6}$ | $91.8_{\pm1.0}$ | $51.6_{\pm0.4}$ | $62.1_{\pm0.9}$ | $59.1_{\pm0.2}$ | $31.8_{\pm0.2}$ |
| | LoRA | $78.1_{\pm0.8}$ | $55.6_{\pm0.3}$ | $59.0_{\pm0.9}$ | $47.6_{\pm0.4}$ | $26.0_{\pm0.6}$ | $83.6_{\pm0.8}$ | $52.1_{\pm0.5}$ | $62.1_{\pm1.0}$ | $53.7_{\pm0.3}$ | $30.8_{\pm0.2}$ |
| | LoRSU | $84.2_{\pm0.9}$ | $52.9_{\pm0.6}$ | $82.2_{\pm0.5}$ | $60.7_{\pm0.6}$ | $64.7_{\pm0.6}$ | $90.8_{\pm0.5}$ | $51.9_{\pm0.4}$ | $61.7_{\pm0.5}$ | $59.5_{\pm0.1}$ | $31.6_{\pm0.2}$ |
| | LoRA-Ppl | $75.1_{\pm0.9}$ | $50.4_{\pm0.9}$ | $75.8_{\pm0.4}$ | $56.5_{\pm0.3}$ | $40.1_{\pm0.9}$ | $89.7_{\pm0.8}$ | $51.6_{\pm0.7}$ | $57.8_{\pm0.8}$ | $54.2_{\pm0.2}$ | $31.5_{\pm0.4}$ |
| | LoRA-F | $74.2_{\pm0.8}$ | $52.7_{\pm0.3}$ | $80.1_{\pm0.9}$ | $59.5_{\pm0.4}$ | $66.0_{\pm0.6}$ | $90.1_{\pm0.8}$ | $52.1_{\pm0.5}$ | $64.7_{\pm1.0}$ | $60.4_{\pm0.4}$ | $32.3_{\pm0.2}$ |
| | LoRSU-Ppl | $79.5_{\pm0.8}$ | $56.1_{\pm0.5}$ | $82.1_{\pm0.9}$ | $59.8_{\pm0.4}$ | $66.1_{\pm0.4}$ | $90.8_{\pm1.0}$ | $51.7_{\pm0.5}$ | $62.1_{\pm0.6}$ | $59.0_{\pm0.3}$ | $31.5_{\pm0.3}$ |
| CL-50 | LoRA-L | $74.9_{\pm0.2}$ | $51.7_{\pm0.2}$ | $81.8_{\pm0.2}$ | $59.8_{\pm0.3}$ | $75.8_{\pm0.1}$ | $91.5_{\pm0.0}$ | $52.0_{\pm0.1}$ | $61.1_{\pm0.2}$ | $57.4_{\pm0.1}$ | $31.8_{\pm0.1}$ |
| | LoRA | $78.7_{\pm0.0}$ | $50.7_{\pm0.0}$ | $62.1_{\pm0.2}$ | $47.4_{\pm0.1}$ | $24.2_{\pm0.2}$ | $82.9_{\pm0.3}$ | $51.7_{\pm0.3}$ | $61.0_{\pm0.2}$ | $54.3_{\pm0.1}$ | $30.8_{\pm0.0}$ |
| | LoRSU | $85.3_{\pm0.1}$ | $54.2_{\pm0.1}$ | $81.9_{\pm0.2}$ | $60.5_{\pm0.2}$ | $61.4_{\pm0.3}$ | $91.0_{\pm0.1}$ | $51.7_{\pm0.2}$ | $62.2_{\pm0.4}$ | $58.9_{\pm0.1}$ | $31.8_{\pm0.1}$ |
| | LoRA-Ppl | $74.2_{\pm0.1}$ | $49.4_{\pm0.2}$ | $76.0_{\pm0.2}$ | $57.9_{\pm0.3}$ | $37.2_{\pm0.0}$ | $89.5_{\pm0.2}$ | $51.7_{\pm0.1}$ | $57.7_{\pm0.1}$ | $55.6_{\pm0.1}$ | $29.8_{\pm0.1}$ |
| | LoRA-F | $71.7_{\pm0.2}$ | $51.7_{\pm0.4}$ | $80.8_{\pm0.4}$ | $58.3_{\pm0.0}$ | $60.9_{\pm0.3}$ | $90.8_{\pm0.1}$ | $52.1_{\pm0.0}$ | $63.3_{\pm0.1}$ | $57.5_{\pm0.0}$ | $30.9_{\pm0.1}$ |
| | LoRSU-Ppl | $82.5_{\pm0.0}$ | $55.8_{\pm0.0}$ | $82.1_{\pm0.2}$ | $59.9_{\pm0.1}$ | $65.4_{\pm0.2}$ | $91.0_{\pm0.3}$ | $51.6_{\pm0.3}$ | $61.7_{\pm0.2}$ | $62.3_{\pm0.1}$ | $32.2_{\pm0.0}$ |

Table 21: Exact accuracy scores (%) for each baseline used to fine-tune the model on the *TSI* dataset under three different continual learning (5, 10, 50 shots) settings. We include error bars over 3 runs.

| Setting | Method | VQA Datasets (Acc %) | | | | | | | | | |
|---|---|---|---|---|---|---|---|---|---|---|---|
| | | GTS | TSI | CAn | AIR | ESAT | DALLE | VSR | HM | MMVP | VisOnly |
| | Zr-Shot | 75.6 | 53.1 | 82.7 | 60.4 | 76.1 | 91.1 | 51.5 | 61.2 | 58.0 | 31.3 |
| CL-5 | LoRA-L | $76.0_{\pm1.5}$ | $59.1_{\pm0.6}$ | $82.7_{\pm0.9}$ | $60.7_{\pm0.7}$ | $75.9_{\pm0.9}$ | $91.5_{\pm1.0}$ | $51.5_{\pm0.9}$ | $63.6_{\pm1.2}$ | $54.1_{\pm0.4}$ | $31.2_{\pm0.4}$ |
| | LoRA | $73.4_{\pm1.0}$ | $53.0_{\pm0.9}$ | $80.2_{\pm0.6}$ | $58.8_{\pm0.7}$ | $59.1_{\pm1.4}$ | $90.2_{\pm1.1}$ | $51.6_{\pm1.3}$ | $61.2_{\pm1.4}$ | $56.7_{\pm0.4}$ | $31.7_{\pm0.4}$ |
| | LoRSU | $75.9_{\pm0.9}$ | $56.3_{\pm0.7}$ | $82.7_{\pm0.9}$ | $60.8_{\pm1.0}$ | $76.2_{\pm1.4}$ | $91.3_{\pm1.2}$ | $51.6_{\pm0.9}$ | $61.7_{\pm0.8}$ | $57.7_{\pm0.3}$ | $31.2_{\pm0.3}$ |
| | LoRA-Ppl | $75.0_{\pm1.0}$ | $64.0_{\pm0.6}$ | $82.8_{\pm1.3}$ | $58.4_{\pm1.0}$ | $60.8_{\pm0.8}$ | $88.7_{\pm1.3}$ | $51.6_{\pm1.4}$ | $61.5_{\pm1.0}$ | $55.0_{\pm0.4}$ | $32.2_{\pm0.4}$ |
| | LoRA-F | $75.3_{\pm0.5}$ | $45.1_{\pm1.1}$ | $82.5_{\pm0.9}$ | $57.2_{\pm1.5}$ | $73.2_{\pm1.0}$ | $83.9_{\pm1.2}$ | $53.8_{\pm0.9}$ | $64.3_{\pm1.3}$ | $45.6_{\pm0.3}$ | $30.9_{\pm0.4}$ |
| | LoRSU-Ppl | $76.1_{\pm1.1}$ | $66.2_{\pm1.0}$ | $83.9_{\pm1.1}$ | $66.1_{\pm0.9}$ | $76.1_{\pm1.2}$ | $91.1_{\pm1.4}$ | $52.0_{\pm0.9}$ | $64.4_{\pm1.4}$ | $60.8_{\pm0.5}$ | $31.1_{\pm0.4}$ |
| CL-20 | LoRA-L | $76.1_{\pm0.7}$ | $59.0_{\pm0.6}$ | $82.4_{\pm0.4}$ | $60.8_{\pm0.4}$ | $75.7_{\pm0.9}$ | $91.3_{\pm0.7}$ | $51.5_{\pm0.9}$ | $63.9_{\pm1.0}$ | $55.4_{\pm0.3}$ | $30.8_{\pm0.3}$ |
| | LoRA | $68.5_{\pm0.7}$ | $61.6_{\pm0.3}$ | $76.7_{\pm0.9}$ | $55.3_{\pm0.7}$ | $55.6_{\pm0.6}$ | $88.8_{\pm0.8}$ | $51.9_{\pm0.3}$ | $61.4_{\pm0.6}$ | $59.1_{\pm0.3}$ | $31.1_{\pm0.3}$ |
| | LoRSU | $75.9_{\pm0.6}$ | $63.7_{\pm0.4}$ | $82.8_{\pm0.8}$ | $60.4_{\pm0.3}$ | $73.4_{\pm0.6}$ | $90.9_{\pm0.6}$ | $51.7_{\pm0.4}$ | $61.5_{\pm0.7}$ | $58.8_{\pm0.2}$ | $31.9_{\pm0.2}$ |
| | LoRA-Ppl | $62.1_{\pm0.6}$ | $59.6_{\pm0.5}$ | $71.9_{\pm0.6}$ | $48.3_{\pm0.7}$ | $42.5_{\pm1.0}$ | $75.8_{\pm0.8}$ | $51.6_{\pm0.6}$ | $49.0_{\pm0.5}$ | $49.7_{\pm0.3}$ | $32.4_{\pm0.2}$ |
| | LoRA-F | $76.1_{\pm0.5}$ | $56.0_{\pm0.5}$ | $82.8_{\pm0.9}$ | $58.2_{\pm0.4}$ | $67.7_{\pm0.9}$ | $87.5_{\pm0.8}$ | $51.6_{\pm0.8}$ | $64.4_{\pm0.5}$ | $40.3_{\pm0.4}$ | $31.2_{\pm0.2}$ |
| | LoRSU-Ppl | $76.4_{\pm0.7}$ | $67.0_{\pm0.4}$ | $83.0_{\pm0.7}$ | $57.4_{\pm0.4}$ | $74.0_{\pm0.8}$ | $88.1_{\pm0.3}$ | $51.8_{\pm0.6}$ | $63.6_{\pm0.5}$ | $57.6_{\pm0.2}$ | $30.8_{\pm0.3}$ |
| CL-50 | LoRA-L | $76.4_{\pm0.2}$ | $63.0_{\pm0.2}$ | $81.9_{\pm0.2}$ | $60.5_{\pm0.2}$ | $75.6_{\pm0.2}$ | $91.1_{\pm0.2}$ | $51.7_{\pm0.2}$ | $64.1_{\pm0.3}$ | $55.6_{\pm0.2}$ | $30.9_{\pm0.0}$ |
| | LoRA | $66.1_{\pm0.2}$ | $71.3_{\pm0.3}$ | $76.0_{\pm0.1}$ | $56.0_{\pm0.1}$ | $44.5_{\pm0.2}$ | $88.9_{\pm0.3}$ | $51.8_{\pm0.1}$ | $60.4_{\pm0.2}$ | $56.3_{\pm0.1}$ | $31.6_{\pm0.1}$ |
| | LoRSU | $75.3_{\pm0.2}$ | $72.2_{\pm0.4}$ | $82.4_{\pm0.3}$ | $59.7_{\pm0.3}$ | $72.5_{\pm0.3}$ | $90.8_{\pm0.3}$ | $51.7_{\pm0.2}$ | $61.7_{\pm0.4}$ | $58.5_{\pm0.1}$ | $31.7_{\pm0.0}$ |
| | LoRA-Ppl | $46.3_{\pm0.3}$ | $51.5_{\pm0.3}$ | $63.4_{\pm0.1}$ | $40.1_{\pm0.1}$ | $41.3_{\pm0.4}$ | $73.9_{\pm0.2}$ | $51.7_{\pm0.3}$ | $49.5_{\pm0.3}$ | $40.2_{\pm0.1}$ | $32.7_{\pm0.1}$ |
| | LoRA-F | $74.0_{\pm0.2}$ | $68.2_{\pm0.1}$ | $81.6_{\pm0.3}$ | $59.2_{\pm0.0}$ | $75.1_{\pm0.2}$ | $88.5_{\pm0.2}$ | $56.8_{\pm0.1}$ | $65.0_{\pm0.3}$ | $50.8_{\pm0.1}$ | $30.4_{\pm0.1}$ |
| | LoRSU-Ppl | $75.8_{\pm0.2}$ | $75.1_{\pm0.2}$ | $82.1_{\pm0.3}$ | $56.0_{\pm0.4}$ | $74.2_{\pm0.4}$ | $86.0_{\pm0.4}$ | $52.0_{\pm0.0}$ | $63.2_{\pm0.0}$ | $58.1_{\pm0.1}$ | $30.2_{\pm0.1}$ |

Table 22: Exact accuracy scores (%) for each baseline used to fine-tune the model on the *CAn* dataset under three different continual learning (5, 10, 50 shots) settings. We include error bars over 3 runs.

| Setting | Method | VQA Datasets (Acc %) | | | | | | | | | |
|---|---|---|---|---|---|---|---|---|---|---|---|
| | | GTS | TSI | CAn | AIR | ESAT | DALLE | VSR | HM | MMVP | VisOnly |
| | Zr-Shot | 75.6 | 53.1 | 82.7 | 60.4 | 76.1 | 91.1 | 51.5 | 61.2 | 58.0 | 31.3 |
| CL-5 | LoRA-L | $75.5_{\pm1.4}$ | $53.1_{\pm0.8}$ | $79.4_{\pm1.4}$ | $59.2_{\pm0.6}$ | $75.2_{\pm0.9}$ | $91.5_{\pm1.1}$ | $52.4_{\pm1.3}$ | $60.2_{\pm1.1}$ | $57.7_{\pm0.5}$ | $32.1_{\pm0.3}$ |
| | LoRA | $69.7_{\pm1.4}$ | $44.8_{\pm1.1}$ | $81.4_{\pm0.7}$ | $56.9_{\pm1.0}$ | $50.7_{\pm1.3}$ | $92.9_{\pm1.3}$ | $52.0_{\pm1.0}$ | $61.8_{\pm1.5}$ | $56.5_{\pm0.4}$ | $31.3_{\pm0.4}$ |
| | LoRSU | $75.2_{\pm0.8}$ | $52.7_{\pm0.9}$ | $83.0_{\pm1.0}$ | $60.1_{\pm0.7}$ | $76.8_{\pm1.0}$ | $91.8_{\pm1.4}$ | $51.6_{\pm1.1}$ | $62.3_{\pm1.2}$ | $58.7_{\pm0.3}$ | $31.4_{\pm0.4}$ |
| | LoRA-Ppl | $65.8_{\pm1.1}$ | $50.7_{\pm0.6}$ | $79.2_{\pm0.5}$ | $48.4_{\pm1.4}$ | $63.0_{\pm1.2}$ | $86.7_{\pm1.3}$ | $51.8_{\pm1.0}$ | $57.2_{\pm1.4}$ | $52.5_{\pm0.3}$ | $32.4_{\pm0.4}$ |
| | LoRA-F | $70.1_{\pm0.6}$ | $52.2_{\pm0.7}$ | $78.6_{\pm0.7}$ | $50.9_{\pm0.9}$ | $73.4_{\pm0.8}$ | $91.3_{\pm1.0}$ | $54.7_{\pm0.8}$ | $62.2_{\pm1.4}$ | $58.0_{\pm0.5}$ | $31.3_{\pm0.3}$ |
| | LoRSU-Ppl | $74.6_{\pm0.9}$ | $51.3_{\pm1.4}$ | $82.9_{\pm1.2}$ | $58.4_{\pm1.2}$ | $77.7_{\pm1.2}$ | $91.8_{\pm1.3}$ | $51.5_{\pm1.1}$ | $64.7_{\pm1.4}$ | $56.5_{\pm0.6}$ | $29.8_{\pm0.3}$ |
| CL-20 | LoRA-L | $73.6_{\pm1.0}$ | $52.2_{\pm0.9}$ | $80.8_{\pm0.9}$ | $56.7_{\pm0.4}$ | $74.7_{\pm0.8}$ | $91.7_{\pm0.5}$ | $52.2_{\pm0.6}$ | $60.9_{\pm0.8}$ | $59.1_{\pm0.3}$ | $31.9_{\pm0.4}$ |
| | LoRA | $67.5_{\pm0.6}$ | $48.9_{\pm0.6}$ | $80.4_{\pm0.4}$ | $57.3_{\pm0.9}$ | $39.7_{\pm0.4}$ | $91.1_{\pm0.6}$ | $51.8_{\pm0.9}$ | $61.7_{\pm0.3}$ | $60.1_{\pm0.2}$ | $31.9_{\pm0.3}$ |
| | LoRSU | $75.3_{\pm0.8}$ | $53.1_{\pm0.9}$ | $83.8_{\pm0.9}$ | $58.8_{\pm1.0}$ | $75.5_{\pm0.7}$ | $92.0_{\pm0.3}$ | $51.9_{\pm0.4}$ | $62.3_{\pm0.6}$ | $60.4_{\pm0.2}$ | $31.6_{\pm0.2}$ |
| | LoRA-Ppl | $65.6_{\pm0.9}$ | $47.0_{\pm0.7}$ | $79.0_{\pm0.4}$ | $46.0_{\pm0.6}$ | $58.9_{\pm0.8}$ | $82.5_{\pm0.8}$ | $51.9_{\pm0.7}$ | $43.9_{\pm1.0}$ | $52.5_{\pm0.4}$ | $30.4_{\pm0.3}$ |
| | LoRA-F | $69.4_{\pm0.9}$ | $54.9_{\pm0.4}$ | $80.6_{\pm0.4}$ | $50.4_{\pm0.5}$ | $72.0_{\pm0.8}$ | $91.2_{\pm0.5}$ | $51.9_{\pm0.9}$ | $64.3_{\pm1.0}$ | $57.0_{\pm0.3}$ | $31.6_{\pm0.3}$ |
| | LoRSU-Ppl | $72.4_{\pm0.6}$ | $49.2_{\pm0.4}$ | $83.2_{\pm0.7}$ | $56.4_{\pm0.9}$ | $75.5_{\pm0.6}$ | $91.8_{\pm0.9}$ | $51.6_{\pm0.5}$ | $61.0_{\pm0.8}$ | $57.7_{\pm0.3}$ | $31.6_{\pm0.3}$ |
| CL-50 | LoRA-L | $73.8_{\pm0.1}$ | $51.6_{\pm0.2}$ | $80.9_{\pm0.2}$ | $56.9_{\pm0.1}$ | $74.9_{\pm0.2}$ | $91.3_{\pm0.3}$ | $51.7_{\pm0.2}$ | $61.2_{\pm0.3}$ | $58.0_{\pm0.1}$ | $32.4_{\pm0.1}$ |
| | LoRA | $66.8_{\pm0.2}$ | $47.8_{\pm0.3}$ | $82.3_{\pm0.2}$ | $55.7_{\pm0.0}$ | $52.0_{\pm0.3}$ | $91.0_{\pm0.3}$ | $51.7_{\pm0.3}$ | $61.6_{\pm0.2}$ | $60.2_{\pm0.0}$ | $31.6_{\pm0.1}$ |
| | LoRSU | $75.0_{\pm0.2}$ | $51.8_{\pm0.1}$ | $84.0_{\pm0.4}$ | $58.5_{\pm0.2}$ | $72.7_{\pm0.3}$ | $91.9_{\pm0.3}$ | $51.7_{\pm0.1}$ | $62.3_{\pm0.4}$ | $58.1_{\pm0.0}$ | $31.7_{\pm0.1}$ |
| | LoRA-Ppl | $56.2_{\pm0.4}$ | $36.4_{\pm0.0}$ | $80.9_{\pm0.1}$ | $48.5_{\pm0.3}$ | $54.1_{\pm0.3}$ | $78.1_{\pm0.2}$ | $53.6_{\pm0.4}$ | $62.3_{\pm0.3}$ | $48.4_{\pm0.1}$ | $32.4_{\pm0.1}$ |
| | LoRA-F | $69.2_{\pm0.2}$ | $52.0_{\pm0.2}$ | $80.6_{\pm0.1}$ | $53.7_{\pm0.3}$ | $74.4_{\pm0.1}$ | $90.7_{\pm0.2}$ | $51.8_{\pm0.4}$ | $66.5_{\pm0.0}$ | $58.7_{\pm0.1}$ | $31.4_{\pm0.1}$ |
| | LoRSU-Ppl | $74.9_{\pm0.4}$ | $49.7_{\pm0.4}$ | $83.7_{\pm0.0}$ | $42.5_{\pm0.4}$ | $74.9_{\pm0.2}$ | $91.2_{\pm0.3}$ | $51.2_{\pm0.3}$ | $52.2_{\pm0.4}$ | $58.5_{\pm0.2}$ | $32.3_{\pm0.2}$ |

Table 23: Exact accuracy scores (%) for each baseline used to fine-tune the model on the *AIR* dataset under three different continual learning (5, 10, 50 shots) settings. We include error bars over 3 runs.

| Setting | Method | VQA Datasets (Acc %) | | | | | | | | | |
|---|---|---|---|---|---|---|---|---|---|---|---|
| | | GTS | TSI | CAn | AIR | ESAT | DALLE | VSR | HM | MMVP | VisOnly |
| | Zr-Shot | 75.6 | 53.1 | 82.7 | 60.4 | 76.1 | 91.1 | 51.5 | 61.2 | 58.0 | 31.3 |
| CL-5 | LoRA-L | $75.6_{\pm0.7}$ | $54.4_{\pm0.5}$ | $81.8_{\pm1.1}$ | $58.7_{\pm0.9}$ | $75.7_{\pm1.4}$ | $92.0_{\pm1.4}$ | $51.6_{\pm0.9}$ | $61.0_{\pm0.6}$ | $59.1_{\pm0.3}$ | $32.2_{\pm0.5}$ |
| | LoRA | $70.9_{\pm0.9}$ | $52.7_{\pm0.6}$ | $79.0_{\pm0.7}$ | $61.7_{\pm0.5}$ | $48.8_{\pm0.7}$ | $90.6_{\pm0.6}$ | $52.0_{\pm0.9}$ | $62.5_{\pm0.8}$ | $60.0_{\pm0.3}$ | $31.1_{\pm0.2}$ |
| | LoRSU | $76.2_{\pm0.8}$ | $53.4_{\pm1.4}$ | $82.5_{\pm1.0}$ | $65.2_{\pm1.3}$ | $76.0_{\pm0.9}$ | $91.8_{\pm0.8}$ | $51.6_{\pm0.8}$ | $62.1_{\pm1.1}$ | $59.0_{\pm0.4}$ | $31.2_{\pm0.3}$ |
| | LoRA-Ppl | $74.9_{\pm0.8}$ | $54.2_{\pm1.2}$ | $79.1_{\pm0.5}$ | $59.7_{\pm0.9}$ | $68.5_{\pm0.9}$ | $90.8_{\pm1.3}$ | $51.8_{\pm0.7}$ | $62.0_{\pm0.7}$ | $55.1_{\pm0.5}$ | $31.1_{\pm0.5}$ |
| | LoRA-F | $72.3_{\pm0.5}$ | $50.6_{\pm1.3}$ | $78.7_{\pm1.4}$ | $70.0_{\pm1.3}$ | $64.4_{\pm0.9}$ | $90.9_{\pm0.6}$ | $54.9_{\pm1.3}$ | $57.7_{\pm1.1}$ | $62.0_{\pm0.6}$ | $32.2_{\pm0.5}$ |
| | LoRSU-Ppl | $75.6_{\pm1.0}$ | $54.6_{\pm1.2}$ | $79.8_{\pm1.0}$ | $66.2_{\pm0.5}$ | $76.4_{\pm1.1}$ | $90.6_{\pm1.3}$ | $51.7_{\pm1.3}$ | $60.1_{\pm0.9}$ | $58.8_{\pm0.4}$ | $31.1_{\pm0.4}$ |
| CL-20 | LoRA-L | $75.4_{\pm0.3}$ | $53.6_{\pm0.4}$ | $82.2_{\pm1.0}$ | $64.1_{\pm1.0}$ | $75.7_{\pm0.5}$ | $92.2_{\pm0.3}$ | $51.5_{\pm0.5}$ | $61.5_{\pm0.8}$ | $58.9_{\pm0.2}$ | $31.9_{\pm0.3}$ |
| | LoRA | $71.8_{\pm0.9}$ | $51.1_{\pm0.8}$ | $78.6_{\pm0.3}$ | $65.7_{\pm0.4}$ | $63.4_{\pm0.8}$ | $89.9_{\pm1.0}$ | $51.7_{\pm0.3}$ | $62.3_{\pm0.3}$ | $56.2_{\pm0.2}$ | $31.5_{\pm0.2}$ |
| | LoRSU | $75.7_{\pm0.9}$ | $52.6_{\pm0.9}$ | $81.4_{\pm0.7}$ | $66.3_{\pm0.7}$ | $73.0_{\pm0.8}$ | $90.9_{\pm0.8}$ | $51.9_{\pm0.8}$ | $61.8_{\pm0.8}$ | $56.9_{\pm0.1}$ | $31.6_{\pm0.3}$ |
| | LoRA-Ppl | $72.1_{\pm0.5}$ | $48.0_{\pm0.8}$ | $72.7_{\pm0.4}$ | $65.2_{\pm1.0}$ | $65.1_{\pm0.5}$ | $90.4_{\pm0.3}$ | $51.8_{\pm0.6}$ | $61.5_{\pm0.8}$ | $55.8_{\pm0.1}$ | $31.7_{\pm0.1}$ |
| | LoRA-F | $74.5_{\pm0.8}$ | $53.0_{\pm0.3}$ | $82.0_{\pm0.6}$ | $76.7_{\pm0.6}$ | $74.9_{\pm0.9}$ | $91.1_{\pm0.3}$ | $52.4_{\pm0.6}$ | $59.3_{\pm0.8}$ | $59.6_{\pm0.4}$ | $31.3_{\pm0.3}$ |
| | LoRSU-Ppl | $76.1_{\pm0.8}$ | $55.5_{\pm0.5}$ | $78.7_{\pm0.8}$ | $66.4_{\pm0.6}$ | $75.7_{\pm0.6}$ | $91.6_{\pm1.0}$ | $51.5_{\pm0.3}$ | $59.8_{\pm0.5}$ | $58.1_{\pm0.4}$ | $31.2_{\pm0.4}$ |
| CL-50 | LoRA-L | $75.6_{\pm0.2}$ | $53.8_{\pm0.1}$ | $83.5_{\pm0.1}$ | $65.0_{\pm0.0}$ | $75.7_{\pm0.1}$ | $92.0_{\pm0.0}$ | $51.8_{\pm0.2}$ | $61.1_{\pm0.1}$ | $58.7_{\pm0.1}$ | $32.3_{\pm0.0}$ |
| | LoRA | $69.8_{\pm0.0}$ | $54.7_{\pm0.0}$ | $77.0_{\pm0.3}$ | $68.2_{\pm0.3}$ | $51.6_{\pm0.1}$ | $90.0_{\pm0.1}$ | $52.0_{\pm0.4}$ | $62.4_{\pm0.0}$ | $57.1_{\pm0.1}$ | $31.5_{\pm0.1}$ |
| | LoRSU | $75.4_{\pm0.4}$ | $52.7_{\pm0.3}$ | $81.6_{\pm0.2}$ | $68.6_{\pm0.3}$ | $69.7_{\pm0.3}$ | $91.5_{\pm0.2}$ | $51.7_{\pm0.4}$ | $62.2_{\pm0.1}$ | $58.7_{\pm0.1}$ | $31.1_{\pm0.1}$ |
| | LoRA-Ppl | $74.4_{\pm0.1}$ | $50.9_{\pm0.4}$ | $76.8_{\pm0.2}$ | $66.6_{\pm0.3}$ | $65.4_{\pm0.2}$ | $91.3_{\pm0.1}$ | $51.6_{\pm0.1}$ | $57.2_{\pm0.2}$ | $53.7_{\pm0.1}$ | $31.5_{\pm0.1}$ |
| | LoRA-F | $74.6_{\pm0.3}$ | $53.2_{\pm0.2}$ | $80.7_{\pm0.4}$ | $78.3_{\pm0.1}$ | $71.4_{\pm0.2}$ | $91.4_{\pm0.0}$ | $52.9_{\pm0.4}$ | $60.0_{\pm0.2}$ | $57.4_{\pm0.0}$ | $31.1_{\pm0.2}$ |
| | LoRSU-Ppl | $75.1_{\pm0.2}$ | $54.5_{\pm0.1}$ | $78.0_{\pm0.4}$ | $69.3_{\pm0.1}$ | $75.7_{\pm0.1}$ | $91.5_{\pm0.1}$ | $51.7_{\pm0.0}$ | $61.5_{\pm0.1}$ | $58.2_{\pm0.0}$ | $30.8_{\pm0.0}$ |

Table 24: Exact accuracy scores (%) for each baseline used to fine-tune the model on the *ESAT* dataset under three different continual learning (5, 10, 50 shots) settings. We include error bars over 3 runs.

| Setting | Method | VQA Datasets (Acc %) | | | | | | | | | |
| --- | --- | --- | --- | --- | --- | --- | --- | --- | --- | --- | --- |
| | | GTS | TSI | CAn | AIR | ESAT | DALLE | VSR | HM | MMVP | VisOnly |
| | Zr-Shot | 75.6 | 53.1 | 82.7 | 60.4 | 76.1 | 91.1 | 51.5 | 61.2 | 58.0 | 31.3 |
| CL-5 | LoRA-L | $75.4_{\pm0.7}$ | $52.2_{\pm1.4}$ | $82.8_{\pm0.6}$ | $60.6_{\pm1.5}$ | $75.9_{\pm1.1}$ | $91.7_{\pm0.9}$ | $51.5_{\pm0.6}$ | $60.2_{\pm0.8}$ | $57.6_{\pm0.5}$ | $31.6_{\pm0.4}$ |
| | LoRA | $73.2_{\pm1.3}$ | $49.3_{\pm1.2}$ | $80.6_{\pm0.9}$ | $60.4_{\pm1.1}$ | $74.5_{\pm0.8}$ | $92.3_{\pm1.3}$ | $52.0_{\pm1.1}$ | $61.6_{\pm1.1}$ | $57.4_{\pm0.4}$ | $31.4_{\pm0.3}$ |
| | LoRSU | $76.2_{\pm1.0}$ | $53.6_{\pm1.1}$ | $82.5_{\pm1.2}$ | $60.8_{\pm0.8}$ | $82.9_{\pm1.0}$ | $91.5_{\pm0.9}$ | $51.6_{\pm0.9}$ | $61.3_{\pm0.7}$ | $57.7_{\pm0.4}$ | $31.9_{\pm0.4}$ |
| | LoRA-Ppl | $76.0_{\pm0.7}$ | $52.6_{\pm1.0}$ | $82.6_{\pm1.3}$ | $60.4_{\pm1.4}$ | $75.5_{\pm0.9}$ | $91.9_{\pm1.0}$ | $51.8_{\pm0.9}$ | $62.8_{\pm0.8}$ | $59.0_{\pm0.4}$ | $31.6_{\pm0.5}$ |
| | LoRA-F | $74.3_{\pm1.3}$ | $51.5_{\pm1.4}$ | $81.1_{\pm1.0}$ | $60.3_{\pm1.1}$ | $81.5_{\pm1.2}$ | $90.8_{\pm1.2}$ | $51.9_{\pm1.2}$ | $61.9_{\pm1.2}$ | $57.7_{\pm0.2}$ | $31.3_{\pm0.5}$ |
| | LoRSU-Ppl | $75.6_{\pm1.4}$ | $52.3_{\pm0.6}$ | $82.0_{\pm1.2}$ | $60.5_{\pm1.0}$ | $79.8_{\pm1.1}$ | $92.3_{\pm0.5}$ | $51.8_{\pm1.2}$ | $62.2_{\pm1.4}$ | $57.7_{\pm0.4}$ | $31.3_{\pm0.4}$ |
| CL-20 | LoRA-L | $75.9_{\pm0.8}$ | $52.4_{\pm0.9}$ | $82.7_{\pm0.7}$ | $60.8_{\pm1.0}$ | $76.8_{\pm0.3}$ | $91.3_{\pm0.5}$ | $51.7_{\pm0.5}$ | $60.4_{\pm0.9}$ | $61.5_{\pm0.3}$ | $31.6_{\pm0.3}$ |
| | LoRA | $71.1_{\pm0.7}$ | $50.9_{\pm0.5}$ | $80.3_{\pm1.0}$ | $59.4_{\pm0.7}$ | $64.6_{\pm0.7}$ | $91.1_{\pm0.7}$ | $52.0_{\pm0.4}$ | $62.3_{\pm0.6}$ | $62.3_{\pm0.2}$ | $31.3_{\pm0.1}$ |
| | LoRSU | $75.3_{\pm1.0}$ | $53.7_{\pm0.8}$ | $82.8_{\pm0.4}$ | $60.7_{\pm0.8}$ | $82.7_{\pm0.7}$ | $91.6_{\pm0.6}$ | $51.6_{\pm0.4}$ | $61.5_{\pm0.4}$ | $58.4_{\pm0.2}$ | $31.4_{\pm0.2}$ |
| | LoRA-Ppl | $75.5_{\pm0.9}$ | $51.6_{\pm0.7}$ | $82.0_{\pm0.4}$ | $59.3_{\pm0.6}$ | $74.9_{\pm0.3}$ | $91.6_{\pm0.5}$ | $51.7_{\pm0.6}$ | $62.8_{\pm0.5}$ | $57.0_{\pm0.1}$ | $32.1_{\pm0.1}$ |
| | LoRA-F | $74.8_{\pm0.7}$ | $52.7_{\pm1.0}$ | $81.6_{\pm0.8}$ | $59.4_{\pm0.9}$ | $71.5_{\pm0.7}$ | $91.0_{\pm0.8}$ | $51.7_{\pm0.7}$ | $63.4_{\pm0.8}$ | $58.9_{\pm0.2}$ | $31.0_{\pm0.2}$ |
| | LoRSU-Ppl | $74.1_{\pm1.0}$ | $52.0_{\pm0.9}$ | $82.5_{\pm0.7}$ | $59.8_{\pm0.8}$ | $79.0_{\pm0.7}$ | $92.1_{\pm0.7}$ | $51.8_{\pm0.9}$ | $61.8_{\pm0.4}$ | $58.7_{\pm0.4}$ | $31.6_{\pm0.3}$ |
| CL-50 | LoRA-L | $75.6_{\pm0.2}$ | $53.0_{\pm0.1}$ | $82.7_{\pm0.3}$ | $60.6_{\pm0.3}$ | $77.1_{\pm0.2}$ | $91.5_{\pm0.2}$ | $51.7_{\pm0.1}$ | $60.7_{\pm0.0}$ | $59.8_{\pm0.1}$ | $31.4_{\pm0.1}$ |
| | LoRA | $62.8_{\pm0.3}$ | $47.2_{\pm0.4}$ | $72.4_{\pm0.4}$ | $54.4_{\pm0.2}$ | $61.6_{\pm0.4}$ | $90.2_{\pm0.3}$ | $51.7_{\pm0.2}$ | $62.0_{\pm0.1}$ | $60.8_{\pm0.0}$ | $30.9_{\pm0.1}$ |
| | LoRSU | $75.4_{\pm0.3}$ | $53.9_{\pm0.1}$ | $83.1_{\pm0.2}$ | $60.3_{\pm0.1}$ | $83.1_{\pm0.1}$ | $92.1_{\pm0.1}$ | $51.6_{\pm0.2}$ | $61.2_{\pm0.0}$ | $57.6_{\pm0.0}$ | $31.1_{\pm0.0}$ |
| | LoRA-Ppl | $74.9_{\pm0.3}$ | $51.7_{\pm0.3}$ | $81.9_{\pm0.2}$ | $59.8_{\pm0.2}$ | $77.8_{\pm0.1}$ | $92.1_{\pm0.3}$ | $51.8_{\pm0.2}$ | $62.9_{\pm0.3}$ | $59.4_{\pm0.2}$ | $31.9_{\pm0.1}$ |
| | LoRA-F | $73.6_{\pm0.0}$ | $51.8_{\pm0.3}$ | $81.2_{\pm0.0}$ | $58.1_{\pm0.1}$ | $66.6_{\pm0.3}$ | $90.7_{\pm0.1}$ | $51.6_{\pm0.1}$ | $63.7_{\pm0.3}$ | $58.4_{\pm0.1}$ | $30.5_{\pm0.0}$ |
| | LoRSU-Ppl | $72.9_{\pm0.1}$ | $51.1_{\pm0.3}$ | $81.3_{\pm0.4}$ | $59.4_{\pm0.4}$ | $75.4_{\pm0.2}$ | $91.6_{\pm0.2}$ | $51.7_{\pm0.1}$ | $62.7_{\pm0.4}$ | $57.5_{\pm0.1}$ | $32.1_{\pm0.0}$ |

Table 25: Exact accuracy scores (%) for each baseline used to fine-tune the model on the *VSR* dataset under three different continual learning (5, 10, 50 shots) settings. We include error bars over 3 runs.

| Setting | Method | VQA Datasets (Acc %) | | | | | | | | | |
| --- | --- | --- | --- | --- | --- | --- | --- | --- | --- | --- | --- |
| | | GTS | TSI | CAn | AIR | ESAT | DALLE | VSR | HM | MMVP | VisOnly |
| | Zr-Shot | 75.6 | 53.1 | 82.7 | 60.4 | 76.1 | 91.1 | 51.5 | 61.2 | 58.0 | 31.3 |
| CL-5 | LoRA-L | $75.3_{\pm0.7}$ | $59.9_{\pm1.4}$ | $81.0_{\pm1.1}$ | $56.2_{\pm0.5}$ | $66.8_{\pm1.3}$ | $90.1_{\pm1.3}$ | $68.3_{\pm1.1}$ | $65.0_{\pm1.4}$ | $57.6_{\pm0.3}$ | $32.5_{\pm0.4}$ |
| | LoRA | $72.6_{\pm1.3}$ | $49.5_{\pm1.5}$ | $78.2_{\pm0.8}$ | $57.5_{\pm1.5}$ | $55.0_{\pm0.9}$ | $88.8_{\pm0.7}$ | $52.0_{\pm1.0}$ | $61.9_{\pm1.5}$ | $59.7_{\pm0.3}$ | $30.4_{\pm0.5}$ |
| | LoRSU | $75.6_{\pm0.7}$ | $52.2_{\pm1.4}$ | $82.2_{\pm0.6}$ | $60.1_{\pm0.9}$ | $77.9_{\pm0.6}$ | $91.1_{\pm1.1}$ | $51.9_{\pm1.3}$ | $62.2_{\pm1.5}$ | $58.4_{\pm0.3}$ | $31.7_{\pm0.3}$ |
| | LoRA-Ppl | $65.8_{\pm0.7}$ | $48.7_{\pm0.8}$ | $65.4_{\pm1.3}$ | $33.8_{\pm1.4}$ | $48.8_{\pm0.5}$ | $81.7_{\pm1.2}$ | $61.7_{\pm0.5}$ | $56.2_{\pm0.7}$ | $43.6_{\pm0.2}$ | $32.8_{\pm0.4}$ |
| | LoRA-F | $76.0_{\pm0.9}$ | $64.5_{\pm0.8}$ | $81.2_{\pm1.3}$ | $57.6_{\pm0.6}$ | $69.7_{\pm1.5}$ | $89.4_{\pm0.8}$ | $69.5_{\pm1.0}$ | $12.8_{\pm0.5}$ | $30.3_{\pm0.5}$ | $13.0_{\pm0.3}$ |
| | LoRSU-Ppl | $73.6_{\pm0.7}$ | $57.5_{\pm1.1}$ | $80.3_{\pm1.1}$ | $57.8_{\pm1.3}$ | $73.1_{\pm1.3}$ | $90.7_{\pm1.1}$ | $62.0_{\pm1.5}$ | $57.4_{\pm0.5}$ | $57.9_{\pm0.6}$ | $30.3_{\pm0.4}$ |
| CL-20 | LoRA-L | $77.1_{\pm0.8}$ | $54.7_{\pm0.9}$ | $84.5_{\pm0.9}$ | $61.4_{\pm0.5}$ | $75.5_{\pm0.7}$ | $90.9_{\pm0.8}$ | $73.7_{\pm0.5}$ | $64.5_{\pm0.8}$ | $56.9_{\pm0.2}$ | $32.6_{\pm0.4}$ |
| | LoRA | $72.6_{\pm0.7}$ | $54.5_{\pm0.9}$ | $76.6_{\pm0.8}$ | $57.4_{\pm0.7}$ | $57.3_{\pm0.4}$ | $87.9_{\pm0.8}$ | $51.9_{\pm0.7}$ | $59.0_{\pm0.5}$ | $57.6_{\pm0.2}$ | $31.3_{\pm0.4}$ |
| | LoRSU | $74.9_{\pm0.6}$ | $54.6_{\pm0.5}$ | $82.1_{\pm0.8}$ | $58.5_{\pm0.7}$ | $75.5_{\pm0.5}$ | $91.6_{\pm0.5}$ | $51.6_{\pm0.6}$ | $62.4_{\pm0.7}$ | $57.5_{\pm0.2}$ | $30.9_{\pm0.2}$ |
| | LoRA-Ppl | $74.9_{\pm0.4}$ | $62.2_{\pm0.4}$ | $82.4_{\pm0.3}$ | $58.2_{\pm0.7}$ | $70.5_{\pm0.7}$ | $89.0_{\pm0.6}$ | $71.0_{\pm0.8}$ | $64.8_{\pm0.5}$ | $55.8_{\pm0.2}$ | $28.6_{\pm0.2}$ |
| | LoRA-F | $75.4_{\pm0.5}$ | $60.6_{\pm0.5}$ | $80.9_{\pm0.9}$ | $56.6_{\pm0.9}$ | $63.1_{\pm0.7}$ | $88.2_{\pm0.6}$ | $74.8_{\pm0.5}$ | $48.7_{\pm0.9}$ | $50.1_{\pm0.4}$ | $20.2_{\pm0.2}$ |
| | LoRSU-Ppl | $72.6_{\pm0.8}$ | $52.7_{\pm0.5}$ | $81.6_{\pm0.8}$ | $60.3_{\pm0.5}$ | $69.4_{\pm0.7}$ | $89.6_{\pm0.5}$ | $74.4_{\pm0.9}$ | $62.5_{\pm0.8}$ | $57.1_{\pm0.3}$ | $29.7_{\pm0.4}$ |
| CL-50 | LoRA-L | $77.2_{\pm0.3}$ | $56.5_{\pm0.1}$ | $84.5_{\pm0.0}$ | $61.4_{\pm0.2}$ | $76.4_{\pm0.2}$ | $91.5_{\pm0.3}$ | $73.4_{\pm0.1}$ | $65.3_{\pm0.2}$ | $54.4_{\pm0.1}$ | $31.5_{\pm0.1}$ |
| | LoRA | $73.4_{\pm0.1}$ | $53.8_{\pm0.0}$ | $74.6_{\pm0.4}$ | $56.7_{\pm0.1}$ | $56.2_{\pm0.1}$ | $87.0_{\pm0.2}$ | $51.9_{\pm0.0}$ | $59.2_{\pm0.2}$ | $57.6_{\pm0.1}$ | $30.8_{\pm0.0}$ |
| | LoRSU | $75.3_{\pm0.1}$ | $54.7_{\pm0.1}$ | $81.6_{\pm0.1}$ | $58.3_{\pm0.2}$ | $75.7_{\pm0.1}$ | $91.4_{\pm0.4}$ | $53.8_{\pm0.2}$ | $62.1_{\pm0.3}$ | $57.3_{\pm0.1}$ | $30.8_{\pm0.0}$ |
| | LoRA-Ppl | $71.7_{\pm0.1}$ | $48.7_{\pm0.1}$ | $75.1_{\pm0.2}$ | $46.3_{\pm0.4}$ | $64.6_{\pm0.3}$ | $87.9_{\pm0.2}$ | $71.7_{\pm0.4}$ | $61.9_{\pm0.2}$ | $55.1_{\pm0.1}$ | $30.9_{\pm0.0}$ |
| | LoRA-F | $76.3_{\pm0.3}$ | $64.2_{\pm0.2}$ | $84.5_{\pm0.4}$ | $58.1_{\pm0.3}$ | $69.6_{\pm0.1}$ | $90.1_{\pm0.1}$ | $72.5_{\pm0.3}$ | $64.6_{\pm0.1}$ | $61.4_{\pm0.1}$ | $30.6_{\pm0.1}$ |
| | LoRSU-Ppl | $72.1_{\pm0.2}$ | $49.8_{\pm0.1}$ | $74.8_{\pm0.3}$ | $57.6_{\pm0.0}$ | $71.0_{\pm0.4}$ | $88.2_{\pm0.1}$ | $74.9_{\pm0.1}$ | $58.3_{\pm0.2}$ | $55.4_{\pm0.2}$ | $30.0_{\pm0.2}$ |

Table 26: Exact accuracy scores (%) for each baseline used to fine-tune the model on the *HM* dataset under three different continual learning (5, 10, 50 shots) settings. We include error bars over 3 runs.

| Setting | Method | VQA Datasets (Acc %) | | | | | | | | | |
|---|---|---|---|---|---|---|---|---|---|---|---|
| | | GTS | TSI | CAn | AIR | ESAT | DALLE | VSR | HM | MMVP | VisOnly |
| | **Zr-Shot** | 75.6 | 53.1 | 82.7 | 60.4 | 76.1 | 91.1 | 51.5 | 61.2 | 58.0 | 31.3 |
| **CL-5** | **LoRA-L** | $76.5_{\pm1.0}$ | $51.5_{\pm1.1}$ | $83.2_{\pm1.2}$ | $60.5_{\pm0.8}$ | $75.7_{\pm1.0}$ | $90.9_{\pm0.9}$ | $51.6_{\pm0.9}$ | $68.6_{\pm0.7}$ | $34.4_{\pm0.5}$ | $31.1_{\pm0.5}$ |
| | **LoRA** | $68.8_{\pm0.8}$ | $47.0_{\pm1.0}$ | $70.5_{\pm0.8}$ | $51.7_{\pm1.1}$ | $54.1_{\pm0.6}$ | $89.1_{\pm0.8}$ | $52.2_{\pm1.5}$ | $60.8_{\pm0.8}$ | $54.7_{\pm0.6}$ | $30.5_{\pm0.3}$ |
| | **LoRSU** | $75.7_{\pm1.2}$ | $54.1_{\pm1.1}$ | $82.9_{\pm0.6}$ | $60.7_{\pm1.0}$ | $76.3_{\pm1.1}$ | $92.2_{\pm0.6}$ | $51.5_{\pm0.9}$ | $61.8_{\pm1.2}$ | $58.1_{\pm0.2}$ | $31.9_{\pm0.5}$ |
| | **LoRA-Ppl** | $76.2_{\pm0.6}$ | $48.4_{\pm1.4}$ | $82.5_{\pm1.2}$ | $57.2_{\pm0.9}$ | $72.8_{\pm0.9}$ | $90.9_{\pm0.9}$ | $51.8_{\pm1.0}$ | $60.0_{\pm1.0}$ | $56.4_{\pm0.4}$ | $33.1_{\pm0.4}$ |
| | **LoRA-F** | $71.8_{\pm1.1}$ | $47.8_{\pm0.8}$ | $79.9_{\pm1.5}$ | $57.6_{\pm1.0}$ | $63.2_{\pm1.1}$ | $90.1_{\pm1.0}$ | $48.0_{\pm0.7}$ | $67.2_{\pm0.9}$ | $49.0_{\pm0.3}$ | $31.5_{\pm0.2}$ |
| | **LoRSU-Ppl** | $76.6_{\pm1.0}$ | $51.7_{\pm1.3}$ | $83.6_{\pm1.4}$ | $60.3_{\pm0.6}$ | $75.2_{\pm0.8}$ | $90.8_{\pm1.0}$ | $51.7_{\pm1.3}$ | $60.4_{\pm1.4}$ | $60.7_{\pm0.5}$ | $31.2_{\pm0.2}$ |
| **CL-20** | **LoRA-L** | $75.1_{\pm0.9}$ | $50.5_{\pm0.3}$ | $82.1_{\pm0.9}$ | $59.3_{\pm0.8}$ | $65.1_{\pm0.6}$ | $91.8_{\pm0.4}$ | $51.9_{\pm0.5}$ | $71.8_{\pm0.8}$ | $52.8_{\pm0.3}$ | $31.7_{\pm0.2}$ |
| | **LoRA** | $68.1_{\pm1.0}$ | $46.8_{\pm0.8}$ | $76.3_{\pm0.4}$ | $56.4_{\pm0.8}$ | $49.6_{\pm0.7}$ | $87.3_{\pm0.6}$ | $51.7_{\pm0.4}$ | $59.4_{\pm0.4}$ | $59.7_{\pm0.3}$ | $31.4_{\pm0.3}$ |
| | **LoRSU** | $76.1_{\pm0.8}$ | $53.0_{\pm0.7}$ | $82.7_{\pm0.5}$ | $60.4_{\pm0.6}$ | $75.7_{\pm0.4}$ | $92.1_{\pm0.7}$ | $51.8_{\pm0.8}$ | $61.9_{\pm0.5}$ | $58.4_{\pm0.2}$ | $31.5_{\pm0.2}$ |
| | **LoRA-Ppl** | $77.0_{\pm0.9}$ | $52.0_{\pm0.4}$ | $83.9_{\pm0.5}$ | $63.6_{\pm0.7}$ | $73.4_{\pm0.5}$ | $90.5_{\pm0.3}$ | $53.1_{\pm0.7}$ | $71.9_{\pm0.7}$ | $54.1_{\pm0.2}$ | $31.1_{\pm0.4}$ |
| | **LoRA-F** | $75.6_{\pm0.4}$ | $50.9_{\pm0.5}$ | $80.6_{\pm0.5}$ | $60.8_{\pm0.5}$ | $71.2_{\pm0.7}$ | $90.9_{\pm0.7}$ | $52.2_{\pm0.7}$ | $72.9_{\pm0.7}$ | $53.6_{\pm0.3}$ | $31.6_{\pm0.1}$ |
| | **LoRSU-Ppl** | $76.1_{\pm0.8}$ | $49.8_{\pm0.9}$ | $83.5_{\pm0.9}$ | $59.8_{\pm0.6}$ | $76.1_{\pm0.9}$ | $91.0_{\pm0.9}$ | $51.7_{\pm0.6}$ | $72.1_{\pm0.4}$ | $59.5_{\pm0.2}$ | $30.5_{\pm0.4}$ |
| **CL-50** | **LoRA-L** | $75.8_{\pm0.2}$ | $49.5_{\pm0.3}$ | $83.4_{\pm0.3}$ | $59.9_{\pm0.3}$ | $71.1_{\pm0.3}$ | $89.9_{\pm0.3}$ | $51.7_{\pm0.1}$ | $71.4_{\pm0.2}$ | $48.7_{\pm0.1}$ | $31.1_{\pm0.0}$ |
| | **LoRA** | $72.7_{\pm0.3}$ | $47.1_{\pm0.2}$ | $72.6_{\pm0.2}$ | $56.7_{\pm0.3}$ | $60.4_{\pm0.1}$ | $89.7_{\pm0.3}$ | $51.9_{\pm0.1}$ | $61.9_{\pm0.1}$ | $57.1_{\pm0.2}$ | $31.1_{\pm0.0}$ |
| | **LoRSU** | $75.3_{\pm0.3}$ | $53.2_{\pm0.1}$ | $83.3_{\pm0.2}$ | $60.5_{\pm0.1}$ | $74.9_{\pm0.1}$ | $92.2_{\pm0.1}$ | $51.6_{\pm0.2}$ | $61.5_{\pm0.0}$ | $58.9_{\pm0.1}$ | $31.3_{\pm0.0}$ |
| | **LoRA-Ppl** | $76.6_{\pm0.2}$ | $49.3_{\pm0.4}$ | $81.9_{\pm0.3}$ | $60.3_{\pm0.4}$ | $72.7_{\pm0.2}$ | $89.8_{\pm0.3}$ | $52.5_{\pm0.2}$ | $73.7_{\pm0.3}$ | $52.7_{\pm0.0}$ | $30.9_{\pm0.1}$ |
| | **LoRA-F** | $74.1_{\pm0.1}$ | $52.0_{\pm0.3}$ | $80.6_{\pm0.2}$ | $57.0_{\pm0.1}$ | $63.5_{\pm0.3}$ | $88.7_{\pm0.1}$ | $53.0_{\pm0.4}$ | $73.5_{\pm0.2}$ | $46.0_{\pm0.1}$ | $31.8_{\pm0.0}$ |
| | **LoRSU-Ppl** | $76.0_{\pm0.1}$ | $50.4_{\pm0.1}$ | $83.4_{\pm0.1}$ | $60.6_{\pm0.4}$ | $76.4_{\pm0.1}$ | $91.4_{\pm0.2}$ | $51.9_{\pm0.1}$ | $73.4_{\pm0.4}$ | $59.8_{\pm0.1}$ | $32.0_{\pm0.1}$ |

Table 27: Exact accuracy scores (%) for each baseline used to fine-tune the model on the *MMVP* dataset under three different continual learning (5, 10, 50 shots) settings. We include error bars over 3 runs.

| Setting | Method | VQA Datasets (Acc %) | | | | | | | | | |
|---|---|---|---|---|---|---|---|---|---|---|---|
| | | GTS | TSI | CAn | AIR | ESAT | DALLE | VSR | HM | MMVP | VisOnly |
| | **Zr-Shot** | 75.6 | 53.1 | 82.7 | 60.4 | 76.1 | 91.1 | 51.5 | 61.2 | 58.0 | 31.3 |
| **CL** | **LoRA-L** | 75.5 | 52.8 | 82.0 | 60.5 | 76.0 | 91.5 | 51.5 | 63.6 | 57.7 | 30.6 |
| | **LoRA-Ppl** | 75.5 | 53.6 | 83.0 | 60.3 | 75.6 | 91.1 | 51.5 | 63.1 | 60.7 | 31.7 |
| | **LoRA-F** | 75.2 | 52.9 | 81.3 | 60.5 | 74.3 | 90.4 | 51.6 | 63.6 | 60.0 | 31.4 |
| | **LoRSU-Ppl** | 75.1 | 52.0 | 81.2 | 57.4 | 75.2 | 90.2 | 51.7 | 63.9 | 60.3 | 30.8 |

Table 28: Exact accuracy scores (%) for each baseline used to fine-tune the model on the *VisOnly* dataset under three different continual learning (5, 10, 50 shots) settings. We include error bars over 3 runs.

| Setting | Method | VQA Datasets (Acc %) | | | | | | | | | |
|---|---|---|---|---|---|---|---|---|---|---|---|
| | | GTS | TSI | CAn | AIR | ESAT | DALLE | VSR | HM | MMVP | VisOnly |
| | Zr-Shot | 75.6 | 53.1 | 82.7 | 60.4 | 76.1 | 91.1 | 51.5 | 61.2 | 58.0 | 31.3 |
| CL-5 | LoRA-L | $76.5_{\pm1.2}$ | $51.9_{\pm0.7}$ | $82.4_{\pm1.4}$ | $60.5_{\pm1.5}$ | $76.1_{\pm1.0}$ | $91.5_{\pm0.6}$ | $51.6_{\pm0.9}$ | $60.3_{\pm1.0}$ | $57.6_{\pm0.3}$ | $31.3_{\pm0.4}$ |
| | LoRA | $70.9_{\pm1.4}$ | $52.1_{\pm1.2}$ | $77.5_{\pm1.3}$ | $55.6_{\pm0.6}$ | $52.6_{\pm0.8}$ | $89.3_{\pm0.6}$ | $51.7_{\pm0.8}$ | $61.7_{\pm0.7}$ | $56.9_{\pm0.6}$ | $30.9_{\pm0.5}$ |
| | LoRSU | $75.9_{\pm0.7}$ | $53.1_{\pm0.8}$ | $82.5_{\pm0.7}$ | $60.4_{\pm1.0}$ | $76.1_{\pm1.5}$ | $91.9_{\pm0.8}$ | $51.5_{\pm1.3}$ | $61.3_{\pm1.2}$ | $58.9_{\pm0.4}$ | $31.5_{\pm0.2}$ |
| | LoRA-Ppl | $76.3_{\pm1.1}$ | $50.7_{\pm1.1}$ | $82.2_{\pm0.9}$ | $61.0_{\pm1.3}$ | $73.4_{\pm0.9}$ | $91.7_{\pm1.3}$ | $52.1_{\pm1.1}$ | $59.3_{\pm1.3}$ | $58.0_{\pm0.2}$ | $35.0_{\pm0.5}$ |
| | LoRA-F | $76.0_{\pm0.8}$ | $51.1_{\pm1.4}$ | $82.9_{\pm1.1}$ | $59.9_{\pm0.7}$ | $71.2_{\pm1.2}$ | $91.7_{\pm1.1}$ | $51.6_{\pm1.3}$ | $60.8_{\pm0.7}$ | $58.4_{\pm0.2}$ | $34.9_{\pm0.4}$ |
| | LoRSU-Ppl | $76.2_{\pm1.1}$ | $53.0_{\pm0.9}$ | $83.4_{\pm0.7}$ | $61.3_{\pm1.4}$ | $76.6_{\pm0.8}$ | $92.3_{\pm0.5}$ | $52.0_{\pm1.0}$ | $61.6_{\pm0.7}$ | $60.7_{\pm0.3}$ | $32.0_{\pm0.5}$ |
| CL-20 | LoRA-L | $77.8_{\pm1.0}$ | $53.0_{\pm0.8}$ | $83.4_{\pm0.4}$ | $62.1_{\pm0.6}$ | $75.5_{\pm0.8}$ | $91.6_{\pm0.4}$ | $52.4_{\pm0.9}$ | $61.2_{\pm0.6}$ | $55.6_{\pm0.3}$ | $32.5_{\pm0.3}$ |
| | LoRA | $73.3_{\pm0.9}$ | $49.3_{\pm0.4}$ | $77.9_{\pm0.6}$ | $56.4_{\pm0.6}$ | $47.7_{\pm0.8}$ | $91.2_{\pm0.6}$ | $51.8_{\pm0.8}$ | $61.5_{\pm0.6}$ | $57.0_{\pm0.3}$ | $32.8_{\pm0.1}$ |
| | LoRSU | $75.7_{\pm0.5}$ | $53.3_{\pm0.7}$ | $82.0_{\pm0.5}$ | $60.0_{\pm0.5}$ | $76.1_{\pm0.6}$ | $91.9_{\pm0.9}$ | $51.7_{\pm0.6}$ | $61.6_{\pm0.3}$ | $58.2_{\pm0.3}$ | $31.5_{\pm0.4}$ |
| | LoRA-Ppl | $78.0_{\pm0.4}$ | $52.8_{\pm0.4}$ | $83.7_{\pm0.6}$ | $60.9_{\pm0.7}$ | $74.3_{\pm0.4}$ | $91.5_{\pm0.7}$ | $51.9_{\pm0.5}$ | $61.7_{\pm0.7}$ | $56.0_{\pm0.2}$ | $32.8_{\pm0.3}$ |
| | LoRA-F | $77.4_{\pm0.6}$ | $51.7_{\pm0.9}$ | $83.7_{\pm0.6}$ | $59.7_{\pm0.7}$ | $73.9_{\pm0.9}$ | $91.2_{\pm0.5}$ | $53.4_{\pm0.4}$ | $62.0_{\pm0.9}$ | $56.9_{\pm0.4}$ | $31.0_{\pm0.3}$ |
| | LoRSU-Ppl | $76.7_{\pm0.5}$ | $53.7_{\pm0.4}$ | $83.8_{\pm0.6}$ | $61.4_{\pm0.3}$ | $75.5_{\pm0.6}$ | $91.2_{\pm0.8}$ | $51.8_{\pm0.3}$ | $61.9_{\pm0.9}$ | $59.6_{\pm0.4}$ | $31.3_{\pm0.2}$ |
| CL-50 | LoRA-L | $76.4_{\pm0.4}$ | $54.5_{\pm0.3}$ | $84.1_{\pm0.3}$ | $61.3_{\pm0.0}$ | $73.9_{\pm0.1}$ | $91.5_{\pm0.1}$ | $51.9_{\pm0.3}$ | $62.8_{\pm0.1}$ | $55.4_{\pm0.0}$ | $32.1_{\pm0.1}$ |
| | LoRA | $70.0_{\pm0.1}$ | $46.8_{\pm0.0}$ | $70.5_{\pm0.1}$ | $51.0_{\pm0.2}$ | $50.9_{\pm0.0}$ | $88.1_{\pm0.0}$ | $52.0_{\pm0.3}$ | $61.2_{\pm0.3}$ | $57.8_{\pm0.2}$ | $31.7_{\pm0.1}$ |
| | LoRSU | $75.6_{\pm0.4}$ | $53.1_{\pm0.1}$ | $81.7_{\pm0.3}$ | $58.2_{\pm0.1}$ | $75.3_{\pm0.2}$ | $91.8_{\pm0.3}$ | $51.7_{\pm0.1}$ | $62.1_{\pm0.1}$ | $58.3_{\pm0.1}$ | $31.9_{\pm0.0}$ |
| | LoRA-Ppl | $76.9_{\pm0.4}$ | $54.6_{\pm0.1}$ | $84.1_{\pm0.3}$ | $60.5_{\pm0.2}$ | $74.9_{\pm0.4}$ | $91.2_{\pm0.3}$ | $51.8_{\pm0.3}$ | $62.5_{\pm0.3}$ | $56.0_{\pm0.1}$ | $33.0_{\pm0.0}$ |
| | LoRA-F | $77.1_{\pm0.0}$ | $53.0_{\pm0.2}$ | $83.9_{\pm0.4}$ | $60.9_{\pm0.1}$ | $73.1_{\pm0.1}$ | $92.2_{\pm0.3}$ | $51.9_{\pm0.2}$ | $61.4_{\pm0.4}$ | $58.0_{\pm0.0}$ | $32.5_{\pm0.1}$ |
| | LoRSU-Ppl | $76.1_{\pm0.3}$ | $51.5_{\pm0.2}$ | $81.6_{\pm0.1}$ | $60.2_{\pm0.0}$ | $75.6_{\pm0.2}$ | $92.2_{\pm0.3}$ | $52.0_{\pm0.2}$ | $61.2_{\pm0.3}$ | $58.3_{\pm0.0}$ | $33.5_{\pm0.1}$ |

Table 29: Ablation study over the effect of the rank $r$ used by *LoRSU* to fine-tune the image encoder, CLIP-L-14. We report the VQA accuracies of the last session in the *50-shot* CL setting. The accuracies on the target dataset are in red color. For this experiment, we use two attention heads to fine-tune with LoRSU.

| FT Dataset | rank ($r$) | VQA Datasets (Acc %) | | | | | | | | | |
|---|---|---|---|---|---|---|---|---|---|---|---|
| | | GTS | TSI | CAn | AIR | ESAT | DALLE | VSR | HM | MMVP | VisOnly |
| GTS | 8 | 83.0 | 53.2 | 81.3 | 60.9 | 61.0 | 91.2 | 51.5 | 61.6 | 60.0 | 31.6 |
| | 16 | 83.9 | 53.4 | 81.5 | 60.2 | 54.0 | 91.4 | 51.5 | 62.1 | 60.7 | 31.6 |
| | 32 | 84.8 | 53.1 | 81.9 | 60.5 | 58.0 | 90.6 | 51.6 | 61.8 | 58.7 | 31.5 |
| | 64 | 84.9 | 53.2 | 81.3 | 60.7 | 61.7 | 90.9 | 51.5 | 61.9 | 59.3 | 31.3 |
| | 128 | 84.3 | 53.2 | 81.8 | 60.6 | 56.8 | 91.5 | 51.6 | 61.8 | 58.7 | 31.2 |
| | 256 | 84.5 | 53.1 | 81.5 | 61.1 | 51.5 | 90.3 | 51.6 | 62.0 | 58.7 | 31.6 |
| TSI | 8 | 75.2 | 67.2 | 82.0 | 59.2 | 71.6 | 91.1 | 51.5 | 61.6 | 58.0 | 31.5 |
| | 16 | 75.4 | 68.0 | 82.3 | 59.1 | 71.0 | 90.6 | 51.6 | 61.6 | 56.7 | 31.2 |
| | 32 | 74.9 | 68.9 | 81.8 | 59.3 | 70.1 | 91.2 | 51.5 | 61.6 | 58.0 | 31.6 |
| | 64 | 75.3 | 72.1 | 82.0 | 59.3 | 72.3 | 90.5 | 51.6 | 61.4 | 58.0 | 31.6 |
| | 128 | 75.1 | 65.8 | 81.7 | 59.0 | 70.0 | 90.6 | 51.5 | 62.1 | 56.7 | 31.6 |
| | 256 | 75.4 | 66.4 | 82.3 | 59.6 | 72.0 | 91.2 | 51.5 | 62.1 | 56.7 | 31.5 |
| Zr-Shot | | 75.6 | 53.1 | 82.7 | 60.4 | 76.1 | 91.1 | 51.5 | 61.2 | 58.0 | 31.3 |

Table 30: Ablation study over the effect of the number of attention heads used by *LoRSU* to fine-tune the image encoder. We report the VQA accuracies of the last session in the *50-shot* CL setting. The accuracies on the target dataset are in red color. For this experiment, we use $r = 64$ for the rank of LoRSU.

| FT Dataset | # heads | VQA Datasets (Acc %) | | | | | | | | | |
|---|---|---|---|---|---|---|---|---|---|---|---|
| | | GTS | TSI | CAn | AIR | ESAT | DALLE | VSR | HM | MMVP | VisOnly |
| GTS | 0 | 83.1 | 52.7 | 82.2 | 60.8 | 60.6 | 91.1 | 51.6 | 61.7 | 59.3 | 31.6 |
| | 1 | 83.9 | 53.8 | 82.0 | 60.7 | 55.4 | 91.2 | 51.6 | 61.6 | 60.0 | 31.8 |
| | 2 | 84.9 | 53.2 | 81.3 | 60.7 | 61.7 | 90.9 | 51.5 | 61.9 | 59.3 | 31.3 |
| | 4 | 84.7 | 53.5 | 81.0 | 60.5 | 60.5 | 90.6 | 51.5 | 61.8 | 58.7 | 31.5 |
| | 8 | 84.9 | 52.9 | 81.2 | 60.5 | 58.8 | 90.5 | 51.5 | 61.6 | 59.3 | 31.5 |
| | 16 | 85.0 | 53.1 | 81.3 | 60.0 | 59.2 | 90.6 | 51.5 | 61.6 | 56.7 | 31.3 |
| TSI | 0 | 75.1 | 64.2 | 82.1 | 59.3 | 72.2 | 90.8 | 51.5 | 61.8 | 57.3 | 31.5 |
| | 1 | 75.3 | 64.8 | 81.9 | 59.5 | 74.0 | 90.5 | 51.5 | 61.6 | 58.0 | 32.0 |
| | 2 | 75.3 | 72.1 | 82.0 | 59.3 | 72.3 | 90.5 | 51.6 | 61.4 | 58.0 | 31.6 |
| | 4 | 74.9 | 66.8 | 82.2 | 58.9 | 74.0 | 90.5 | 51.5 | 62.1 | 58.0 | 31.4 |
| | 8 | 74.7 | 67.4 | 81.7 | 59.1 | 71.5 | 91.2 | 51.5 | 62.2 | 58.0 | 31.7 |
| | 16 | 75.3 | 65.2 | 81.8 | 59.9 | 69.1 | 90.5 | 51.5 | 61.6 | 58.0 | 31.3 |
| Zr-Shot | | 75.6 | 53.1 | 82.7 | 60.4 | 76.1 | 91.1 | 51.5 | 61.2 | 58.0 | 31.3 |

Table 31: Robustness comparison of LoRSU with respect to the number of training epochs. We consider LoRSU, *LoRSU-Rand* where the $k$ attention heads are chosen randomly and *LoRSU-AAH* where all the attention heads are chosen for fine tuning. We use *50 shots* on the *GTS* for each method and we report the Target Improvement ($TI$) on this dataset and the Control Change ($CC$) using only ESAT as a control dataset. We include error bars over 3 runs.

| # Epochs | LoRSU-Rand | | LoRSU-AAH | | LoRSU | |
|---|---|---|---|---|---|---|
| | TI ($\uparrow$) | CC ($\uparrow$) | TI ($\uparrow$) | CC ($\uparrow$) | TI ($\uparrow$) | CC ($\uparrow$) |
| 2 | $5.2_{\pm 0.9}$ | $-11.1_{\pm 1.1}$ | $6.1_{\pm 0.3}$ | $-11.6_{\pm 0.7}$ | $5.6_{\pm 0.4}$ | $-9.7_{\pm 0.8}$ |
| 5 | $7.6_{\pm 0.8}$ | $-15.0_{\pm 0.9}$ | $9.3_{\pm 0.4}$ | $-15.6_{\pm 0.6}$ | $8.6_{\pm 0.3}$ | $-12.6_{\pm 0.5}$ |
| 10 | $7.8_{\pm 0.5}$ | $-18.1_{\pm 0.8}$ | $9.1_{\pm 0.1}$ | $-19.6_{\pm 0.5}$ | $9.7_{\pm 0.1}$ | $-14.3_{\pm 0.7}$ |
| 20 | $5.9_{\pm 0.6}$ | $-20.0_{\pm 0.7}$ | $8.1_{\pm 0.1}$ | $-21.5_{\pm 0.6}$ | $7.4_{\pm 0.2}$ | $-15.7_{\pm 0.6}$ |

Table 32: Comparison of the importance of choosing a small subset of attention heads. The GTS dataset is used for fine-tuning. We include error bars over 3 runs. The highest accuracies across methods are underlined.

| Setting | Scores | LoRSU-Rand | LoRSU-AAH | LoRSU |
|---|---|---|---|---|
| CL-5 | TI ($\uparrow$) | $4.1_{\pm 0.4}$ | $5.9_{\pm 0.8}$ | $\underline{6.4}_{\pm 1.3}$ |
| | CC ($\uparrow$) | $-1.0_{\pm 0.5}$ | $-0.9_{\pm 0.3}$ | $\underline{-0.7}_{\pm 0.6}$ |
| CL-20 | TI ($\uparrow$) | $6.2_{\pm 0.6}$ | $7.5_{\pm 0.6}$ | $\underline{8.6}_{\pm 0.9}$ |
| | CC ($\uparrow$) | $-1.4_{\pm 0.3}$ | $\underline{-0.7}_{\pm 0.4}$ | $-1.0_{\pm 0.5}$ |
| CL-50 | TI ($\uparrow$) | $7.8_{\pm 0.4}$ | $9.1_{\pm 0.1}$ | $\underline{9.7}_{\pm 0.1}$ |
| | CC ($\uparrow$) | $-1.7_{\pm 0.2}$ | $\underline{-0.9}_{\pm 0.2}$ | $-1.3_{\pm 0.1}$ |

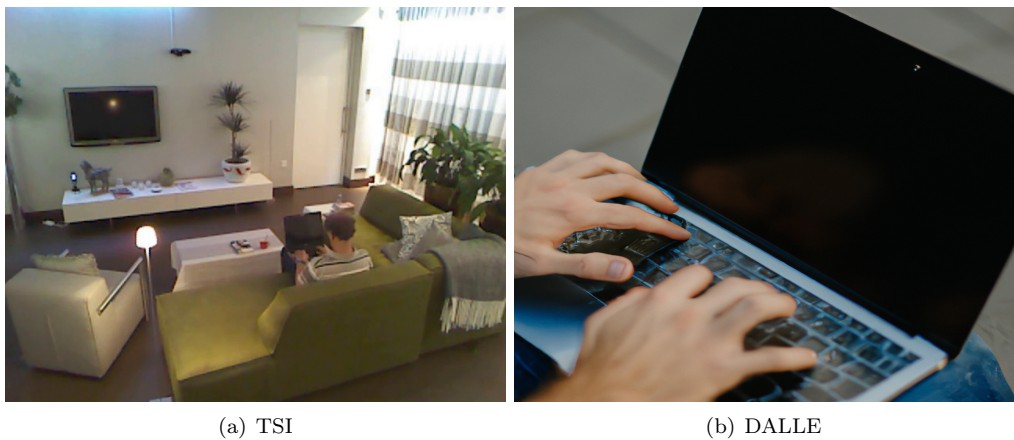

(a) TSI        (b) DALLE

Figure 5: Instances of the 'Use Laptop' action.

### E.3 Robustness on the Choice of Attention Heads

We show in Table 31 that LoRSU's mechanism of choosing the most important attention heads provides a clear advantage in terms of robustness over the other two LoRSU's variants, LoRSU-Rand and LoRSU-AAH. We can see that TI and CC decline in a lower rate compared to that of LoRSU-RAnd and LoRSU-AAH, as we increase the number of training epochs.. As expected, LoRSU-Rand appears to be the least robust method since the random choice of the attention heads constitute it more unstable.

## F  TSI vs. DALLE

In Figures 5 through 8, we present examples of images from TSI and DALLE for different actions. In general, we observe that TSI comprised of natural, unposed images of senior individuals performing daily tasks, reflecting real-life scenarios. The images are broader, showing the surrounding environment, which is crucial for context. On the other hand, DALLE images are idealized or stylized images. The focus is narrower, with emphasis on the object of the action (e.g. tablet, glass, etc.).

## G  Limitations

LoRSU is highly efficient, but it comes with a few key caveats. First, to date LoRSU has only been evaluated on CLIP-based encoders within LLaVA; extending it to other VLM architectures and image encoders remains future work, as does integrating smaller LLM proxies to reduce compute further . Finally, because it relies on binary masks to isolate updates, scaling LoRSU's masking strategy to much larger parameter spaces (e.g., full LLMs) poses challenges, and more scalable masking or parameter-selection mechanisms will be needed to apply it beyond vision encoders.

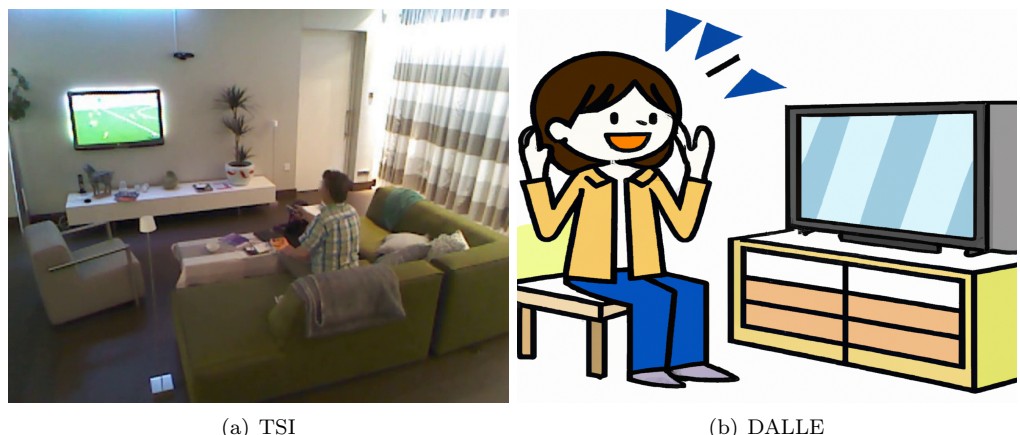

(a) TSI          (b) DALLE

Figure 6: Instances of the 'Watching TV' action.

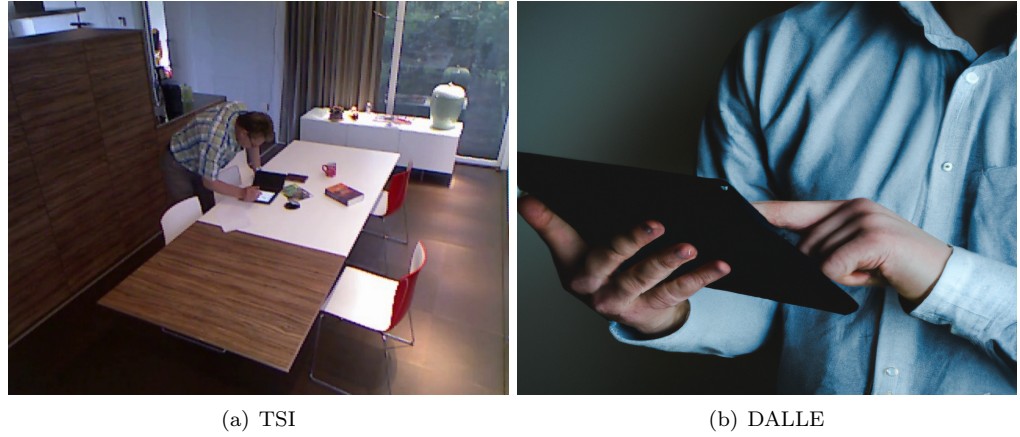

(a) TSI          (b) DALLE

Figure 7: Instances of the 'Use Tablet' action.

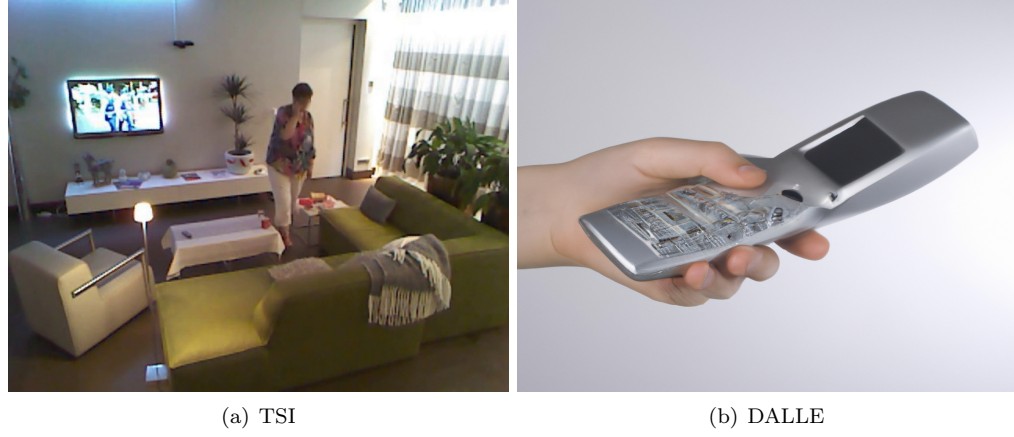

(a) TSI          (b) DALLE

Figure 8: Instances of the 'Use a telephone' action.

