# OpenReview forum: "Efficient Few-Shot Continual Learning in Vision-Language Models"
_TMLR — Accepted by TMLR_

### Review · Reviewer_7192 · 2025-09-01

**Summary Of Contributions:**

The paper introduces LoRSU (Low-Rank Adaptation with Structured Updates), a parameter-efficient fine-tuning (PEFT) method for few-shot continual learning (FSCL) in Vision-Language Models (VLMs). The primary goal is to adapt the VLM's vision encoder to new tasks from a few examples, without replaying old data and without catastrophically forgetting previously learned knowledge. The key idea is to selectively update a small subset of parameters in the vision encoder, specifically by applying structured sparse updates to MLP layers and low-rank adaptation (LoRA) to the most sensitive attention heads, identified via gradient magnitudes.

Key Strengths:

The paper tackles a highly relevant and practical problem: efficiently adapting large VLMs in dynamic, real-world settings where data is scarce.
The experimental evaluation is extensive, spanning over 700 experiments across 10 VQA datasets and multiple FSCL settings. The proposed method, LoRSU, demonstrates consistently strong empirical performance, outperforming numerous baselines in both target task improvement and knowledge retention (low forgetting).
The introduction of new VQA datasets (TSI, DALLE) to specifically probe the limitations of pretrained encoders is a valuable contribution to the community.

Key Weaknesses:

The conceptual novelty of the method feels somewhat limited. LoRSU appears to be a well-engineered combination of existing techniques—namely structured sparsity similar to SPU and selective low-rank adaptation—rather than a fundamentally new approach.
The theoretical justification for the method is superficial. While it proves that the chosen parameter selection strategy is optimal for a defined objective, it fails to provide a deeper theoretical link explaining why this objective leads to effective continual learning (e.g., reduced forgetting).

**Audience:**

Yes

**Audience Explanation:**

N/A

**Claims And Evidence:**

Yes

**Claims Explanation:**

The paper's claims of superior scalability, efficiency, and accuracy are well-supported by a comprehensive and rigorous set of experiments. The authors compare LoRSU against a wide array of relevant baselines, including full fine-tuning, regularization-based methods (EWC), and various PEFTs (LoRA, AdaLoRA, SPU). The results, presented in Tables 2, 3, and 4, clearly show that LoRSU consistently achieves a better trade-off between target task improvement (TI) and forgetting on control datasets (CC) across different few-shot settings (5, 20, 50 shots). The computational efficiency gains reported in Figure 3 are also convincing.

**Requested Changes:**

To strengthen the paper and secure a recommendation for acceptance, I believe the following changes are critical:

(Critical) Deepen the Theoretical Analysis: The current theoretical justification is insufficient. The authors should move beyond simply proving that their selection criterion optimizes their chosen objective. I strongly suggest they provide a more profound analysis that connects their method to the core challenges of few-shot continual learning.

Suggestion: Can the authors link the gradient magnitude criterion to the Fisher Information of the parameters? High-magnitude gradients might correspond to parameters with low Fisher Information for previous tasks, making them "safer" to update.
Suggestion: Could a connection be made to work on neural network pruning or the lottery ticket hypothesis? Perhaps the selected parameters form a "subnetwork" that is critical for the new task but less entangled with prior knowledge.
Suggestion: Explore the geometry of the loss landscape. Does this method guide the update in a direction that lies in a "flat" basin for old tasks while moving towards a sharp minimum for the new one?
A more substantive theoretical discussion that provides intuition and a principled reason for why the LoRSU update rule works for FSCL would transform the paper from a purely empirical one to one with significant conceptual depth.
(Critical) Re-evaluate and Clarify the Novelty: The authors should be more precise in positioning their contribution. Instead of presenting LoRSU as an entirely new paradigm, it would be more accurate and intellectually honest to frame it as a novel and effective synthesis of existing PEFTs.

Suggestion: In the Introduction and Related Work sections, explicitly state that LoRSU integrates principles from structured sparsity (like SPU) and low-rank adaptation (LoRA).
Suggestion: The core of the novelty could then be articulated as the insight that selectively applying these methods to different components of a transformer block (MLP vs. Attention) and using a gradient-based importance score for head selection is key to achieving a superior plasticity-stability balance in VLMs. This reframing does not diminish the paper's value but makes its claims more accurate.
In summary, the paper presents an empirically powerful method for a very important problem. However, its limited conceptual novelty and weak theoretical grounding prevent it from being a top-tier contribution in its current form. Addressing the requested changes, particularly by providing a deeper theoretical analysis, would significantly strengthen the submission.

---

> ### Author Response · Authors · 2025-10-19
> **Response to Reviewer 7192**
>
> We thank the reviewer for their evaluation and constructive feedback. We appreciate the recognition of our extensive empirical work and practical contributions. All the changes in the revised paper can be found in blue. Below, we address each requested change:
>
> ## Deepening the Theoretical Analysis
> Regarding the suggestion of "linking our gradient magnitude criterion to Fisher Information",  we appreciate this insightful suggestion, and we note an important connection: our gradient magnitude criterion is indeed fundamentally related to the empirical Fisher Information Matrix. The squared gradient magnitudes we use correspond to the diagonal entries of the empirical Fisher, which measures parameter sensitivity to the current task.
>
> However, our approach represents a conceptually distinct and inverse paradigm compared to traditional Fisher-based regularization methods like EWC. EWC/F-EWC approach estimates Fisher Information for previous tasks and uses it to regularize important parameters from changing, attempting to preserve past knowledge. On the other hand, LoRSU estimates Fisher Information (via gradients) for the current task and uses it to select which parameters to update, allowing task-relevant parameters to adapt while freezing others.
>
> This inverse analogy makes sense precisely because we start from a strong foundation model: rather than preventing important-to-past parameters from moving, we allocate capacity for the new task by identifying and updating only parameters with high sensitivity to current data.
>
> Our experimental setup includes F-EWC as a baseline (Tables 3, 8-12), and F-EWC consistently underperforms LoRSU across all datasets and settings. This empirical evidence suggests that backward-looking regularisation of past-important parameters is insufficient for effective continual learning in VLMs, while forward-looking allocation of parameters for the current task provides a more effective strategy. The failure of F-EWC indicates that Fisher-based regularisation alone cannot capture the dynamics needed for successful FSCL in VLMs, particularly when starting from a pretrained foundation model.
>
> The other suggested theoretical directions, like the lottery ticket hypothesis or loss landscape geometry, would each require substantial additional work beyond the scope of a revision. Lottery ticket hypothesis connections would necessitate extensive pruning experiments and analysis of subnetwork behaviour across multiple initialisation seeds. Loss landscape geometry exploration would require computing Hessian eigenvalues or conducting extensive interpolation studies across task sequences. Each would effectively constitute a separate research contribution requiring considerable time for additional experimentation and analysis.
>
> However, we have conducted additional analysis that provides empirical insight into why LoRSU achieves the plasticity-stability trade-off. Specifically, we computed the Jaccard similarity of the binary parameter masks selected by LoRSU across the 5 continual learning sessions (Fig 4 and Tbls 9-13):
>
> The consistent 15-25% parameter overlap across sessions suggests that LoRSU naturally discovers a sweet spot between plasticity and stability. The sufficient separation (75-85% distinct parameters) provides plasticity for learning new tasks without interfering with previous knowledge and the small overlap (15-25% shared parameters) ensures that some parameters are consistently utilized across tasks, creating shared representations. This prevents complete parameter isolation across tasks; if each task used entirely disjoint parameters, the model would fail to leverage the foundation model's general capabilities and would be vulnerable to catastrophic forgetting when later tasks happen to re-select previously frozen parameters.
>
> This empirical finding provides intuition for why gradient-based parameter selection works: high-magnitude gradients identify parameters that are both (1) critical for the new task and (2) exhibit moderate rather than extreme overlap with previous task parameters, balancing learning and retention.
>
> Importantly, this forward-looking parameter allocation mechanism appears more effective than backward-looking Fisher Information-based regularisation (as evidenced by F-EWC's poor performance). Whilst both approaches are based on Fisher Information, the key distinction is the direction: we allocate parameters for upcoming tasks rather than regularising for previous tasks, which proves more effective for FSCL in pretrained VLMs.
>
> We added a new subsection (Section 4.8 "Parameter Overlap Analysis") presenting these Jaccard similarity matrices and discussing how this small overlap pattern explains LoRSU's effectiveness.
>
> ## Clarify the Novelty
> We appreciate this feedback, and we are happy to include the reviewer's suggestion. We added this part  “"Our work builds ... with head selection based on cross-parameter gradient importance.” in the “Related Work” section.

---

### Review · Reviewer_NZmk · 2025-09-08

**Summary Of Contributions:**

The paper presents an efficient method LoRSU for few shot continual learning for Vision Language Models (VLMs) that improves downstream performance.

Strengths:
- Improves efficiency and downstream task performance of VLMs.
- The method is replay-free that is does not use the same example multiple times.
- Introduces novel Visual QA datasets to expose issues with pre-trained image encoders in VLMs.
- Provides theoretical justification of the importance score for attention heads.
- Measures both gain of new abilities in VLMs (TI) and retention of prior capabilities (CC).

Weaknesses:
- Some lack of clarification in experimental choices.
- Lack of clarification on the novelty of TSI dataset.

**Audience:**

Yes

**Audience Explanation:**

It presents an efficient method for few shot continual learning for VLMs with theoretical reasoning and shows improvements through experimental results in both downstream performance and reduction in Tflops and trainable parameters.

**Claims And Evidence:**

Yes

**Claims Explanation:**

- Clearly shows efficiency gain using TFlops and number of trainable parameters.
- Shows gains in downstream tasks using ACC and BWT on various VQA datasets.
- Shows improvement on new data using TI and retention of old capabilities using CC.

**Requested Changes:**

- Clarify if the TSI dataset is novel as the authors also provide a reference Das et al. (2019).
- Clarify is there is any reason to go forward with only Llava model.
- Clarify the differences of datasets used in Table 2,3 and 4. Why not show the results for all datasets in all experiments?

---

> ### Author Response · Authors · 2025-10-19
> **Response to Reviewer NZmk**
>
> We thank the reviewer for their positive assessment of our work and for highlighting the strengths of LoRSU. All the changes in the paper can be found in blue. We address each of the requested clarifications below:
>
> ## Clarification on TSI Dataset Novelty
> We appreciate the opportunity to clarify this point. The TSI (Toyota Smarthome Images) dataset is our novel contribution, created specifically for this work to evaluate VLM performance under visual domain shift. The confusion arises from our use of the Toyota Smarthome video dataset (Das et al., 2019) as the source material.
> What we did:
> - Started with the Toyota Smarthome video dataset (Das et al., 2019).
> - Extracted individual frames from these videos (2 fps, selecting middle frames).
> - Curated and filtered action classes to remove ambiguous categories.
> - Created VQA-style prompts for each action class.
> - Formatted this as a novel VQA dataset for vision-language model evaluation.
>
> The original Das et al. (2019) work is a video action recognition dataset. We transformed this into a VQA dataset specifically designed to expose CLIP encoder limitations when dealing with elderly individuals, blurred faces, low-resolution mounted camera footage, and real-world domestic settings. To our knowledge, this VQA formulation focused on visual domain shift in elderly activity recognition has not been presented before.
> We clarify this distinction in the Appendix C2 to avoid confusion.
>
> ## Rationale for Using Only LLaVA Model
> We chose to focus primarily on LLaVA for several strategic reasons: The most important reason was computational constraints; with over 700 experiments, testing multiple VLM architectures would have been computationally prohibitive on our available resources (single A100 GPU). (Representative architecture) LLaVA is widely adopted and uses CLIP-L-14, one of the most common vision encoders in VLMs, making our findings broadly relevant. (Validation across encoders) We did include MiniGPTv2 experiments (Table 1, Section 4.3) which uses EVA-CLIP-g-14, demonstrating that our findings about vision encoder updates generalise across different CLIP variants.
>
> We added a sentence in Section 4.2 (Implementation Details) explicitly stating these rationale points. Note that in the “Limitations section” (Appendix G), we already mention this: "Whilst we focus on CLIP and LLaVA due to computational constraints, our method is generic to any transformer model", Finally, in the Discussion (Section 5), we already emphasize that extending to other VLM architectures (e.g., Qwen2-VL, InternVL) is important future work.

---

### Review · Reviewer_3hPB · 2025-10-04

**Summary Of Contributions:**

The authors introduce Low-Rank adaptation with Structured Updates (LoRSU), a new technique for fine-tuning large models. Its key innovation is a "surgical" approach that identifies and updates only the most critical parameters within a model's vision encoder—specifically, the most important attention heads and MLP layer parameters—based on their gradient signals. This is combined with LoRA to ensure high efficiency.


They created two new visual question-answering (VQA) datasets, TSI and DALLE, specifically designed to test a model's robustness to "visual domain shift" (i.e., how well it handles images that look different from its original training data).

This work presents the first large-scale exploration of Few-Shot Continual Learning (FSCL) for generative visual-language models. They conducted an extensive evaluation (over 700 experiments) across ten datasets, comparing LoRSU against a wide range of state-of-the-art methods.

**Audience:**

Yes

**Audience Explanation:**

The CL community would be interested in a novel, replay-free method that demonstrates strong performance and reduced forgetting, especially in a challenging few-shot setting.

 Researchers working on PEFT methods (like LoRA, AdaLoRA) would be interested in a new, structured update approach that combines gradient-based parameter selection with low-rank adaptation.

**Broader Impact Concerns:**

The newly introduced TSI dataset, which "depict elderly people activities," could potentially contain or amplify societal biases related to age. The authors should add a statement acknowledging this possibility and describing steps taken to document and mitigate such biases in the dataset.

**Claims And Evidence:**

Yes

**Claims Explanation:**

This is supported by a large-scale empirical study (over 700 experiments) across ten diverse VQA datasets and three few-shot settings (CL-5, CL-20, CL-50). The results in Tables 2, 3, and 4 consistently show LoRSU achieving superior Target Improvement (TI) while maintaining Control Change (CC) close to zero, outperforming 12 strong baselines.

**Requested Changes:**

While ablation studies are mentioned, key results (e.g., the impact of the number of heads k, the LoRA rank r, and the comparison to random/all-head selection) should be summarized in the main text with a small table or figure. This is crucial for validating the design choices of LoRSU and should not be relegated entirely to the appendix.

---

> ### Author Response · Authors · 2025-10-19
> **Response to Reviewer 3hPB**
>
> We thank the reviewer for their feedback and positive assessment of our work. All the changes in the paper can be found in blue. We address the requested changes below:
>
> ## Ablation Studies in Main Text
> We appreciate this suggestion and agree that key ablation results should be more prominently featured. We added a concise summary table to Section 4.6, presenting:
> - Impact of LoRA rank (r): As shown in our appendix Table 23, performance peaks at r=64 for both GTS and TSI datasets, with graceful degradation at other values, demonstrating robustness.
> - Number of attention heads (k): Table 24 shows that k=2 provides an optimal balance between performance and efficiency, with diminishing returns beyond this point.
> - Head selection strategy comparison: Table 26 demonstrates that our gradient-based head selection (LoRSU) consistently outperforms both random selection (LoRSU-Rand) and updating all heads (LoRSU-AAH) across all CL settings, validating our structured update approach.
>
> We included a condensed version of this analysis in the main paper with a table highlighting these key findings, whilst maintaining full details in the appendix.
>
> ## TSI Dataset Bias Concerns
> We acknowledge this concern. The TSI dataset does focus on elderly individuals performing daily activities, which could potentially reflect or amplify age-related biases. We added a short statement in the appendix C.2 that explicitly notes that TSI depicts primarily elderly individuals in domestic settings, which may not represent the full diversity of how these activities are performed across age groups, abilities, and cultural contexts. Note, however, we just created a CL benchmark from the original Toyota SmartHome dataset by extracting frames from the videos; we did not collect any extra videos. More details about the dataset can be found in the original paper Das et al. 2019.

---

### Decision · Action_Editor_bbz5 · 2025-12-02

**Recommendation:** Accept as is

**Audience:**

Yes

**Audience Explanation:**

The manuscript addresses the task of continual learning of vision language models using few examples, which is a very relevant problem and will be of interest to the TMLR audience.

**Claims And Evidence:**

Yes

**Claims Explanation:**

The manuscript was reviewed by 3 reviewers. Finally, after the author's response and discussion round, one reviewer was satisfied and gave accept. The second reviewer appreciated the "technically sound and well-evaluated approach", but gave "Reject" primarily because of limited novelty. The third reviewer also mentioned this as a concern in the original review, and gave suggestions to better position the paper with respect to related work, which the authors did in their response.

Taking into account all the concerns and the discussions, I feel that though the concern of limited novelty is valid, the proposed framework with its extensive evaluation will be valuable to the community. So I would recommend accepting the paper.